# Quality control and correction method for air-temperature data from a citizen science weather station network in Leuven, Belgium

Eva Beele[1], Maarten Reyniers[2], Raf Aerts[3,4], Ben Somers[1,5]

[1] Division Forest, Nature and Landscape, University of Leuven (KU Leuven), Celestijnenlaan 200E-2411, BE-3001 Leuven, Belgium
[2] Royal Meteorological Institute of Belgium, Ringlaan 3, BE-1180 Brussels, Belgium
[3] Risk and Health Impact Assessment, Sciensano (Belgian Institute of Health), Juliette Wytsmanstraat 14, BE-1050 Brussels, Belgium
[4] Division Ecology, Evolution and Biodiversity Conservation, University of Leuven (KU Leuven), Kasteelpark Arenberg 31-2435, BE-3001 Leuven, Belgium
[5] KU Leuven Urban Studies Institute, University of Leuven (KU Leuven), Parkstraat 45-3609, BE-3000 Leuven, Belgium

*Correspondence to*: Eva Beele (eva.beele@kuleuven.be)

**Abstract.**

The growing urbanization trend and increasingly frequent extreme weather events urge further monitoring and understanding of weather in cities. In order to gain information on these intra-urban weather patterns, dense high quality atmospheric measurements are needed. Crowdsourced weather stations (CSW) could be a promising solution to reach such monitoring networks in a cost-efficient way. Because of their non-traditional measuring equipment and installation settings, the quality of these datasets remains an issue of concern. This paper presents crowdsourced data from the Leuven.cool network, a citizen science network of around 100 low-cost weather stations (Fine Offset WH2600) distributed across Leuven, Belgium (50°52'N 4°42'E). The dataset is accompanied by a newly developed station specific temperature quality control (QC) and correction procedure. The procedure consists of three levels removing implausible measurements, while also correcting for inter (in between stations) and intra (station-specific) station temperature biases by means of a random-forest approach. The QC method is evaluated using data from four WH2600 stations installed next to official weather stations belonging to the Royal Meteorological Institute of Belgium (RMIB). A positive temperature bias with strong relation to the incoming solar radiation was found between the CSW data and official data. The QC method is able to reduce this bias from $0.15 \pm 0.56$ °K to $0.00 \pm 0.28$ °K. After evaluation, the QC method is applied to the data of the Leuven.cool network, making it a very suitable data set to study in detail local weather phenomena such as the urban heat island (UHI) effect. (https://doi.org/10.48804/SSRN3F)

## 1 Introduction

More than 50% of the world population currently lives in urban areas and this number is expected to grow to 70 % by 2050 (UN, 2018). Keeping this growing urbanisation trend in mind and knowing that both the frequency and intensity of extreme weather events will increase (IPCC, 2021), it becomes clear that both cities and their citizens are vulnerable for climate change.

To plan efficient mitigation and adaptation measures, and hence mitigate future risks, information on intra-urban weather patterns is needed (Kousis et al., 2021). Dense high-quality atmospheric measurements are thus becoming increasingly important to investigate the heterogeneous urban climate. Due to their high installation and maintenance costs and strict siting instructions (WMO, 2018), official weather station networks are however sparse. As a results, most cities only have one or even no official station at all (Muller et al., 2015). Belgium only counts around 30 official weather stations distributed across a surface area of 30,689 km$^2$. 18 of them (Sotelino et al., 2018) are owned and operated by the Royal Meteorological Institute of Belgium (RMIB). These classical observation networks operate at a synoptic scale and are thus not suitable to observe city-specific or intra-urban weather phenomena such as the urban heat island (UHI) effect (Chapman et al., 2017).

The UHI can be measured by a number of methods. Fixed pair stations (e.g., Bassani et al., 2022; Oke, 1973) or mobile transect approaches (e.g., Kousis et al., 2021) have traditionally been used to quantify this phenomenon. Both methods are however not ideal as pair stations lack detailed spatial information while transects often miss a temporal component (Chapman et al., 2017; Heaviside et al., 2017). Other studies have quantified the UHI using remote sensing data derived from thermal sensors. Such methods can provide spatially continuous data over large geographical extents but are limited to land surface temperatures (LST) (Arnfield, 2003; Qian et al., 2018). As opposed to LST, canopy air temperatures (Tair) are however more closely related to human health and comfort (Arnfield, 2003). Finding the relationship between LST and Tair is known to be rather difficult and inconsistent (Yang et al., 2021). Numerical simulation models (e.g. UrbClim (De Ridder et al., 2015), SURFEX (Masson et al., 2013)) in which air temperature is continuously modelled over space and time could be a possible solution. They do however still have some drawbacks. Due to computational power capacity, models only take into account a limited number of variables, making them less suitable for real-life applications (Rizwan et al., 2008). Additionally, they often lack observational data to train and validate their simulations (Heaviside et al., 2017).

The rise of crowdsourced air-temperature data, especially in urban areas, could be a promising solution to bridge this knowledge gap (Muller et al., 2015). Such data are obtained through a large number of non-traditional sensors, mostly set up by citizens (cf. citizen science) (Muller et al., 2015; Bell et al., 2015). Crowdsourced datasets have already been successfully used for monitoring air temperature (Chapman et al., 2017; de Vos et al., 2020; Fenner et al., 2017; Napoly et al., 2018; Meier et al., 2017; Hammerberg et al., 2018; Feichtinger et al., 2020), rainfall (de Vos et al., 2019, 2020, 2017), wind speed (Chen et al., 2021; de Vos et al., 2020) and air pollution (EEA, 2019; Castell et al., 2017) within complex urban settings. Because of their non-traditional measuring equipment and installation settings, the quality of these datasets remains however an issue of concern (Bell et al., 2015; Napoly et al., 2018; Chapman et al., 2017; Meier et al., 2017; Muller et al., 2015; Cornes et al., 2020; Nipen et al., 2020). Quality uncertainty arises due to several issues: (1) calibration issues in which the sensor could be biased either before the installation or drifts over time, (2) design flaws in which the design of the station makes it susceptible to inaccurate observations, (3) communication and software errors leading to incorrect or missing data, (4) incomplete metadata (Bell et al., 2015) and (5) unsuitable installation locations (Feichtinger et al., 2020; Cornes et al., 2020).

Recent studies have therefore highlighted the importance of performing a data quality control in data processing applications (Båserud et al., 2020; Longman et al., 2018), especially before analysing crowdsourced air-temperature data (Bell et al., 2015; Jenkins, 2014; Chapman et al., 2017; Meier et al., 2017; Napoly et al., 2018; Cornes et al., 2020; Nipen et al., 2020; Feichtinger et al., 2020). Jenkins et al. (2014) and Bell et al. (2015) both conducted a field comparison in which multiple crowdsourced weather stations (CWS) were compared with official, and thus professional, observation networks. Both found a profound positive instrument temperature bias during daytime with strong relation to the incoming solar radiation. The use of crowdsourced data thus requires quality assurance and quality control (QA/QC) that both removes gross errors and corrects station-specific instrument biases (Bell et al., 2015). Using the findings of Bell et al. (2015) as a basis, Cornes et al. (2020) corrected crowdsourced air-temperature data across the Netherlands using radiation from satellite imagery and background temperature data from official stations belonging to the Royal Netherlands Meteorological Institute (KMNI). To investigate the UHI in London, UK, Chapman et al. (2017) used Netatmo weather stations and removed crowdsourced observations that deviated more than three standard deviations from the mean of all stations. Meier et al. (2017) developed a detailed QC procedure for Netatmo stations using reference data from two official observation networks in Berlin, Germany. The QC consists of four steps, each identifying and removing suspicious temperature data. Their methods highlight the need for standard, calibrated and quality-checked sensors in order to assess the quality of crowdsourced data (Cornes et al., 2020; Chapman et al., 2017; Meier et al., 2017). Such official sensors are however not present in most cities, hindering the transferability of these QC methods. To this end, Napoly et al. (2018) developed a statistically based QC method for Netatmo stations independent of official networks (the R-package *CrowdQC*). The QC method was developed on data from Berlin, Germany and Toulouse, France and was later applied to Paris, France to demonstrate the transferability of this method. The procedure consists of four main and three optional QC levels, removing suspicious values, correcting for elevation differences and interpolating single missing values. Since the *CrowdQC* filtered dataset still contained some radiative errors, Feichtinger et al. (2020) combined the methods of Napoly et al. (2018) and Meier et al. (2017) to study a high temperature period in August 2018 in Vienna. Most recently, Fenner et al. (2021) presented the QC R-package *CrowdQC+*, which is a further development of the existing package *CrowdQC* developed by Napoly et al. (2018). The core enhancements deal with radiative errors and sensor response time issues (Fenner et al., 2021).

Current QC studies mostly identify and remove implausible temperature measurements (Chapman et al., 2017; Meier et al., 2017; Napoly et al., 2018), instead of correcting for known temperature biases (Cornes et al., 2020). We do however know that both the siting and the design of CWS can introduce such a bias. By parameterising this bias, it can be learned and corrected for, hereby limiting the number of observations that is eliminated (Bell et al., 2015). Additionally, most QC procedures require data from official networks (Cornes et al., 2020; Chapman et al., 2017; Meier et al., 2017), while most cities do not have such measurements available (Muller et al., 2015). Lastly, previous research also noted that biases can be station specific, this because the design of a CWS is an important uncertainty source (Bell et al., 2015), indicating the need for station-specific

quality control methods. Thus, there is a need for station specific quality control and correction methods, independent of official weather station networks.

Here we report on a statistically-based QC method for the crowdsourced air-temperature data of the Leuven.cool network, a citizens science network of almost 100 weather stations distributed across private gardens and (semi-) public locations in Leuven, Belgium. The Leuven.cool network is a uniform network in the sense that only one weather station type (Fine Offset WH2600) is used for the entire network. To our knowledge, no quality control method has been developed for this sensor type. The stations were installed following a strict protocol, lots of metadata is available and both the dataflow and station siting are continuously controlled. This novel QC method removes implausible measurements, while also correcting for inter (in between stations) - and intra (station-specific) station temperature biases. The QC method only needs an official network during its development and evaluation stage. Afterwards the method can be applied independently of the official network that was used in the development phase. Transferring the method to other networks or regions would require the recalibration of the QC parameters. After applying this quality control and correction method, the crowdsourced Leuven.cool dataset becomes suitable to monitor local weather phenomena such as the urban heat island (UHI) effect.

The paper is organised as follows. Section 2 describes materials and methods, providing information on the study area, crowdsourced (Leuven.cool) dataset and official reference dataset. The development of the quality control method is explained in Section 3. In Section 4 the newly developed QC method is first tested on four crowdsourced stations installed next to three official stations from the Royal Meteorological Institute of Belgium (RMIB). This allows us to quantify the data quality improvement after every QC level. In Section 5 the QC method is applied to a network of CWS in Leuven, Belgium. Section 6 shortly focusses on the application potential of the dataset. Concluding remarks are summarized in Section 7.

## 2 Materials & methods

### 2.1 Study area

The QC method is developed for a citizens science weather station network "Leuven.cool", based in Leuven, Belgium (50°52'39" N 4°42'16" E 65m ASL). The Leuven.cool project is a close collaboration between the KU Leuven, the city of Leuven and the RMIB aiming to measure the micro-climate in Leuven and gain knowledge on the mitigating effects of green and blue infrastructures (Leuven.cool, 2020). Leuven has a warm temperate climate with no dry season and a warm summer (Cfb) with no influence from mountains or seas and overall weak topography (Kottek et al., 2006). Leuven is the capital and largest city of the province of Flemish Brabant and is situated in the Flemish region of Belgium, 25 kilometres east of Brussels, the capital of Belgium. The city comprises the districts of Leuven, Heverlee, Kessel-Lo, Wilsele and Wijgmaal, covering an area of 56.63 km$^2$. The main characteristics of the study area are summarized in Table 1.

**Table 1: Main characteristics of the study area Leuven**

| Climate | | |
|---|---|---|
| Annual Min/Mean/Max daily temperature (°C) | 6.9/11.2/15.5 | Leuven, 1991-2020 (RMI, 2020) |
| Mean annual rainfall (mm y$^{-1}$) | 780.7 | Leuven, 1991-2020 (RMI, 2020) |
| Köppen's classification | Cfb | (Kottek et al., 2006) |
| **Demographics** | | |
| Size (km$^2$) | 56.63 | Figure 2 |
| Population | 101 315 | (Demografie, 2021) |

## 2.2 Leuven.cool dataset

Data from the citizens science network Leuven.cool are presented in this paper. The crowdsourced weather station network consists of 106 weather stations distributed across Leuven and surroundings. The meteorological variables are measured by low-cost consumer weather stations produced by the manufacturer Fine Offset: the WH2600 wireless digital weather station (Figure 1). The station's specifications, as defined by the manufacturer, are summarized in Appendix A.1. The weather station consists of an outdoor unit (sensor array) and a base station. The outdoor sensor array measures temperature (°C, add 273.15 for K), humidity (%), precipitation (mm), wind speed (m/s), wind direction (°), solar radiation (W/m$^2$), and UV (-) every 16 seconds. This outdoor sensor array transmits its measurements wirelessly, through the 868 MHz radiofrequency, to the base station. This base station needs both power and internet - via a LAN connection - supply in order to send the data to a server. The data is forwarded to the Weather Observations Website (Kirk et al., 2020), a crowdsourcing platform initiated and managed by the UK Met Office. RMIB participates in this initiative and operates its own WOW portal (Weather Observations Website - Belgium). The outdoor unit is powered by three rechargeable batteries which are recharged by a small built-in solar panel. A radiation shield protects both the temperature and humidity sensors against extreme weather conditions and the direct exposure of solar radiation.

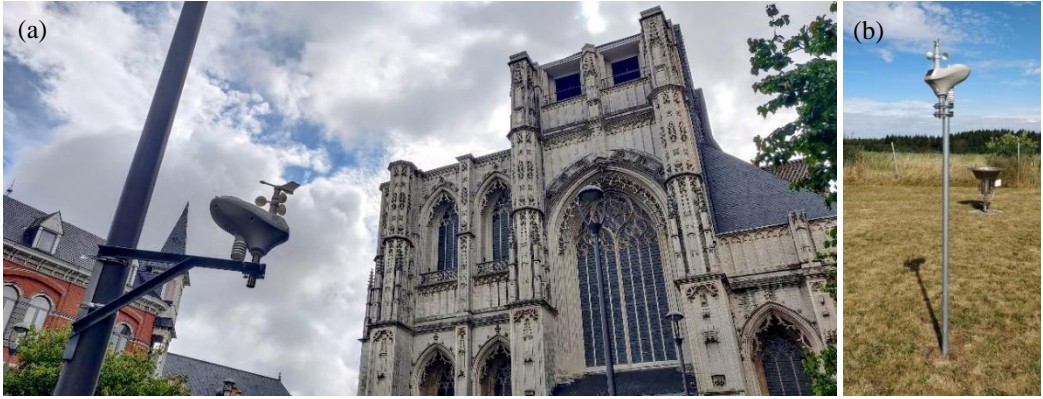

**Figure 1: The outdoor unit of the WH2600 wireless digital weather station at the Mathieu Layensplein in Leuven (LC-105) (a) and next to the official AWS equipment in Humain (LC-R05) (b). Pictures: Maarten Reyniers.**

From July 2019 onwards the weather stations were distributed along an urban gradient from green (private) gardens to public grey locations following a sampling design based on the concept of Local Climate Zones (LCZ) (Figure 2) (Stewart and Oke, 2012). This LCZ scheme was originally developed as an objective tool for classifying urban-rural gradients, herby capturing important urban morphological characteristics (Verdonck et al., 2018). Stewart & Oke (2012) formally define these zones as "regions of uniform surface cover, structure, material, and human activity that span hundreds of meters to several kilometres in horizontal scale".

Stewart & Oke (2012) define 17 LCZ classes, divided into 10 urban LCZs (1-10) and 7 natural LCZs (A-G). A LCZ map for Leuven was developed using a supervised random forest classification approach based on fine-scale land use, building height, building density and green ratio data. Details on this LCZ map are available in Appendix B. Table 2 summarises the LCZs present in Leuven and the number of weather stations in each LCZ class.

Table 2: LCZs present in Leuven and the number of weather stations in each LCZ class.

| LCZ ID | LCZ description | # Stations (106) |
| --- | --- | --- |
| LCZ 2 | Compact midrise | 20 |
| LCZ 3 | Compact low-rise | 7 |
| LCZ 4 | Open high-rise | 3 |
| LCZ 5 | Open midrise | 24 |
| LCZ 6 | Open low-rise | 19 |
| LCZ 8 | Large low-rise | 2 |
| LCZ 9 | Sparsely built | 16 |
| LCZ A | Dense trees | 0 |
| LCZ B | Scattered trees | 10 |
| LCZ D | Low plants | 4 |
| LCZ E | Bare rock or paved | 0 |
| LCZ G | Water | 1 |

It can be noted that the weather stations are not evenly distributed across the different LCZ classes, also the number of weather station within a LCZ class does not represent the spatial coverage of this LCZ class. Due to the complex urban settings in which the network is deployed, practical limitations apply to the eligible locations for installation. We rely on volunteering citizens, private companies and government institutions giving permission to install a weather station on their property. Further,

the middle-sized city of Leuven does not contain all available LCZ classes. In the urban context compact high-rise, lightweight low-rise and heavy industry is missing. In the natural context, brush or shrub vegetation and bare soil or sand are not present in sufficiently large areas (Table 2). Furthermore, the number of stations within more natural settings is limited due to the technical limitations of the weather stations; each outdoor unit needs a base station, with both power and LAN connection, within 50 to 100 meters in order to transmit its data. Lastly, the network was implemented with the intention of gaining knowledge on the mitigating effect of green and blue infrastructures within urban settings. The weather station network thus mostly focusses on urban classes.

The weather stations were installed according to a strict protocol. In private gardens the weather stations were installed at 2 meters height using a steel pole with a length of 2.70 meters. Dry concrete was used to anchor the pole into the soil at a depth of 70 cm. Following the station's guidelines, stations were installed at an open location within the garden, at least 1 meter from interfering objects, such as nearby buildings and trees. In order to maximize the absorption of solar radiation by the solar panel and to assure correct measurements of wind direction and precipitation, the weather station was levelled horizontally and the solar panel of the weather station was directed towards the south Weather stations located on public impervious surfaces were installed on available light poles using specially designed L-structures to avoid direct contact with the pole. For security reasons an installation height between 3-4 meters was used.

The data is currently available from July 2019 (2019Q3) until December 2021 (2021Q4). The dataset can be downloaded in periods of three months and is thus available for each quarter. The raw 16 seconds measurements are aggregated (temporally averaged) to 10 minutes observations. This is done for three reasons: (1) an extremely high temporal resolution of 16 seconds is too high for most meteorological analyses, (2) the aggregation to 10 minutes is necessary to exclude the natural small-scale variability and noise on the observations and most importantly (3) the reference dataset of official measurement is only available in a 10 minute resolution. After resampling the data, some basic data manipulation steps are performed to obtain the correct units and resolution for every meteorological variable. The final dataset contains air temperature with the three quality level stages (see further), relative humidity, dewpoint temperature, solar radiation, rain intensity, daily rain sum, wind direction, and windspeed. We must stress that only the air-temperature measurements undergo a quality check and correction procedure, further explained in the next sections. The variables other than temperature are, however, used in the correction procedure. A qualitative assessment of the data quality of these variables in included in Appendix C.

The maintenance of the network is controlled by PhD students and the technical staff at the Division of Forest, Nature & Landscape of the KU Leuven and with support of the RMIB. Since most of the weather stations are installed in private gardens, our volunteers keep an eye out for generic problems as well (e.g. leaves in the rain gauge, …).

## 2.3 Reference dataset

Standard, calibrated and quality controlled reference measurements are used to develop the QC method and evaluate its
performance. Since no official measurements are available in Leuven, we used data from three official RMIB stations in Uccle
(6447 – 50.80°N 04.26°E, alt 100 m), Diepenbeek (6477 - 50.92°N 5.45°E, alt 39 m) and Humain (6472 - 50.19°N 5.26°E, alt
295 m) (Figure 2).

The meteorological observation network of the RMIB consists of 18 automatic weather stations (AWS), ensuring continuous
data collecting and limiting human errors. These weather stations report meteorological parameters such as air pressure,
temperature, relative humidity, precipitation (quantity, duration), wind (speed, gust, direction), sunshine duration, shortwave
solar radiation and infrared radiation every 10 minutes. The AWS network is set up according to the WMO guidelines (WMO,
2018).

Since there is no AWS station available in the region of Leuven, four low-cost WH2600 weather stations were installed next
to the official and more professional equipment of the RMIB in Uccle, Diepenbeek and Humain. Since these stations will serve
as a reference, they were defined as LC-R01, LC-R02, LC-R04 and LC-R05 (Table 3). LC-R03 was installed for a short time
in Diepenbeek, but has been removed due to communication problems and is not taken into account in our further analysis.
Since January 2020, the oldest reference station LC-R01 is no longer active. This setup enables us to calculate the temperature
difference or bias between the low cost reference stations and the official RMIB stations in Uccle, Diepenbeek and Humain.

**Table 3: Specifics of Leuven.cool low-cost reference stations.**

| Station ID | Location | Installation date |
|---|---|---|
| Leuven.cool R01 | Uccle | 11/09/2018 |
| Leuven.cool R02 | Uccle | 02/09/2019 |
| Leuven.cool R04 | Diepenbeek | 06/11/2019 |
| Leuven.cool R05 | Humain | 20/08/2020 |

In the rest of the paper, the terminology of Table 4 is used to refer to the different datasets and stations.

**Table 4: Terminology of datasets and stations used in this paper**

| Terminology | Description |
|---|---|
| LC-X | The Leuven.cool (WH2600) stations installed in the study area (area of Leuven, Belgium) |
| LC-R | The Leuven.cool (WH-2600) stations installed next to the official weather stations operated by RMIB. |

| AWS | The Automatic Weather Stations owned and operated by RMIB. In our study, the AWS in Uccle, Diepenbeek and Humain are used. |

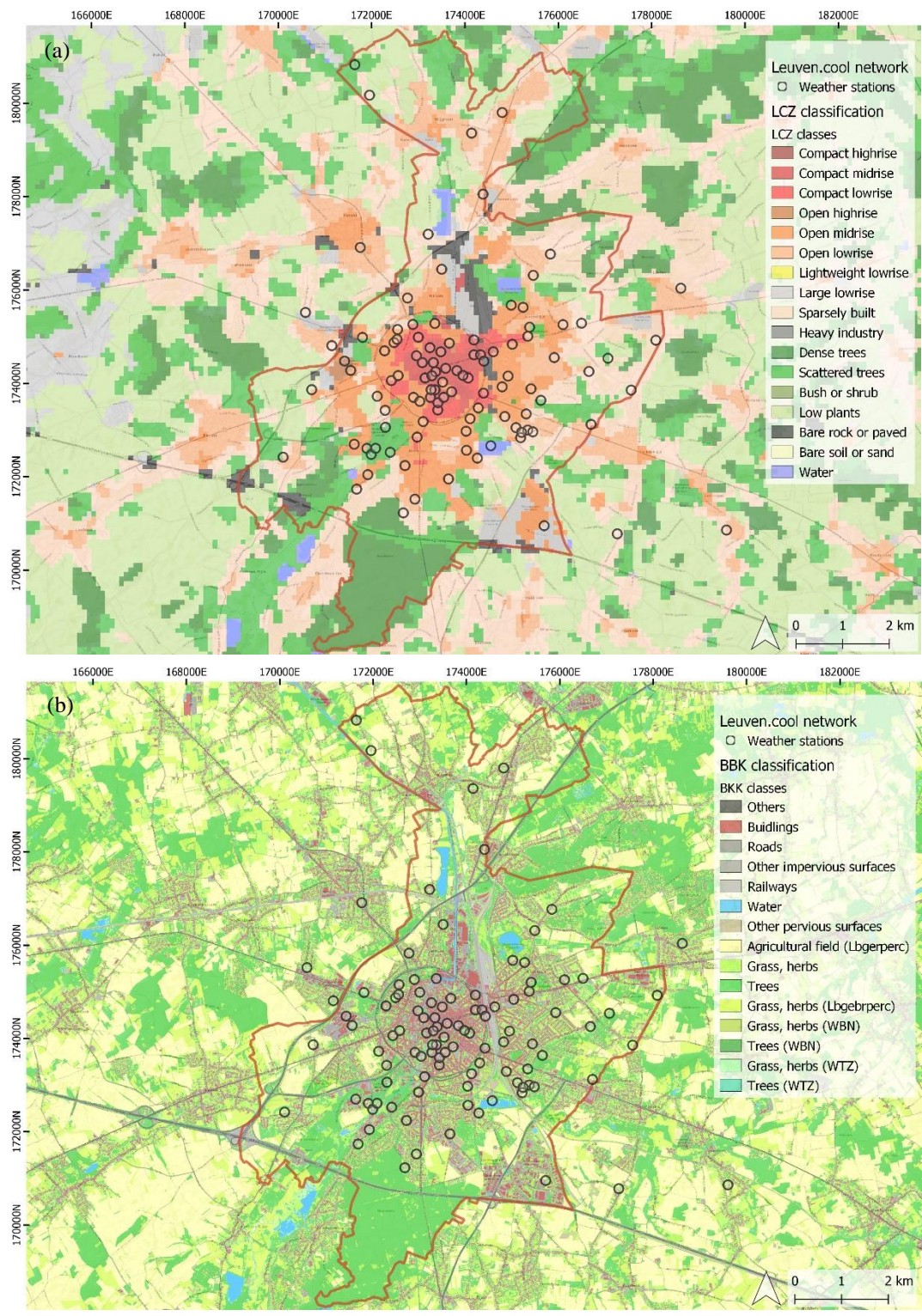

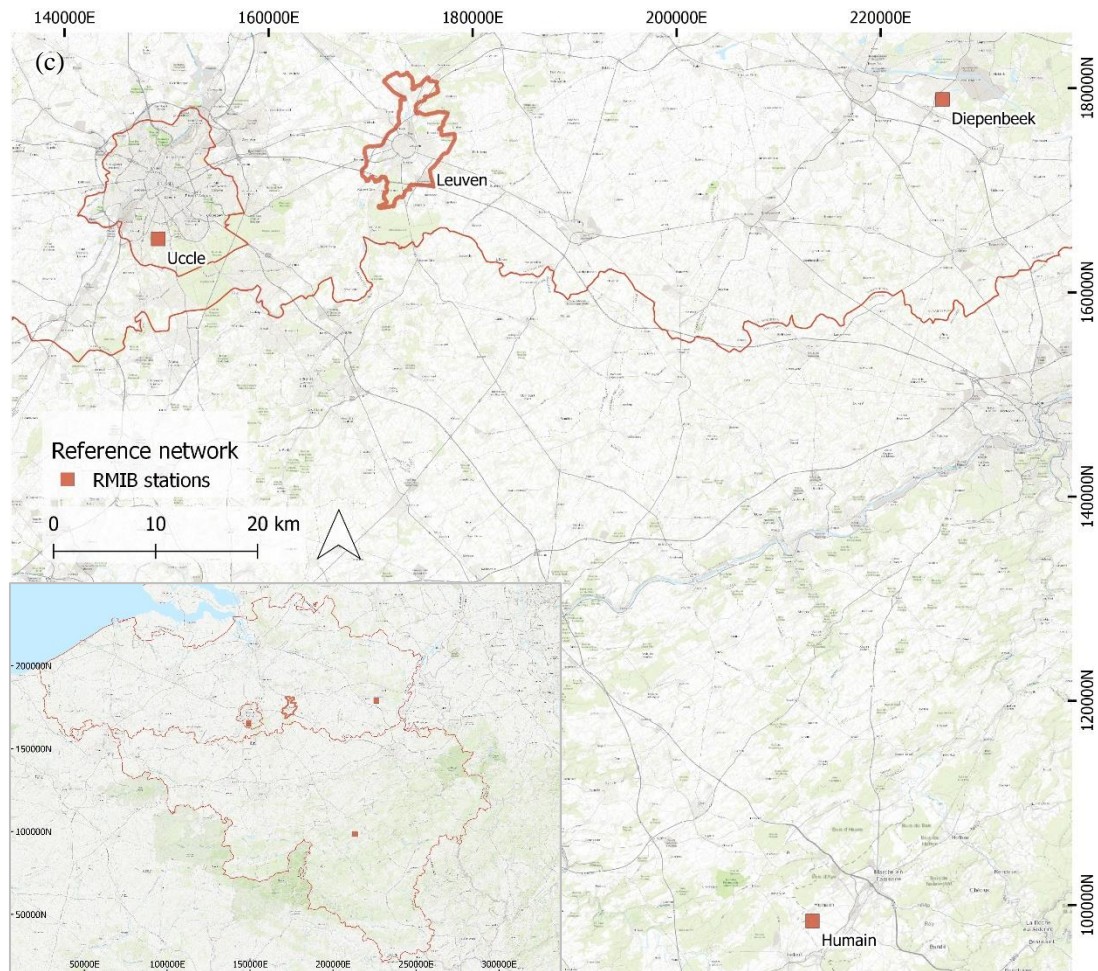

Figure 2: The Leuven.cool network (LC-X) with LCZ classification (a) and BBK (bodembedekkingskaart; land use map) classification (b) and Belgium delineated by the three official regions (Flanders, Wallonia and Brussels) with the location- of Leuven and the three RMIB stations (AWS) used in this study (c). Background map: Esri.

## 3. Description of the quality control and correction method

The newly developed QC control method consists of three levels (Table 5), mostly focussing on eliminating calibration issues, design flaws and communication or software errors. Due to the strict installation protocol used for the Leuven.cool station network, some of the typical uncertainty sources are a priori discarded. Both the location and metadata of each station were controlled by experts, eliminating incomplete metadata or unsuitable installation locations. We further know that the low cost station used in this study has some design flaws (e.g. during clear sky with low sun conditions both the radiation and

thermometer sensors experience shadow from the anemometer). Our correction method, however, is designed in such a way that these errors will be accounted for.

The first QC level removes implausible values mostly caused by software or communication errors. The second and third level correct for temperature biases. Both fixed inter (in between stations) biases due to sensor calibration uncertainties and variable

intra (station-specific) biases due to the station's design and siting are parameterized and corrected for.

**Table 5: Quality control levels, criteria for data filtering and potential error sources for crowdsourced air-temperature measurements.**

| Quality control level | Description | Potential error sources |
|---|---|---|
| L1 Outlier detection | | |
| L1.1 Range test | Range check against climatological extremes | Sensor malfunctioning |
| L1.2 Temporal outliers | Ensure realistic change in magnitude between consecutive observations of a specific station | Battery loss, server failure, connection issues, sensor malfunctioning |
| L1.3 Spatial outliers | Ensure realistic observation compared to neighbouring stations | Battery loss, server failure, sensor malfunctioning, outdoor sensor set up inside (not applicable in our setup due to the installation by team members) |
| L2 Inter station bias correction | Model the fixed in-between temperature bias | Sensor calibration issues |
| L3 Intra station bias correction | Model the variable station-specific temperature bias | Design flaws, outdoor sensor set up in sunlit conditions (no active ventilation) |

### 3.1 Quality control level 1 – Outlier detection

The outlier detection algorithm uses a flag system in which every 10 min observation is assigned flag of either 0, 1 or -1 referring to "no outlier", "outlier" or "not enough information to determine whether observation is an outlier". The outlier detection method consists of three steps: a range test, a temporal outlier test and a spatial outlier test. The thresholds of the parameter settings used during each of these steps are explained in Table 6. We used an iterative procedure for threshold

optimization. Observations which received a flag of 1 and are thus defined as outliers are set to NA in the quality controlled
(QC) level 1 dataset, hereby not considered during the following QC levels.

**Table 6: Parameter settings for QC level 1 - Outlier detection.**

| Outlier parameter | Value (unit) | Description |
|---|---|---|
| **Range outliers (RO)** | | |
| dev_reference | 1 (°C) | Max allowed deviation between climatological min and max temperature of AWS stations in Uccle/Diepenbeek/Humain and LC-R in Uccle/Diepenbeek/Humain |
| dev | 5 (°C) | Max allowed deviation between climatological min and max temperature of AWS station in Uccle and LC-X in Leuven |
| **Temporal outliers (TO)** | | |
| $TOa_{ThresMin}$ | -3 (°C) | Min allowed difference between sequential 10 min observations |
| $TOa_{ThresMax}$ | 2.5 (°C) | Max allowed difference between sequential 10 min observations |
| $TOb_{ThresMin}$ | 0.05 (°C) | Min difference that should be noted in $TOb_{Timespan}$ |
| $TOb_{Timespan}$ | 19 (-) | Number of consecutive 10 min observations in which temperature should change with $TOb_{ThresMin}$ |
| **Spatial outliers (SO)** | | |
| range | 2500 (m) | Range used to define neighbouring stations |
| $SO_{ThresMin}$ | -3 (-) | Min allowed Z-score |
| $SO_{ThresMax}$ | 3 (-) | Max allowed Z-score |
| nstat | 1 (-) | Minimum requirement of measurements in range |

### 3.1.1 QC level 1.1 - Range outliers

During QC L1.1 a range test based on climatology is performed. Range outliers can occur when a station is malfunctioning or
installed in a wrongful location. The latter has been largely eliminated by the installation protocol described in Section 2.
Observations are flagged as 1 whenever they exceed the maxima or minima climate thresholds, plus/minus an allowed
deviation ($T_{max/min\_AWS} \pm dev$). In this study these thresholds are based on historical data from nearby official weather stations
while the allowed deviations from the climate thresholds are based on local knowledge on environmental phenomena The
thresholds are calculated as the maximum and minimum temperature from the official AWS station in Uccle within the 3
month period that currently undergoes the QC. Observations receive a flag equal to -1 when the no temperature observation is
available.

### 3.1.2 QC level 1.2 - Temporal outliers

In QC L1.2 temporal outliers are detected using both (a) a step test and (b) persistence test. Temporal outliers occur when an
observation of a specific station is not in line with the surrounding observations of this station. The step test ensures that the change in magnitude between two consecutive observations lies within a certain interval; the test checks the rate of change and flags unrealistic jumps in consecutive values. Flags are set to 1 when observations increase more than 2.5 °C ($TOa_{ThresMax}$) or decrease more than 3 °C ($TOa_{ThresMin}$) in 10 minutes. Such steep increases or decreases in temperature are found when a station reconnects with its receiver after a period of hitches. The values of $TOa_{ThresMin}$ and $TOa_{ThresMax}$ differ for meteorological
reasons. The cooling down of air temperatures will, from a meteorological point of view, occur faster (e.g. through the passing of a cold front or thunderstorm) compared to the heating up of air temperatures (Ahrens, 2009). Observations are assigned a flag equal to -1 when the difference between sequential observations cannot be calculated.

The persistence test, on the other hand, makes sure that observations change minimally with time. Here we detect stations with
connection issues, transmitting the same observation repeatedly. Observations changing less than 0.05 °C ($TOb_{ThresMin}$) within 3 hours ($TOb_{Timespan}$) are flagged as 1. Whenever the difference between sequential observations cannot be calculated an observation gets a flag equal to -1.

### 3.1.3 QC level 1.3 - Spatial outliers

QC L1.3 detects spatial outliers in the dataset. Spatial outliers occur when the observation of a specific station is too different
compared to the observations from neighbouring stations. First neighbouring stations are defined as stations located within a 2.5 km radius (range). Next the Z-score or standard score is calculated for each observation following Eq. (1)

$$Z = \frac{x - \mu}{\sigma} , \tag{1}$$

where x is the observed value, $\mu$ the mean value and $\sigma$ the standard deviation across all neighbours. This standard score can be explained as the number of standard deviations by which the observed value is above or below the mean value of what is
being observed. Whenever the Z-score is lower than -3 ($SO_{ThresMin}$) or higher than 3 ($SO_{ThresMax}$) the observation is seen as a spatial outlier and receives a flag equal to 1. When there are no neighbours available within the predefined range, or the Z-score cannot be calculated, each observation is flagged with -1.

### 3.2 Quality control level 2 – Inter station bias correction

The second quality control level corrects the data for the fixed offset or inter station temperature bias between the weather
stations. This step is necessary since the temperature sensor are only calibrated by the manufacturer, and small calibration differences are expected for this consumer-grade weather sensor. Moreover, the Leuven.cool stations originate from different production batches, with possible hardware changes in the electronics. Calibration tests between multiple LC-X stations in the

same controlled environment were both technically and logistically not feasible. Simultaneous measurements are only available for two LC-R stations (LC-R01 and LC-R02 at the AWS of Uccle) for a period of four months, showing that sensor differences indeed exist and are non-negligible. A temperature difference of 0.2°C was found, which cannot be explained by the resolution of the temperature sensor (0.1°C).

In order to quantify this inter station temperature bias, a rather pragmatic approach was followed to mimic a controlled environment: we selected episodes for which a similar temperature across the study area is expected. Such episodes occur under breezy cloudy conditions with no rainfall (Arnfield, 2003; Kidder and Essenwanger, 1995). In practice, the database is searched for suitable episodes every 6 months, currently ranging from 2019S2 to 2021S2. All 10 minute observations are resampled to 2 hour observations, hereby calculating the mean temperature, windspeed, radiation and rainfall across all weather stations. Next, suitable episodes are found by selecting episodes where the average rainfall intensity equals 0 mm/h and the average radiation lies below 100 W/m2. The selected episodes are ordered on average windspeed and limited to the top 10 results.

For these episodes, one can assume the temperature to be very uniform over the study area, and solely controlled by altitude (Lu Aigang et al., 2009). In practise, only episodes with a high correlation between temperature and altitude (> 0.7) are retained. By regressing temperature versus altitude for every episode and calculating the residuals i.e. the difference between the observed and predicted temperature, a fixed offset for each station and every episode is obtained. Finally, the median offset across all episodes is considered as the true offset for each station. These offsets are added to the QC level 1 temperature data in order to obtain the corrected QC level 2 temperature data.

**3.3 Quality control level 3 – Intra station bias correction**

During the third quality control level the QC level 2 temperature data is further corrected for the variable intra station temperature biases. This bias is present in the data since the measurements are made with non-standard equipment as compared to the AWS measurements (e.g., passive instead of active ventilation, dimension of the Stevenson screen). These biases change during day and night time, and according to their local environment (e.g., radiation and windspeed patterns). (Bell et al., 2015).

By identifying the climatic variables mostly correlated with the temperature bias between the low-cost reference stations (LC-R) and the official RMIB stations in Uccle, Diepenbeek and Humain (AWS), a predictor for temperature bias is created. To produce a robust model, data from all low-cost reference stations (LC-R01, LC-R02, LC-R04 and LC-R05), ranging from their installation data until December 2021, were used simultaneously to create a predictor for the intra station temperature bias.

For the construction of a predictor model, the dataset was randomly split in training (0.60) and validation (0.40) data. The training data was used to train simple regression models, multiple regression models, random forest (RF) models and boosted

regression trees (BRT). Since previous research (Bell et al., 2015; Jenkins, 2014; Cornes et al., 2020) has shown that both radiation and wind speed highly influence the temperature bias, the simple and multiple regression models are mostly based on these variables. A previous study by Bell et al. (2015) also suggested that past radiation measurements, using an exponential weighting, are an even better prediction of the temperature bias, resulting in an advanced correction model (Bell et al., 2015).

The potential predictor models are validated using the validation data, ensuring an fair evaluation of the model. The coefficient of determination ($R^2$) and root mean square error (RMSE) are calculated to identify the most optimal prediction model for the temperature bias. After validation, the prediction model can be applied to the weather station network in Leuven (LC-X), hereby providing a temperature bias for each observation of every station in function of its local climatic conditions. The predicted temperature bias is subtracted from the QC level 2 temperature data to obtain the QC level 3 corrected temperature

dataset.

### 3.3.1 The intra station temperature bias

The overall temperature bias (i.e., all LC-R stations together) between the LC-R and the AWS data has a mean value of 0.10°C and a standard deviation of 0.55 °C (Figure 3). By splitting up the temperature bias for day (radiation > 0 W/m$^2$) and night (radiation = 0 W/m$^2$), a positive mean temperature bias during daytime (0.32 °C) and a negative mean temperature bias during

night time (-0.10 °C) is obtained. Figure 3 further suggests a higher standard deviation during daytime (0.61 °C) compared to night time (0.37 °C), both with a remarkably skewed (and opposite) distribution.

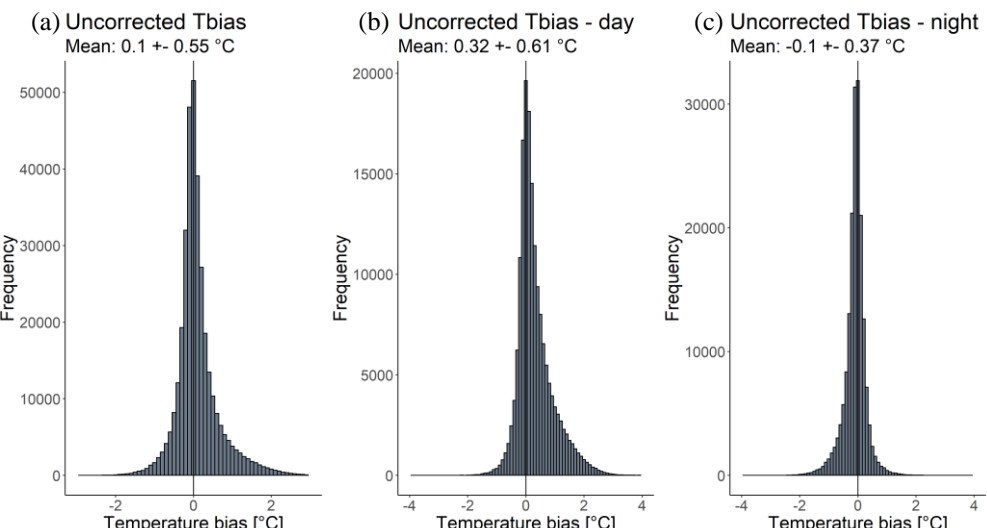

**Figure 3: Histograms of the temperature bias between the low-cost reference stations (LC-R) and the official RMIB stations (AWS)**
**for day and night (a), daytime defined by a radiation > 0 W/m$^2$ (b) and night time defined by a radiation = 0 W/m$^2$ (c). Mean biases and their standard deviations are given above the graphs. Note that the ranges on the y axis differ for the different subplots. The temperature bias was calculated for all measurements between the installation data of each LC-R and December 2021.**

To get a better understanding of the monthly and daily patterns of this temperature bias, Figure 4a shows the mean temperature bias in function of month and hour of the day. The stations show clear diurnal and seasonal patterns, confirming the positive temperature bias during daytime and negative temperature bias during night time previously observed in Figure 3. In general, we see a positive bias that is high around midday and is more pronounced during the summer months, lasting for several hours during the day. The night-time temperature bias is low for all months. A temperature bias of 0°C is reached for every month at a certain time of the day, the specific time at which this minimal temperature bias occurs, depends however on the season.

Figure 4b shows the mean temperature bias in function of windspeed and radiation. As expected, a positive temperature bias is noticed for high solar radiation and low windspeed conditions. The rather strange high values at low radiation and high wind speed can be explained as outliers (Figure 4c). Figure 4c shows the sample size of each cell. After removing cells with sample size lower than 10, the final graph is obtained (Figure 4d). The shallow local minimum seen around noon during the summer months (Figure 4a) and the fact that the largest biases are found in the middle of the radiation range rather than at the top (Figure 4d) are probably related to the station design itself. Two effects are at play here: (1) the placement of the radiation sensor and (2) the placement of the temperature sensor. For (1), certainly for lower solar elevations (during winter), the wind vane drops its shadow at the radiation sensor for a short time of the day. For (2), which we might consider more important, the temperature sensor is more shaded around midday (highest radiation) by the body of the station (during all seasons).

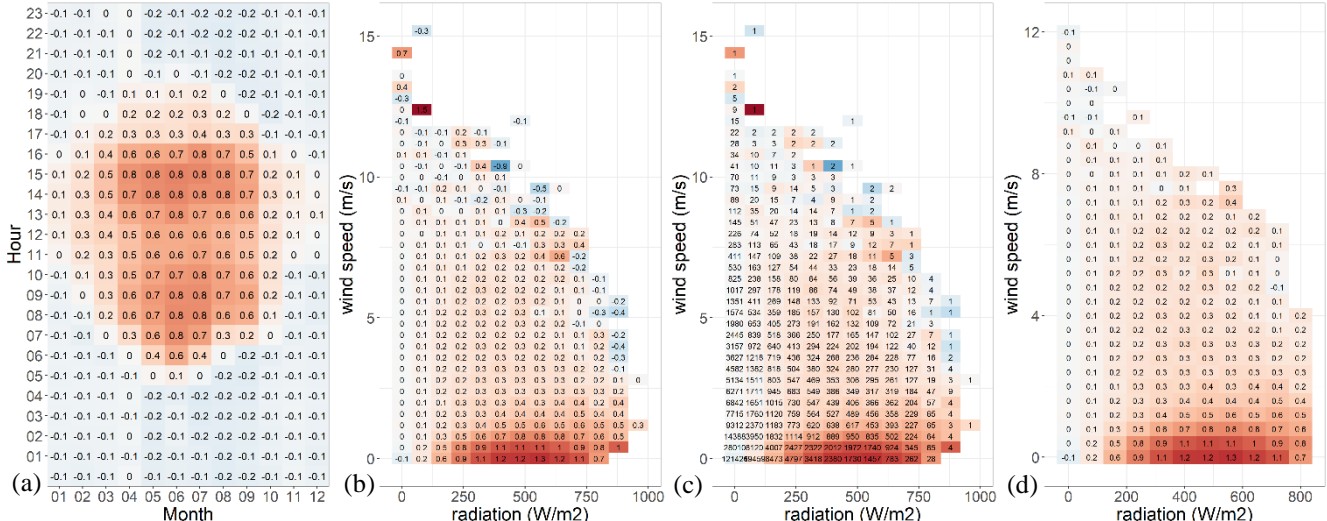

**Figure 4: Temperature bias (°C) as a function of hour of the day and month of the year for all LC-R (a), temperature bias (°C) as a function of radiation and wind speed for all LC-R (b), temperature bias (°C) as a function of radiation and wind speed for all LC-R, the values written in each cell signify the sample size (c), temperature bias (°C) as a function of radiation and wind speed for all LC-R, cells with sample size lower than 10 are not shown in the graph (d). Note that the ranges on the y axis differ for the different subplots. Background colours, ranging from blue (-1.7°C) to red (1.7°C), represent the temperature bias. The temperature bias was calculated for all measurements between the installation data of each LC-R and December 2021.**

### 3.3.2 Building a predictor for the intra station temperature bias

A correlation matrix between the temperature bias and other meteorological variables, measured by the low-cost weather station, is calculated (Table 7). The values indicate how the temperature bias will change for different meteorological conditions.

**Table 7: Pearson correlation matrix of temperature bias with other meteorological variables measured by the low-cost station.**

| Temperature | Dew point temperature | Humidity | Radiation | Radiation60 | Wind speed |
|---|---|---|---|---|---|
| 0.41 | 0.18 | -0.48 | 0.49 | 0.56 | -0.01 |


The most correlated variable is radiation (0.49) directly followed by humidity (-0.48), temperature (0.41), dew point temperature (0.18) and wind speed (-0.01). As expected, taking the past radiation measurements into account further improves the correlation, reaching a maximum value when considering the last 60 minutes (0.56). An exponential weighting, giving higher importance to the radiation measurements closer to the temperature measurement, was used. This variable is further
denoted as Radiation60 (Rad60).

The variables listed in Table 7 were used to build a predictor for the temperature bias. For this purpose, multiple models were calibrated in which the temperature bias is described as a function of only one or multiple meteorological variables. Below (Table 8 and Figure 5) only the models with the best performance are shown. Figure 5 shows the uncorrected temperature bias
(a) as well as the corrected temperature biases after validation with six different models; a simple linear regression with the past radiation (b), a multiple linear regression with the past radiation and windspeed (c), with the past radiation and humidity (d), with the past radiation, windspeed and humidity (e), a random forest model (f) and a boosted regression trees model both including temperature, dew point temperature, humidity, radiation, radiation60, windspeed, altitude, month and hour.

**Table 8: The coefficient of determination (R2) and root mean square error (RMSE) of the different models.**

| Model | $R^2$ | RMSE |
|---|---|---|
| Radiation60 | 0.321 | 0.450 |
| Radiation60 & windspeed | 0.327 | 0.448 |
| Radiation60 & humidity | 0.336 | 0.445 |
| Radiation60 & windspeed & humidity | 0.342 | 0.443 |
| Random Forest | 0.741 | 0.279 |
| Boosted regression trees | 0.658 | 0.319 |

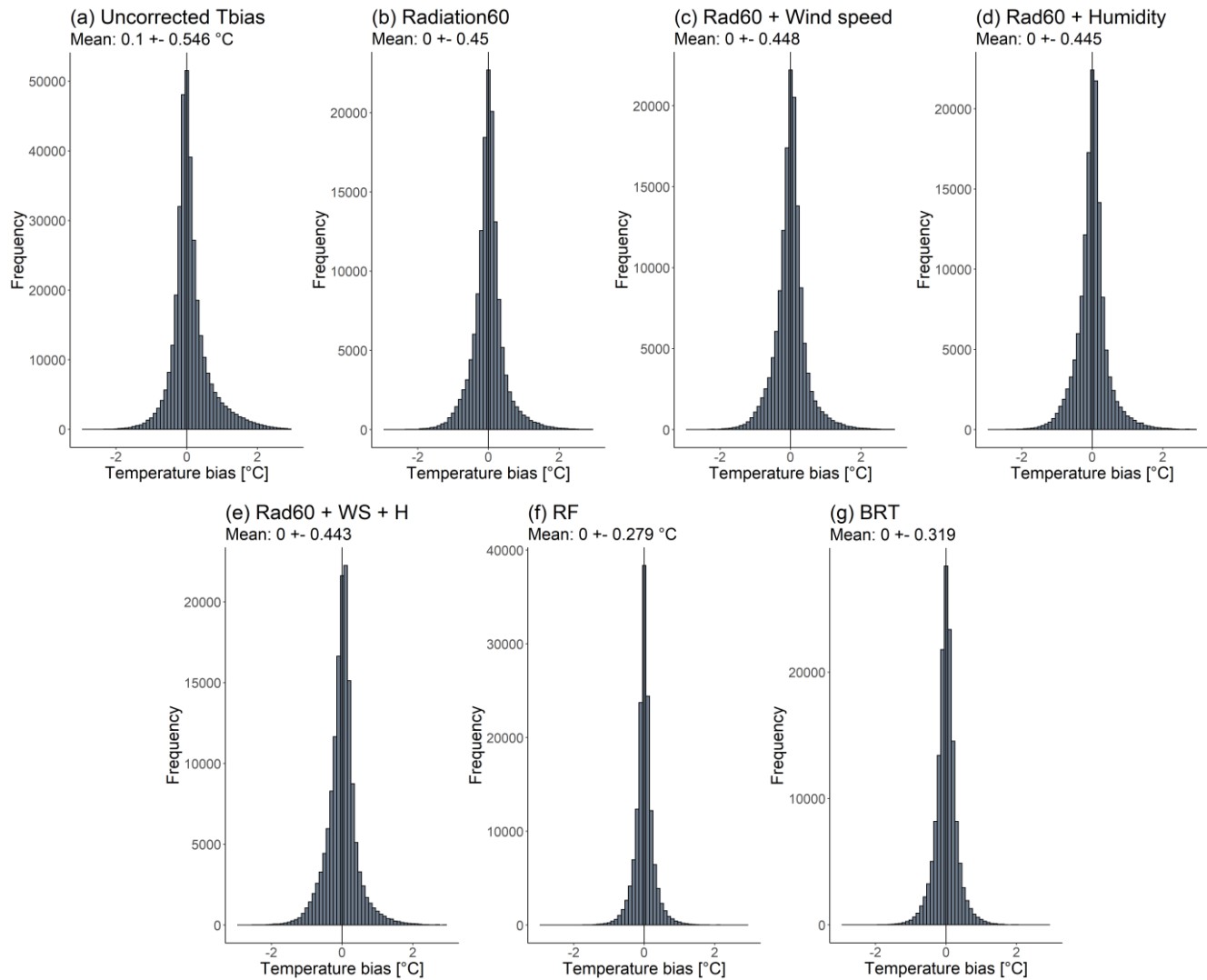

**Figure 5: The uncorrected temperature bias (a) and the corrected temperature bias after validation with a simple linear regression with the past radiations (b), a multiple linear regression with the past radiations and windspeed (c), a multiple linear regression with the past radiations and humidity (d), a multiple linear regression with the past radiations, humidity and windspeed (e), a random forest model (f) and boosted regression trees model including temperature, dew point temperature, humidity, radiation, radiation60, windspeed, altitude, month and hour (g). Mean biases and their standard deviations are given above the graphs. Note that the ranges**
**on the y axis differ for the different subplots.**

A simple linear regression based on the past radiation is already sufficient to suppress the mean temperature bias. Adding additional variables to the model, such as wind speed, humidity or both is statistically significant but only further decreases the RMSE by 0.003 °C to 0.008 °C. The RF and BRT models result in an RMSE of 0.279 and 0.319 and $R^2$ of 0.741 and 0.658

respectively, indicating a better precision and more robust models.

The random forest prediction of temperature bias showed the best results. By splitting up the results for day (radiation > 0 W/m$^2$) and night (radiation = 0 W/m$^2$) (Figure 6), a smaller standard deviation of the bias during night time (0.25) compared to daytime (0.31) is obtained. This differentiation between night and day is only for illustrative purposes, only one RF was built for both day and night. The statistical details of the random forest model are further summarized in Table 9.

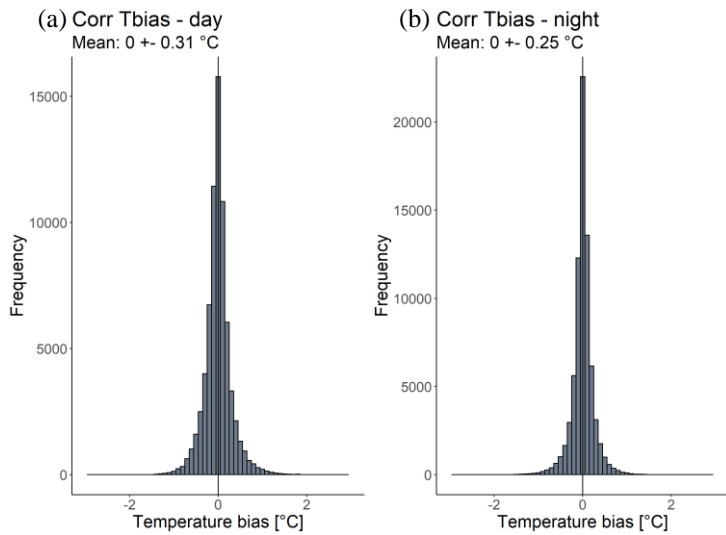

**Figure 6: The corrected Tbias after validation with the random forest model for daytime (a) and for night time (b). Mean biases and their standard deviations are given above the graphs. Note that the ranges on the y axis differ for the different subplots.**

**Table 9: Statistical details of the random forest temperature bias prediction model.**

| | |
|---|---|
| Formula | Tbias ~ QC L2 Temperature + Humidity + Dew point temperature + Radiation + Radiation60 + Windspeed + Altitude + Month + Hour |
| Number of trees | 500 |
| Number of variables tried at each split | 3 |
| Mean of squared residuals | 0.091 |
| % variance explained | 69.5 |

## 4 Evaluation of the quality control and correction method

To evaluate the quality of the developed QC method, it is first applied to the four low cost WH2600 stations (LC-R) (Table 3) installed next to the official measuring equipment in Uccle, Diepenbeek and Humain (AWS). Comparing this LC-R dataset with AWS dataset, allows us to investigate the improvement or deterioration of the data quality after each QC level.

### 4.1 Quality control level 1 – Outlier detection

For QC L1.1 the range outliers are detected by comparing the temperature of each LC-R station with the climatic thresholds set by its nearby official AWS station ($T_{max/min\_AWS} \pm dev\_reference$). As can be seen in Table 6, the allowed deviation from the climatic thresholds is smaller for the LC-R stations compared to the LC-X stations in the study area. This is due to the fact that the LC-R stations are installed next to the official AWS stations, no environmental factors should thus be taken into account. The temporal outliers are detected in QC L1.2 by comparing the rate of change between consecutive observations with the thresholds defined in Table 6. In QC L1.3 spatial outliers are detected using the Z-score. We should however stress that this analysis here is not ideal since every reference station only has 1 neighbour, the official AWS station. Only LC-R01 and LC-R02 located in Uccle have two neighbours during a period of 4 months, when both stations were active simultaneously.

The results show no spatial outliers for the LC-R stations. Some observations are however highlighted as range or temporal outliers. Table 10 summarises the number and percentage of observations flagged as 1. These observations are set to NA resulting in the temperature dataset with quality level 1. The temperature profiles of the LC-R stations versus the official AWS temperature (Figure 7; full coloured versus dashed grey line) highlight observations defined as range or temporal outliers as circles or squares respectively. Scatterplots in which the temperature of the LC-R stations is compared to the temperature of the AWS stations (Figure 8) use the same layout. The temperature difference between the LC-R stations and AWS stations ($\Delta T = T_{LC-R} - T_{AWS}$) is calculated as an effective quality measure.

Table 10: Number and percentage of observations flagged as outliers during QC level 1.

| QC level | # flagged observations | % flagged observations |
|---|---|---|
| QC L1.1 Range test | 21 | 0.006 |
| QC L1.2 Temporal outliers | 180 | 0.048 |
| QC L1.3 Spatial outliers | 0 | 0.000 |
| Total | 201 | 0.054 |

The results show only a few range outlier and even no spatial outliers for the LC-R reference stations. The procedure does however highlight 180 observations as temporal outliers (Table 10). These observations were highlighted during the persistence test, 180 observations change less than 0.05 °C within 2 hours. With only 0.054% of the observations flagged as

outliers we can conclude that the LC-R reference dataset does not contain a lot of outliers. Because of their importance in this

QC method, especially in QC L3, these reference stations are indeed closely monitored hereby preventing and minimising the occurrence of outliers. Since only 0.054% of the data was set to NA, no difference in the ΔT statistics occurred. The histograms in Figure 9 thus represent the data with both QC level 0 and QC level 1. The mean temperature difference and standard deviation for all reference stations equals 0.15 ± 0.56 °C.

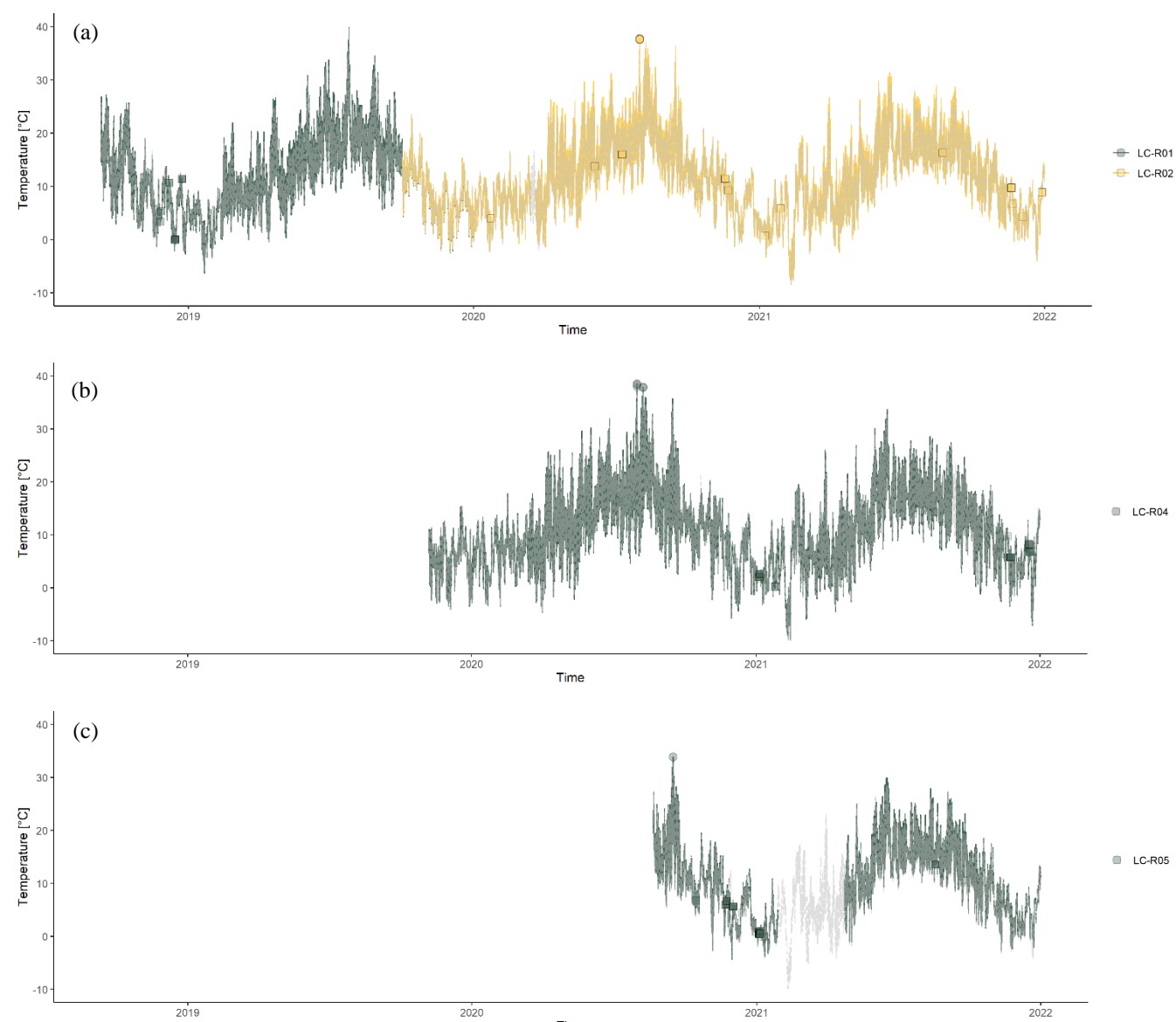


**Figure 7: Temperature profile of LC-R stations in Uccle (LC-R01, LC-R02) (a), Diepenbeek (LC-R04) (b) and Humain (LC-R05) (c). The grey dashed line represents the official AWS temperature of a specific location. Observations defined as range outliers are symbolized by a circle, temporal outliers as a square. The temperature profiles include all measurements between the installation**

**data of each LC-R and December 2021.**

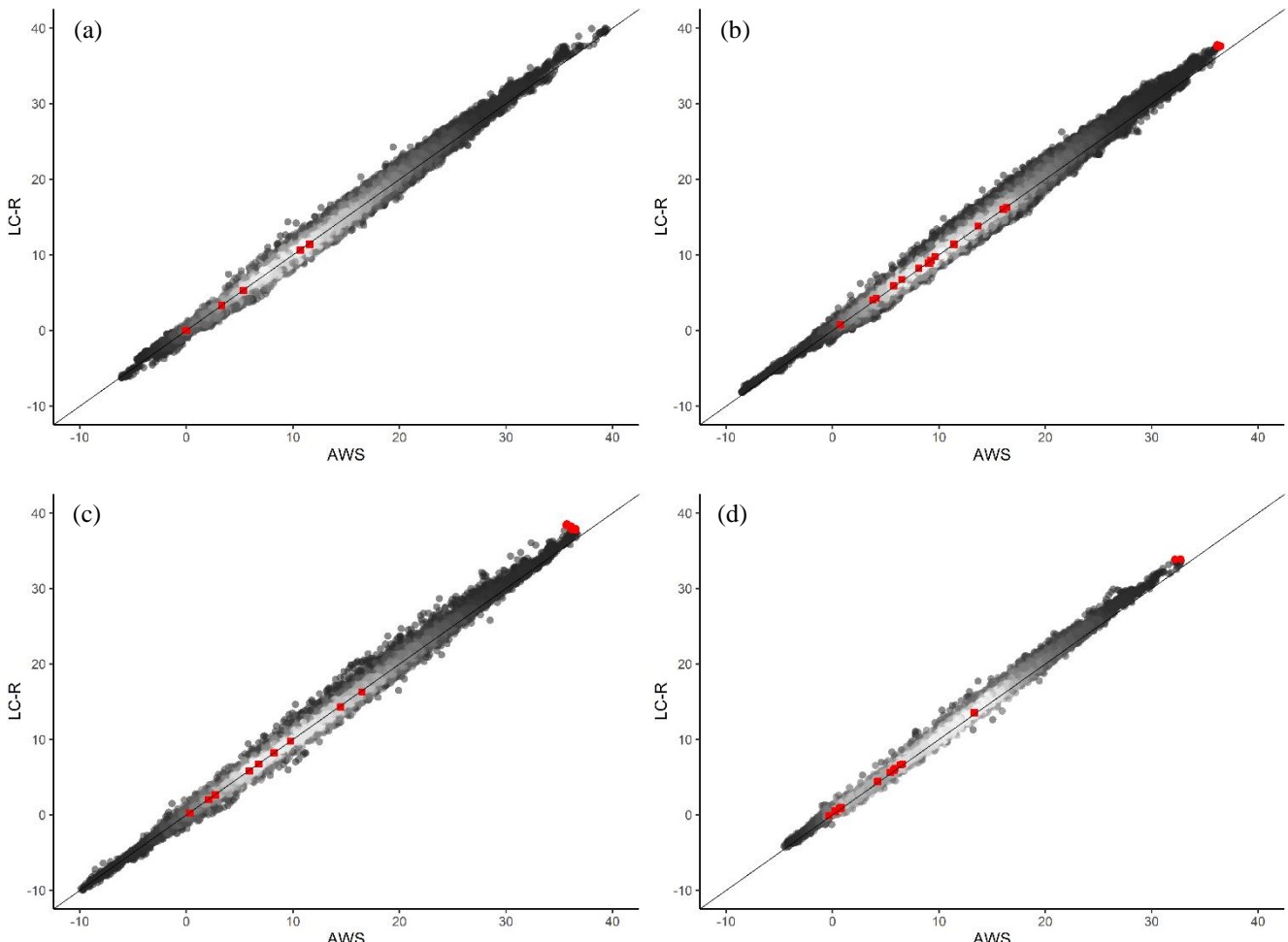

**Figure 8: Scatterplots of LC-R versus AWS temperature for each reference station LC-R01 (a), LC-R02 (b), LC-R04 (c), LC-R05 (d) at QC level 0. Observations defined as range outliers are symbolized by a red circle, temporal outliers by a red square. The identity line is shown in black. The colour scale indicates the density of observations, white indicating the highest, black the lowest. The scatterplots include all measurements between the installation data of each LC-R and December 2021.**

Reference station LC-R01 only has a small positive mean ΔT, while the mean ΔT of LC-R02 and especially LC-R05 is remarkably higher. The mean ΔT of LC-R04 equals zero. The standard deviations of LC-R04 and L-R05 are noticeably smaller than those of LC-R01 and LC-R02 (Figure 9). As expected, the temperature difference between the LC-R and AWS stations is not constant and is correlated with other variables. A higher difference is obtained during the summer months (Figure 10) under low cloud and low windspeed conditions (Figure 11).

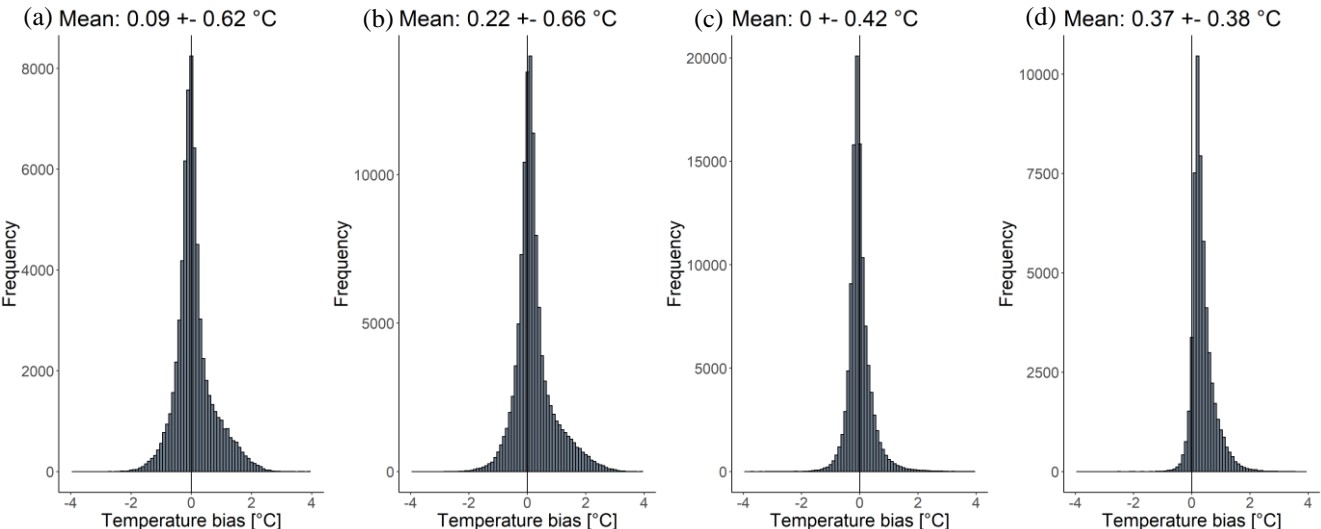

**Figure 9: Histograms of temperature difference ($\Delta T = T_{LC-R} - T_{AWS}$) for each reference station LC-R01 (a), LC-R02 (b), LC-R04 (c), LC-R05 (d) at QC level 0 and QC level 1. Mean differences and their standard deviations are given above the graphs. Note that the ranges on the y axis differ for the different subplots. The temperature difference was calculated for all measurements between the installation data of each LC-R and December 2021.**

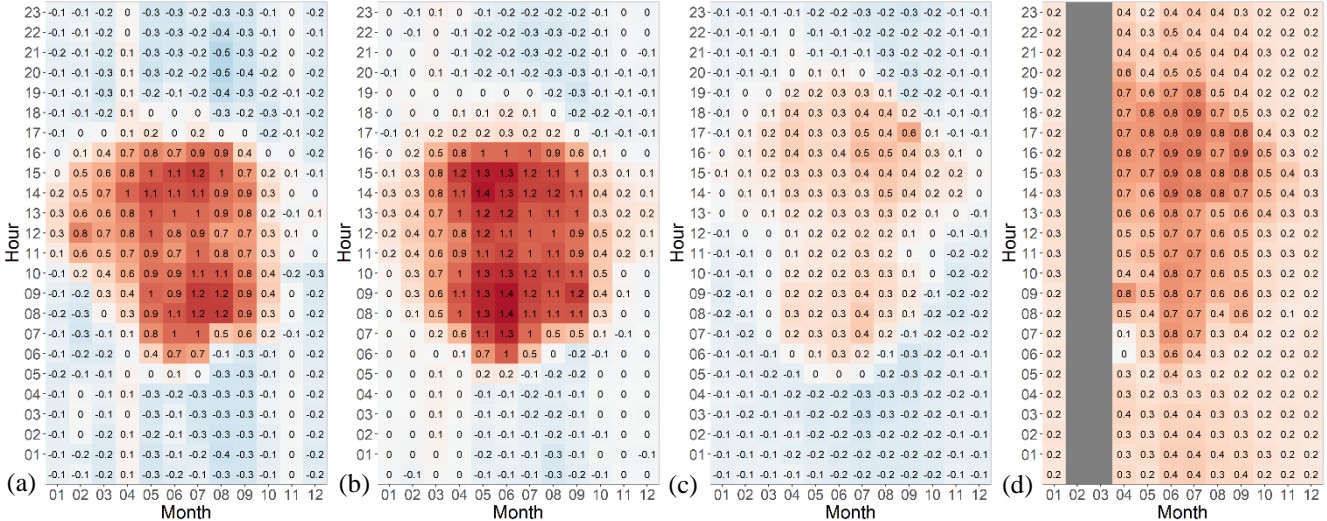

**Figure 10: Temperature difference ($\Delta T = T_{LC-R} - T_{AWS}$) as a function of hour of the day and month of the year for each reference station LC-R01 (a), LC-R02 (b), LC-R04 (c), LC-R05 (d) at QC level 1. Background colours, ranging from blue (-1.7°C) to red (1.7°C), represent the $\Delta T$. The temperature difference was calculated for all measurements between the installation data of each LC-R and December 2021.**

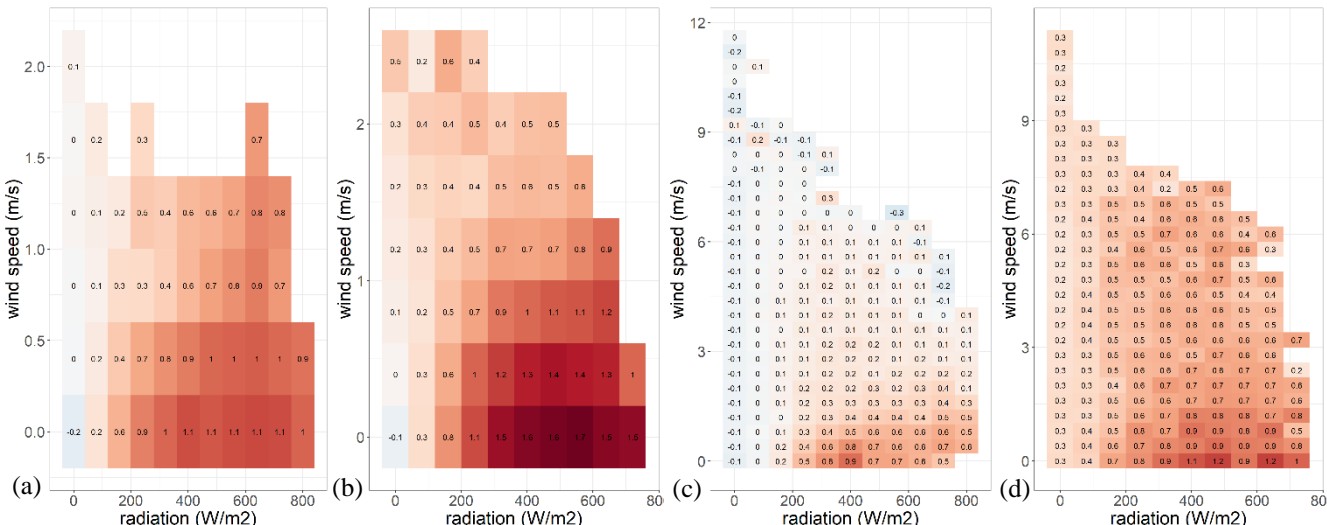

Figure 11: Temperature difference ($\Delta T = T_{LC-R} - T_{AWS}$) as a function of radiation and wind speed for each reference station LC-R01 (a), LC-R02 (b), LC-R04 (c), LC-R05 (d) at QC level 1. Cells with sample size lower than 10 are not shown in the graph. Note that the ranges on the y axis differ for the different subplots. Background colours, ranging from blue (-1.7°C) to red (1.7°C), represent the $\Delta T$. The temperature difference was calculated for all measurements between the installation data of each LC-R and December 2021.

## 4.2 Quality control level 2 – Inter station bias correction

During QC level 2 temperatures are corrected for the fixed offset between stations or inter station bias, due to intrinsic sensor differences at the level of the electronics. The proposed methodology of searching for episodes with a very uniform temperature field over the study area (see Section 3), cannot be applied here, due to the large distance between the three locations with LC-R stations.

Here we selected episodes for which we expect a similar temperature between the LC-R and AWS stations. This occurs again under breezy cloudy conditions with no rainfall (Figure 11). For each reference station, all 10 minute observations are resampled to 2 hour observations, hereby calculating the mean temperature, windspeed, radiation and rainfall. Next, suitable episodes are found by selecting episodes where the average rainfall intensity equals 0 mm/h and the average radiation lies below 100 W/m2. The selected episodes are ordered on average windspeed and limited to the top 10 results. The mean LC-R and AWS temperature is calculated for each episode, next an offset between both is calculated. Finally the median offset across all episodes is considered as the true offset for each station. These offsets are subtracted from the QC L1 temperature data in order to obtain a corrected temperature; the QC level 2 dataset.

Reference stations LC-R01 and LC-R04 have a small negative offset, equal to -0.029 °C and -0.072 °C respectively. Stations LC-R02 and LC-R05 have positive and notably larger offsets equal to 0.113 °C and 0.243 °C (Figure 12).

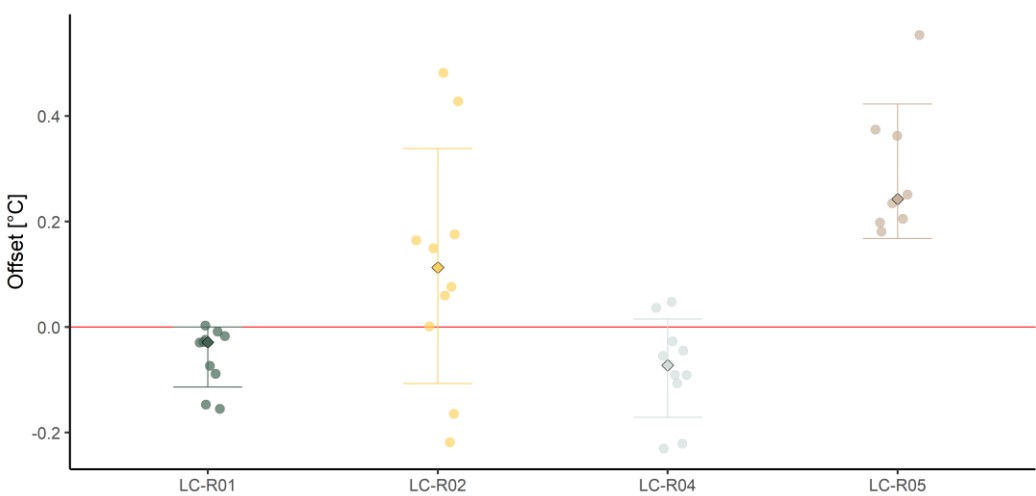

**Figure 12: Offsets during the selected episodes (dots), the median offset (diamond) and its error bar (mean ± standard deviation) for each reference station (LC-R). For reference the zero-line is plotted in red.**

To check the quality improvement of QC level 2, the temperature difference ($\Delta T = T_{LC-R} - T_{AWS}$) between the LC-R and AWS stations is again calculated for every station (Figure 13). Reference stations LC-R01 and LC-R04 show a small increase in
their mean $\Delta T$, the mean $\Delta T$ of LC-R02 and LC-R05 however decreases. As a result, the $\Delta T$ of all stations becomes more equal. Since QC level 2 only added a fixed temperature offset, the standard deviation of all $\Delta T$ remain the same. The inter station bias correction further highlights the seasonal and daily pattern of the $\Delta T$, especially for reference station LC-R05 (Figure 14).

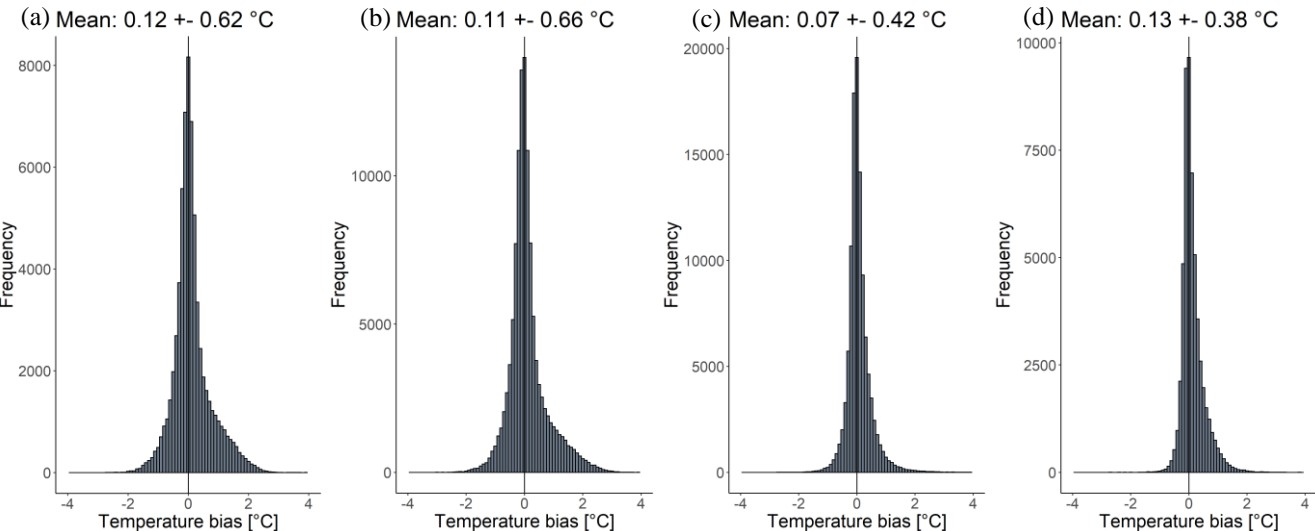

 Figure 13: Same as Figure 9 for QC level 2.

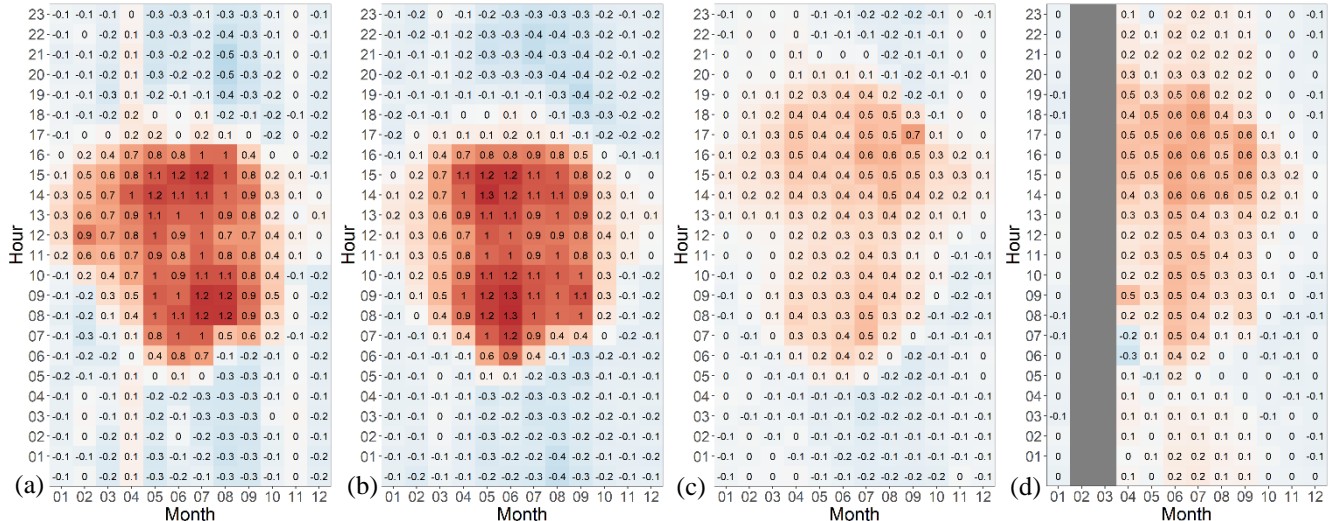

Figure 14: Same as Figure 10 for QC level 2.

## 4.3 Quality control level 3 – Intra station bias correction

From Section 3, we recall that the random forest prediction of temperature bias showed the best results After applying this prediction model on the reference dataset (LC-R), a level 3 corrected temperature is obtained for each LC-R station. These results could be biased since the RF model is trained on 60 % of the LC-R dataset, and then applied on the complete LC-R dataset. To account for this we did apply the prediction model to both the complete dataset and the test data set (which was not used for training the model). The outcome for each LC-R is listed in Appendix D. It can be noted that mean T difference

remains equal for both datasets, the standard deviation does slightly increase with 0.5 to 0.7 °C when using the test dataset. The results below are based on the test dataset only.

To evaluate the quality improvement of QC level 3, the temperature difference ($\Delta T = T_{LC-R} - T_{AWS}$) between the LC-R and AWS stations is again calculated for every station (Figure 15). Histograms of the temperature show a mean $\Delta T$ of almost 0°C

for each reference station, the standard deviation clearly decreased compared to QC level 2 (Figure 13). When the $\Delta T$ is plotted in function of each month and hour of the day, one can notice that the diurnal and seasonal pattern is completely corrected for (Figure 16). Also effects of wind speed and radiation are effectively eliminated (Figure 17). The mean temperature difference and standard deviation for all LC-R stations equals to $0.00 \pm 0.28$ °C.

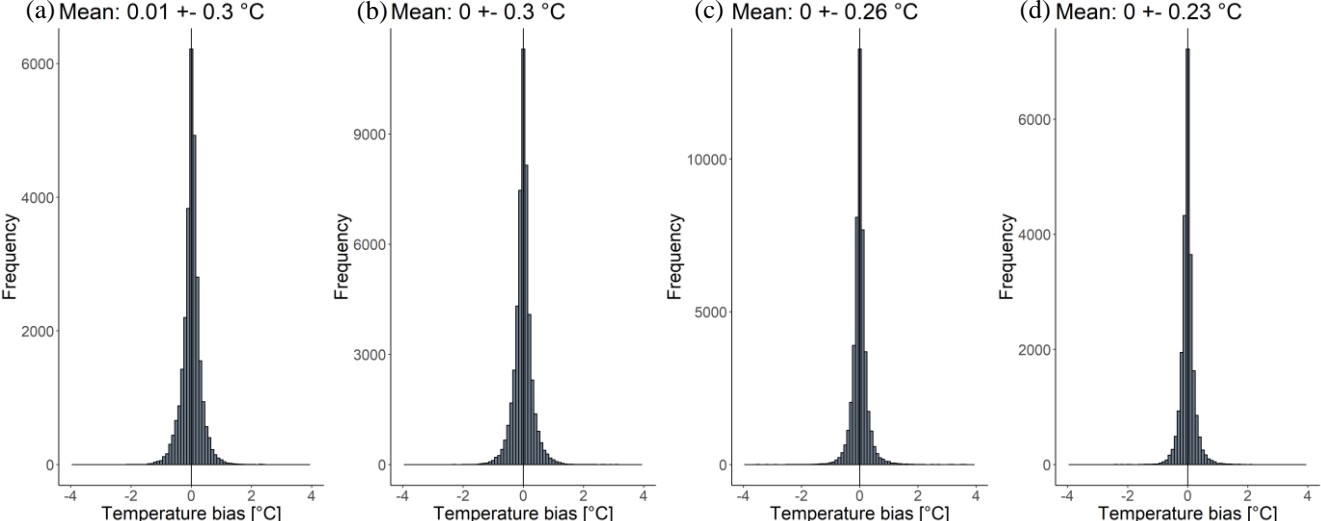


**Figure 15: Same as Figure 9 for QC level 3.**

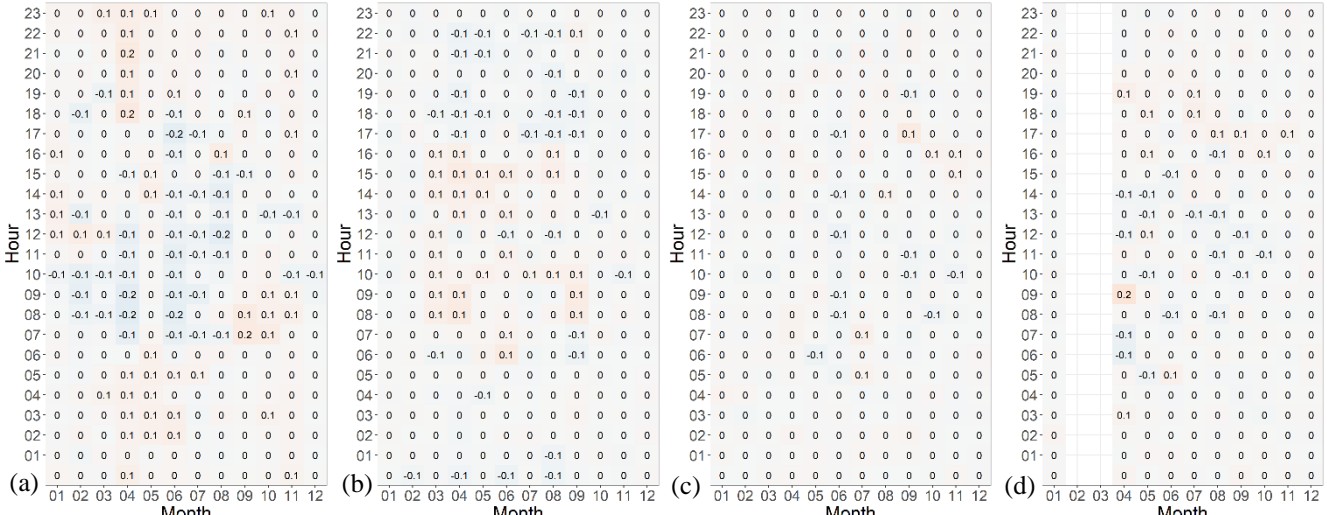

Figure 16: Same as Figure 10 for QC level 3.

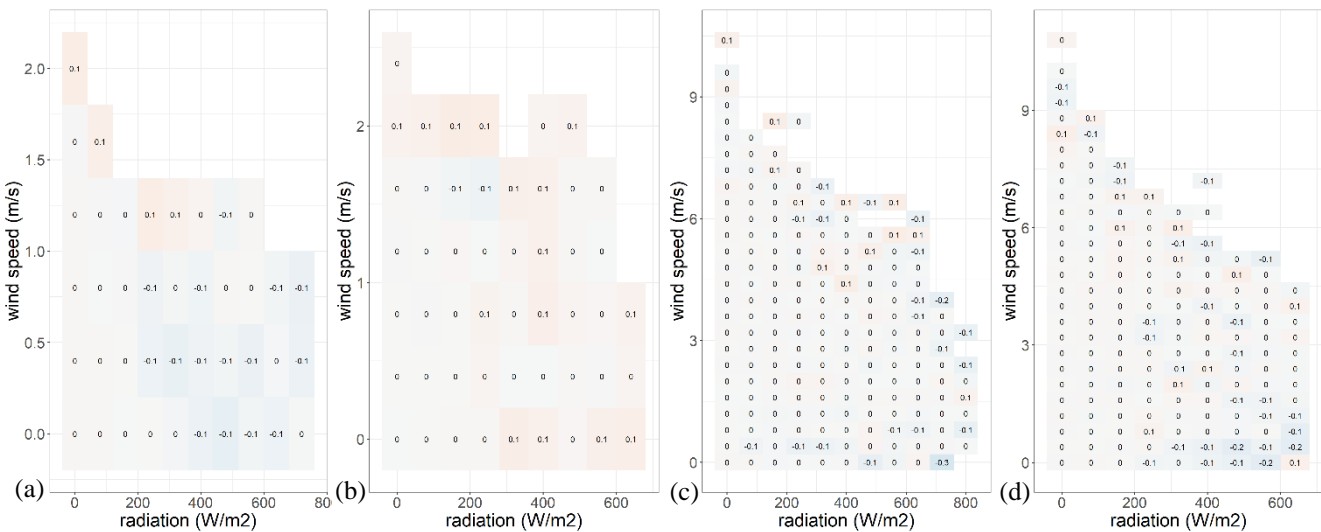

Figure 17: Same as Figure 11 for QC level 3.

## 5 Application of the QC method to the stations in the study area

In this section the newly developed QC method is applied to the low-cost stations of the Leuven.cool network (LC-X). The Leuven.cool dataset currently ranges from July 2019 (2019Q3) until December 2021 (2021Q4). The QC method is performed
4 times a year, each time for a period of three months.

### 5.1 Quality control level 1 – Outlier detection

During QC level 1 range, temporal and spatial outliers are removed using climatological thresholds from Uccle, neighbouring observations and neighbouring stations respectively. If no official weather station would be available, thresholds can be based on existing climate classification maps. Table 11 summarizes the number of observations flagged as outliers in each step. For
each year, only between 0.5 % and 1 % of the data is defined as outliers and thus eliminated, indicating that the raw data quality is rather good compared to other citizens science networks.

Table 11: Number and percentage of observations flagged as outliers during QC level 1 for 2019 Q3-4, 2020 Q1-2-3-4 and 2021 Q1-2-3-4 for all LC-X measurements.

| QC level | # 2019 | % 2019 | # 2020 | % 2020 | # 2021 | % 2021 |
|---|---|---|---|---|---|---|
| QC L1.1 Range test | 0 | 0.000 | 9 | 0.000 | 26 | 0.000 |
| QC L1.2 Temporal outliers: step | 22 | 0.001 | 276 | 0.006 | 169 | 0.003 |
| QC L1.2 Temporal outliers: persistence | 796 | 0.039 | 1216 | 0.025 | 4796 | 0.092 |
| QC L1.3 Spatial outliers | 11769 | 0.581 | 31846 | 0.658 | 26186 | 0.503 |
| Total | 12587 | 0.621 | 33317 | 0.689 | 31177 | 0.599 |


The simple spatial outliers test performed by Chapman et al. (2017) yielded comparable results: 1.5 % of the data was omitted in this study. Other studies reported a much higher fraction of eliminated data. Meier et al. (2017) only kept 47 % of the raw data after conducting their 4 step QC analysis. Napoly et al (2018) and Feichtinger et al. (2020) kept 58 % and 55 % of the data respectively. CrowdQC+ as a further development of CrowdQC, results in an even lower data availability, only 30 % of
the raw data remains after the QC (Fenner et al., 2021). These high numbers of omitted data can however be explained by (1) a great number of CWS installed indoors (not applicable in our setup), thereby lacking the typical diurnal temperature patterns, and (2) radiative errors due to solar radiation exposure of poorly designed devices, resulting in very high temperature observations (Napoly et al., 2018; Fenner et al., 2021). In the QC method presented in this paper, calibration and radiative errors are, however, rather than omitted, corrected for during QC level 2 and 3.

## 5.2 Quality control level 2 – Inter station bias correction

In QC level 2 a fixed offset for each weather station is obtained. These offsets, induced by the intrinsic differences on the level of the sensors' electronics, are subtracted from the station's temperature, thereby accounting for calibration errors. The obtained offsets, median offset and error bar for each station are plotted in Figure 18. The mean offset of all stations equals 0.010 °C. Station LC-102 has the highest offset equal to 0.349 °C, station LC-074 has the lowest offset equal to -0.220 °C. It can be noticed that the error bars of most stations are rather small, which reinforces our confidence of a valid determination of the fixed calibration offset.

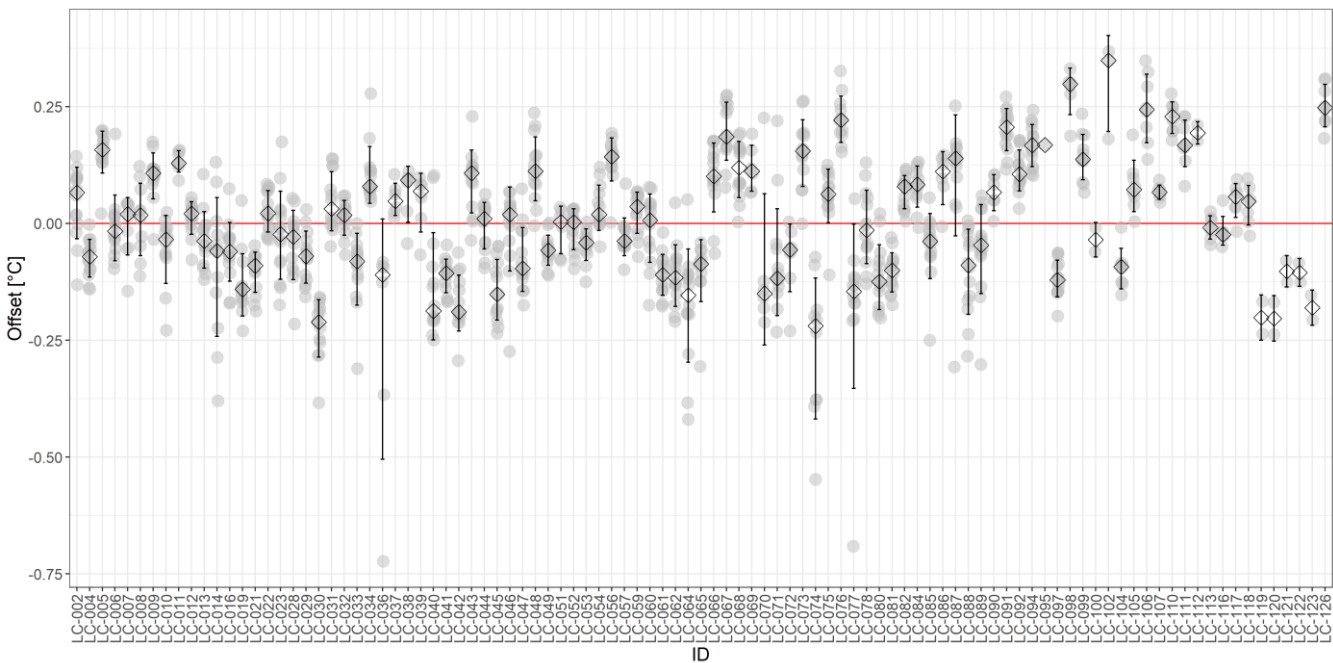

**Figure 18: Offsets during the selected episodes (grey dots), the median offset (black diamond) and its error bar (mean ± standard deviation) for each LC-X station active during at least one episode. For reference the zero-line is plotted in red.**

For stations that were not active during one of the selected timeframes, we were not able to determine their calibration offset (stations LC-003, LC-096, LC-108, LC-109, LC-114, LC-124, LC-125, LC-127 and LC-128). As a consequence, no corrected temperature could be calculated, meaning that those stations are not considered during the following QC level (QC level 3). The search for episodes should be extended with upcoming periods of 6 months in order to resolve this problem.

## 5.3 Quality control level 3 – Intra station bias correction

In QC level 3 the random forest model is applied to each temperature observation of all LC-X stations in order to obtain a site- and time-specific prediction for its temperature bias. As expected, this prediction shows the same pattern as seen for the LC-R stations: generally, we see a positive bias that peaks around midday and is more pronounced during both summer months and low cloud and low windspeed conditions (Figure 19). Note that the actual bias calculation in QC level 3 is performed for

every timestamp and every LC-X station separately, using the other weather variables measured by the station, as input for the
RF model. After subtracting this temperature bias from the observed temperature, corrected temperature for each station is
obtained.

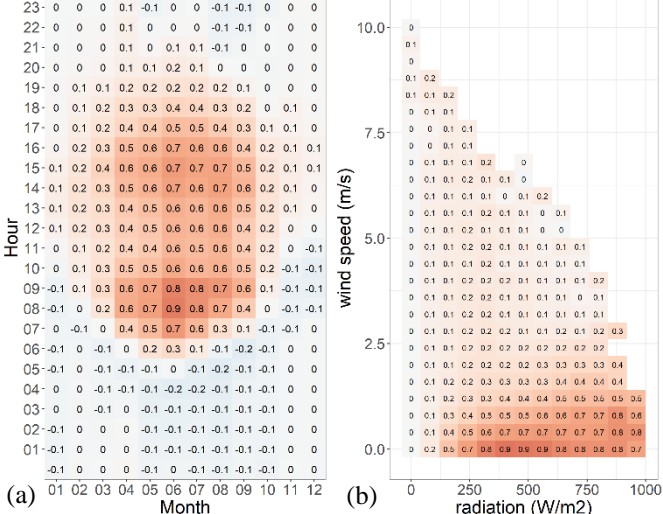

**Figure 19: Prediction of the temperature bias (°C) as a function of hour of the day and month of the year for all LC-X stations (a) and prediction of the temperature bias (°C) as a function of radiation and wind speed for all LC-X stations, cells with sample size**
**lower than 10 are not shown in the graph (b). Background colours, ranging from blue (-1.7°C) to red (1.7°C), represent the temperature bias. The prediction of the temperature bias was calculated for all LC-X measurements July 2019 and December 2021.**

These results are in line with the findings of Jenkins et al. (2014) and Bell et al. (2015):both found a significant positive instrument temperature bias during daytime with strong relation to the incoming solar radiation for multiple types of
crowdsourced weather stations. To our knowledge, previous to our study, only Cornes et al. (2020) has used the findings of Bell et al. (2015) to actually correct crowdsourced air-temperature data. Radiation from satellite imagery and background temperature data from official stations was used to parameterize the short-wave radiation bias, as a consequence no correction was performed for night time. The data correction has reduced the error from ± 0.2 - 0.8 °C to ± 0.2 - 0.4 °C. These results are comparable with our results, although here slightly smaller errors of only 0.23 - 0.30 °C are obtained (Figure 15). Cornes et
al. (2020) do suggest to incorporate wind speed as an additional co-variate in order to incorporate the effect of passive ventilation. The random forest model described in this study does include additional co-variates, including wind speed, but most importantly only needs data from the weather station itself. No satellite imagery or official stations are needed once the random forest model is built. Cornes et al. (2020) further highlights the need for station specific quality controls in order to remove the confounding effect of different instrument types. With the use of a unique station type, we aimed at minimizing
such effects in our dataset. QC level 2 (Sect. 5.2) showed that these effects were indeed limited.

### 5.4 Overall impact of the QC method on the dataset

To assess the impact of the different QC stages on the global dataset, several violin boxplots were created both on monthly and yearly base. Figure 20 illustrates the monthly violin plots for 2019, 2020 and 2021 at each quality control level. Table 12 summarises the mean monthly temperature and its standard deviation for each quality control level.


The violin plots, and accompanying mean temperature and standard deviation, do not change much over the different QC levels. Since QC level 1 removes outliers from the dataset we would expect a lower standard deviation for QC level 1 compared to QC level 0. Figure 20 does indicate the removal of some outliers, but Table 12 does not confirm the expected decrease in standard deviation. This can be explained by the low percentage of observations defined as outliers, per year only 0.5 to 1 %

of the data was defined as outlier. Due to the strict installation protocol, most errors were already eliminated upfront. If errors do occur as a result of station malfunctioning, they are quickly resolved since the dataflow and station siting are continuously controlled.

During QC level 2 each station is corrected for its inter station temperature bias. Since both positive and negative biases,

ranging from 0.349 °C to -0.220 °C, are possible no clear change in the mean temperature is expected between QC level 1 and QC level 2. Because this QC level corrects each station with a fixed offset the standard deviation should stay the same. Both of these assumptions are confirmed by Figure 20 and Table 12.

During the third QC level we do see a clear change in mean temperature and standard deviation. For the summer months a

reduction in both the mean temperature and standard deviation up to respectively -0.40 °C and -0.36 °C is noted. The change in standard deviation shows a monthly pattern with a higher reduction during the summer months and almost no change during winter. The change in mean temperature is not as consistent and seems dependent on the observed temperatures. A higher reduction in mean temperature is noted for the hot summers of 2019 and 2020 compared to the rather cold summer of 2021. These results can easily be explained by the daily and seasonally patterns of the predicted temperature bias (Figure 19).


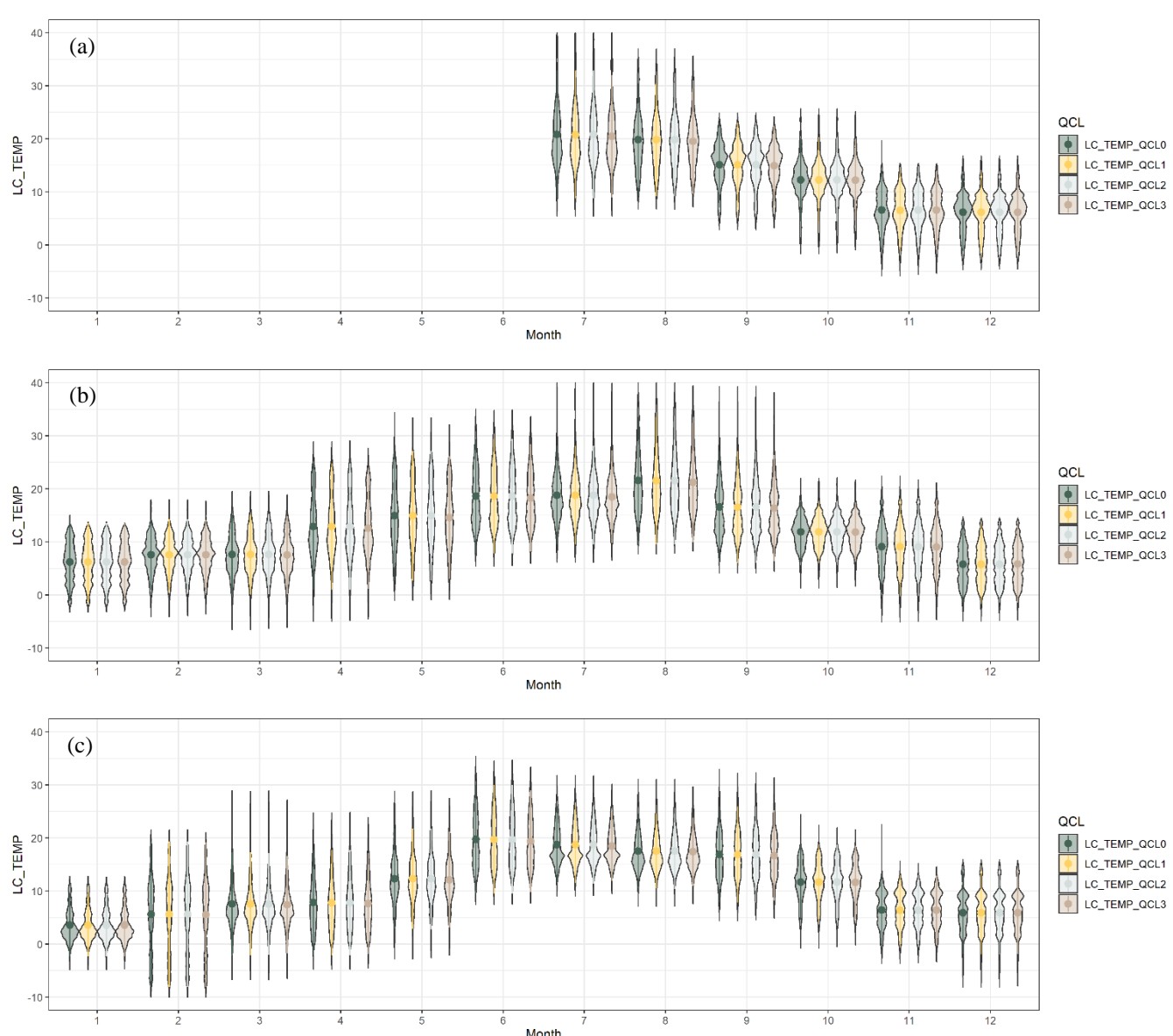

**Figure 20: Monthly violin plots of temperature data (°C) of all LC-X stations at quality control level 0 (raw data), level 1 (outliers removed), level 2 (inter station bias correction) and level 3 (intra station bias correction) for 2019 Q3-4 (a), 2020 Q1-2-3-4 (b) and 2021 Q1-2-3-4 (c).**

**Table 12: The mean monthly temperature (°C) and its standard deviation of all LC-X stations at quality control level 0 (raw data), level 1 (outliers removed), level 2 (inter station bias correction) and level 3 (intra station bias correction).**

| QC level 0 | | QC level 1 | | QC level 2 | | QC level 3 | | ΔL0-L3 | |
|---|---|---|---|---|---|---|---|---|---|
| Mean T | Std T | Mean T | Std T | Mean T | Std T | Mean T | Std T | Mean T | Std T |

| 2019 | | | | | | | | | | |
|---|---|---|---|---|---|---|---|---|---|---|
| 7 | 20.95 | 6.21 | 20.93 | 6.23 | 20.96 | 6.23 | 20.54 | 5.89 | -0.40 | -0.32 |
| 8 | 19.82 | 5.32 | 19.81 | 5.33 | 19.84 | 5.34 | 19.53 | 4.96 | -0.29 | -0.36 |
| 9 | 15.08 | 3.99 | 15.08 | 3.99 | 15.08 | 3.99 | 14.91 | 3.76 | -0.17 | -0.23 |
| 10 | 12.26 | 3.93 | 12.25 | 3.93 | 12.26 | 3.93 | 12.22 | 3.83 | -0.04 | -0.10 |
| 11 | 6.54 | 3.70 | 6.53 | 3.70 | 6.54 | 3.70 | 6.54 | 3.68 | 0.00 | -0.03 |
| 12 | 6.12 | 3.65 | 6.12 | 3.65 | 6.12 | 3.65 | 6.15 | 3.65 | 0.03 | 0.00 |
| 2020 | | | | | | | | | | |
| 1 | 6.21 | 3.68 | 6.20 | 3.68 | 6.21 | 3.68 | 6.21 | 3.66 | 0.00 | -0.02 |
| 2 | 7.57 | 3.33 | 7.57 | 3.33 | 7.57 | 3.33 | 7.56 | 3.29 | -0.02 | -0.05 |
| 3 | 7.62 | 3.76 | 7.62 | 3.77 | 7.62 | 3.76 | 7.49 | 3.69 | -0.13 | -0.07 |
| 4 | 12.89 | 5.97 | 12.88 | 5.97 | 12.88 | 5.97 | 12.55 | 5.71 | -0.33 | -0.26 |
| 5 | 14.95 | 6.10 | 14.94 | 6.11 | 14.93 | 6.10 | 14.58 | 5.79 | -0.37 | -0.32 |
| 6 | 18.60 | 5.38 | 18.60 | 5.38 | 18.61 | 5.39 | 18.26 | 5.05 | -0.35 | -0.33 |
| 7 | 18.79 | 4.66 | 18.79 | 4.67 | 18.80 | 4.67 | 18.47 | 4.36 | -0.33 | -0.30 |
| 8 | 21.55 | 6.00 | 21.54 | 6.00 | 21.52 | 6.00 | 21.18 | 5.63 | -0.37 | -0.37 |
| 9 | 16.54 | 5.28 | 16.53 | 5.28 | 16.54 | 5.28 | 16.35 | 5.01 | -0.19 | -0.27 |
| 10 | 11.88 | 2.86 | 11.88 | 2.86 | 11.87 | 2.85 | 11.84 | 2.79 | -0.05 | -0.08 |
| 11 | 9.08 | 4.34 | 9.07 | 4.34 | 9.07 | 4.34 | 9.05 | 4.31 | -0.03 | -0.03 |
| 12 | 5.79 | 3.63 | 5.78 | 3.63 | 5.79 | 3.62 | 5.80 | 3.62 | 0.01 | 0.00 |
| 2021 | | | | | | | | | | |
| 1 | 3.58 | 2.95 | 3.57 | 2.96 | 3.57 | 2.95 | 3.58 | 2.94 | 0.01 | -0.02 |
| 2 | 5.59 | 6.88 | 5.59 | 6.89 | 5.59 | 6.88 | 5.50 | 6.84 | -0.09 | -0.04 |
| 3 | 7.55 | 4.84 | 7.55 | 4.84 | 7.51 | 4.80 | 7.40 | 4.61 | -0.16 | -0.23 |
| 4 | 7.80 | 4.95 | 7.80 | 4.95 | 7.78 | 4.96 | 7.60 | 4.76 | -0.21 | -0.20 |
| 5 | 12.30 | 4.75 | 12.30 | 4.75 | 12.29 | 4.76 | 12.10 | 4.52 | -0.20 | -0.23 |
| 6 | 19.70 | 5.11 | 19.70 | 5.11 | 19.69 | 5.12 | 19.37 | 4.77 | -0.33 | -0.34 |
| 7 | 18.72 | 3.64 | 18.71 | 3.64 | 18.71 | 3.65 | 18.48 | 3.29 | -0.23 | -0.34 |
| 8 | 17.51 | 3.53 | 17.51 | 3.54 | 17.51 | 3.54 | 17.38 | 3.22 | -0.13 | -0.31 |
| 9 | 16.88 | 4.52 | 16.87 | 4.52 | 16.86 | 4.52 | 16.68 | 4.26 | -0.20 | -0.26 |
| 10 | 11.68 | 3.55 | 11.67 | 3.55 | 11.66 | 3.56 | 11.63 | 3.44 | -0.05 | -0.11 |
| 11 | 6.38 | 3.29 | 6.37 | 3.29 | 6.36 | 3.29 | 6.37 | 3.26 | -0.01 | -0.03 |
| 12 | 5.90 | 3.92 | 5.89 | 3.92 | 5.88 | 3.93 | 5.90 | 3.92 | 0.00 | 0.00 |

## 6. Application potential of the quality controlled and corrected Leuven.cool dataset

A validation of the proposed QC method showed that it can reduce the mean temperature difference and standard deviation from 0.15 ± 0.56 °C to 0.00 ± 0.28 °C. The QC method can correct the temperature difference equally across different hours of the day and months of the year (Figure 16) as well and under different radiation and windspeed conditions (Figure 17).

The quality-controlled Leuven.cool dataset enables a detailed comparison with other crowdsourced datasets for which less or even no metadata is available. As such, the Leuven.cool stations can serve as gatekeepers for other crowdsourced observations. In the past this role has been limited to standard weather station network which mostly only have a limited number of observations available (Chapman et al., 2017).

Numerous studies have shown that the UHI effect causes night time temperature differences up 6 to 9 °C during clear nights (Chapman et al., 2017; Venter et al., 2021; Stewart, 2011; Napoly et al., 2018; Feichtinger et al., 2020). These thresholds are much higher than mean bias obtained after correction. The dense quality controlled Leuven.cool dataset thus allows for microscale modelling of urban weather patterns, including the urban heat island (Chapman et al., 2017; de Vos et al., 2020; Napoly et al., 2018; Feichtinger et al., 2020). Since such high-quality datasets contain measurements with both high spatial and temporal resolution, they can easily be used to obtain spatially continuous temperature patterns across a region (e.g. Napoly et al., 2018; Feichtinger et al., 2020). Interpolation methods based on single pair stations or mobile transect methods are much less trustworthy (Napoly et al., 2018). Dense weather station networks can be used to investigate the inter- and intra LCZ variability within a city (Fenner et al., 2017; Verdonck et al., 2018). The dataset can further help investigate the relation between temperature and human and ecosystem health (e.g. Demoury et al., 2022; Troeyer et al., 2020) and their effect on evolutionary processes (e.g. Brans et al., 2022).

The dataset can also help refine existing weather forecast models which are currently mostly based on official rural observations (Sgoff et al., 2022). Nipen et al. (2020) showed that the inclusion of citizen observations improves the accuracy of short-term temperature forecasts in regions where official stations are sparse. Mandement and Caumont (2020) used crowdsourced weather stations to improve the observation and prediction of near convection. After quality control and correction, also the wind (Chen et al., 2021) and precipitation measurements (de Vos et al., 2019) can be useful to improve detection and forecasting. Further, the Leuven.cool dataset could be a useful input in air pollution prediction models (e.g. IFD-model (Lefebvre et al., 2011)).

## 7 Conclusion

This study presents the data from the citizen science weather station network Leuven.cool, which consists of around 100 weather stations in the city of Leuven, Belgium. The crowdsourced weather stations (Fine Offset WH2600) are distributed across Leuven and surroundings, measuring the local climate since July 2019. The dataset is accompanied by a newly developed station specific temperature quality control procedure. The quality control method consists of three levels, removing implausible measurements, while also correcting for inter (in between stations) - and intra (station-specific) station temperature biases. This QC method combines suggestions of previous developed methods but improves them by correcting aberrant temperature observations rather than removing them. As a result, more data can be retained allowing researchers to study the highly heterogeneous urban climate in all its detail. Moreover, the QC method uses information from the crowdsourced data itself and only requires reference data from official stations during its development and evaluation stage. Afterwards the method can be applied independently of the official network that was used in the development phase. Transferring the method to other networks or regions would require the recalibration of the QC parameters. Specifically, for QC L1.1 some indication of climate thresholds is needed. QC L1.3 and QC L2 require a dense weather station network, the QC method is thus less suitable for single or few stations. For QC L3 quite some other data (e.g. radiation, windspeed, …) are needed. The random forest model is however easily adaptable to the parameters that are available.

A validation of the proposed QC method was carried out on four Leuven.cool stations installed next to official equipment, and showed that it is able to reduce the mean temperature difference and standard deviation from $0.15 \pm 0.56$ °C to $0.00 \pm 0.28$ °C. The quality-controlled Leuven.cool dataset enables a detailed comparison with other crowdsourced datasets for which less or even no metadata is available. The dense dataset further allows for microscale modelling of urban weather patterns, such as the urban heat island, and can help identify the relation between temperature and human and ecosystem and their effect on evolutionary processes. Lastly the dataset could be used to refine existing forecast models which are currently mostly based on official rural observations. Knowing that both the frequency and intensity of heat waves will only increase during the upcoming years, dense high-quality datasets such as the Leuven.cool datasets become highly valuable for studying local climate phenomena, planning efficient mitigation and adaptation measures and hence mitigating future risks.

## Data and code availability

All data described in this paper and scripts used to design, evaluate and apply the QC method are stored in RDR, KU Leuven's Research Data Repository and accessible through the following DOI: https://doi.org/10.48804/SSRN3F (Beele et al., 2022). The dataset is accompanied by an extensive README file explaining the content of the dataset. The dataset includes a metadata file (01_Metadata.csv), the actual observations per 3 months/one quarter (LC_YYYYQX.csv) and the R scripts needed to build and apply the QC method (LC-R_QCL1-2-3.Rmd and LC-X_QCL1-2-3.Rmd). The metadata file contains info on the weather station's coordinates, altitude above sea level, installation height, LCZ class, dominant landcover, mean SVF and mean building height in a buffer of 10m around each weather station. Coordinates have rounded to 3 decimals for privacy reasons. The actual observations are aggregated to 10 minutes and include Relative humidity [%], Dew point temperature [°C], Number of 16 second observations in 10 minutes aggregate, Solar radiation [W/m2], Rain intensity [mm/h], Daily rain sum [mm], Wind direction [°], Wind speed [m/s], Date in YYYY-MM-DD, Year in YYYY, Month in MM, Day in DD, Hour in HH, Minute in MM, Weighted radiation during last 60 minutes [W/m2], Temperature at QCL0 [°C], Temperature at QCL1 [°C], Temperature at QCL2 [°C], Temperature at QCL3 [°C].

## Author contributions

All authors contributed to the design of the weather station network. EB and MR practically implemented the design and collected the CWS data, MR assembled AWS data and EB processed all data used. All authors contributed to the design of the QC method, EB did the programming of the QC. EB carried out all analyses and mainly wrote the manuscript. All authors discussed the results and contributed to the writing of the manuscript.

## Competing interests

The authors declare that they have no conflict of interest.

## Acknowledgements

We thank the many people who have contributed to the establishment and maintenance of the Leuven.cool project. In particular we thank all citizens and private companies of/within Leuven who voluntary made their garden/terrain available for our research. We also thank Tim Guily (city of Leuven) and Hanne Wouters (Leuven2030) for their logistic support and Margot Verhulst, Jordan Rodriguez Milis, Remi Chevalier and Jingli Yan (KU Leuven) for their technical support during the realisation of the network. The protocol of this study was approved by the Social and Societal Ethics Committee of the KU Leuven (G-2019 06 1674). The weather stations used in this study were sponsored by both the city of Leuven and the KU Leuven. E.B. holds a SB-doctoral fellowship of the Research Foundation Flanders (FWO, 1SE0621N).

## Appendices

**Appendix A. Specifications of the WH2600 digital weather station**

The technical specifications of the Fine Offset WH2600 weather station are given in Table A1.

**Table A1: Technical specifications of the Fine Offset WH2600 weather station as given by the manufacturer.**

| Outdoor sensor array | |
| --- | --- |
| Transmission distance in open field | 100- m |
| Temperature range | -40 °C – 60 °C |
| Temperature accuracy | +/- 1 °C |
| Temperature resolution | 0.1 °C |
| Relative humidity range | 1 % - 99 % |
| Relative humidity accuracy | +/- 5 % |
| Rain volume range | 0 – 9999 mm |
| Rain volume accuracy | +/- 10 % |
| Rain volume resolution | 0.3 mm (if rain volume < 1000 mm) |
| | 1 mm (if rain volume > 1000 mm) |
| Wind speed range | 0 – 50 m/s |
| Wind speed accuracy | +/- 1 m/s (wind speed < 5 m/s) |
| | +/- 10 % (wind speed > 5 m/s) |
| Light range | 0 – 400 k Lux |
| Light accuracy | +/- 15 % |
| Measuring interval | 16 sec |


## Appendix B. Specifications LCZ map

The LCZ map was created using a supervised random forest classification approach based on fine-scale land use, building height, building density and green ratio data within R. The input data used for the creation of the LCZ map is listed in Table B1. All input datasets were rasterized and cropped to spatial resolution of 100 m.


**Table B1: Input data used for the creation of the LCZ map**

| Input data | Dataset | Source |
| --- | --- | --- |
| Land use | Land use data Flanders, 10 m, 2019 | (Landgebruik - Vlaanderen - toestand 2019, 2022) |
| Building height | 3D GRB, 2015 | (3D GRB, 2022) |
| Building density | 3D GRB, 2015 | (3D GRB, 2022) |
| Green ratio | Green map Flanders, 1 m, 2018 | (Groenkaart Vlaanderen 2018, 2022) |

A grid area of 15 by 15 km was drawn around the city centre of Leuven. Within this grid area training polygons for 12 LCZ types (7 urban LCZs and 5 natural LCZs) were drawn. The delineation of the urban training areas was based on the same input

data layers. The threshold values used during this process are further described in Figure B1. The training areas are plotted in Figure B2.

The training polygons were randomly split in training (0.7) and validation (0.3) data. Subsequently a random forest model was trained and validated using both datasets. A majority filter with a 3x3 matrix (3x3 moving window) was applied using the

*focal* function in R to obtain a more realistic and clustered LCZ map (Demuzere et al., 2020). The LCZ map was projected to ESPG: 31370 – Belge Lambert 72 using the *projectRaster* function and nearest neighbour method in R.

The resulting LCZ map has an overall accuracy of 0.79 and Kappa equal to 0.76. The confusion matrix is presented in Table B2.


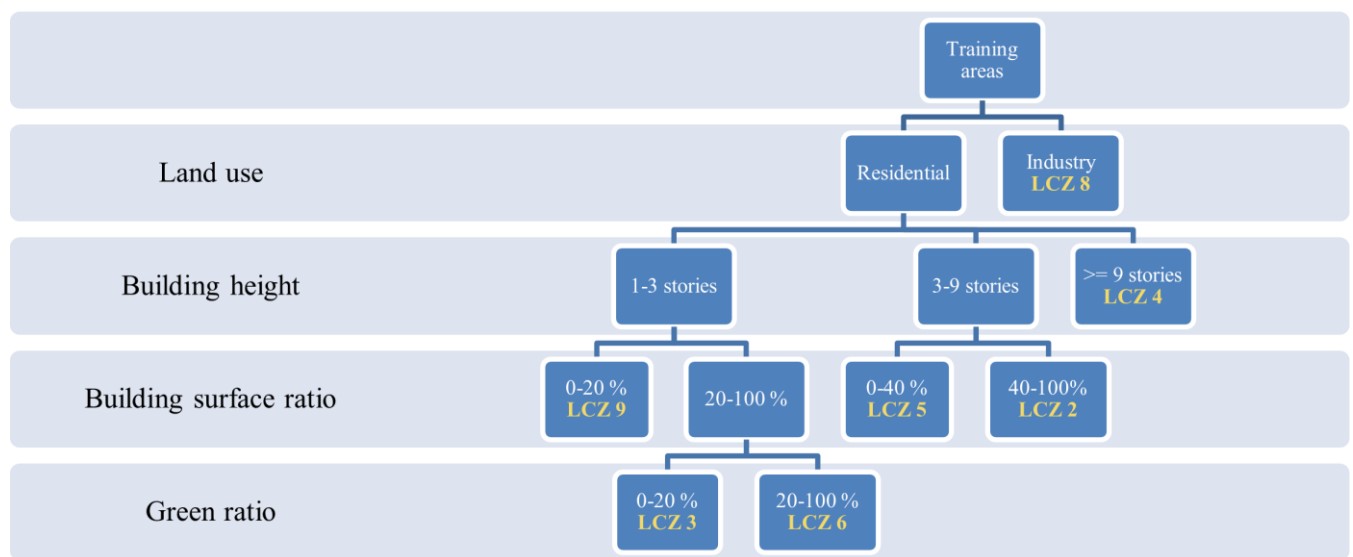

**Figure B1: Decision tree used to delineate training areas for urban LCZ classes.**

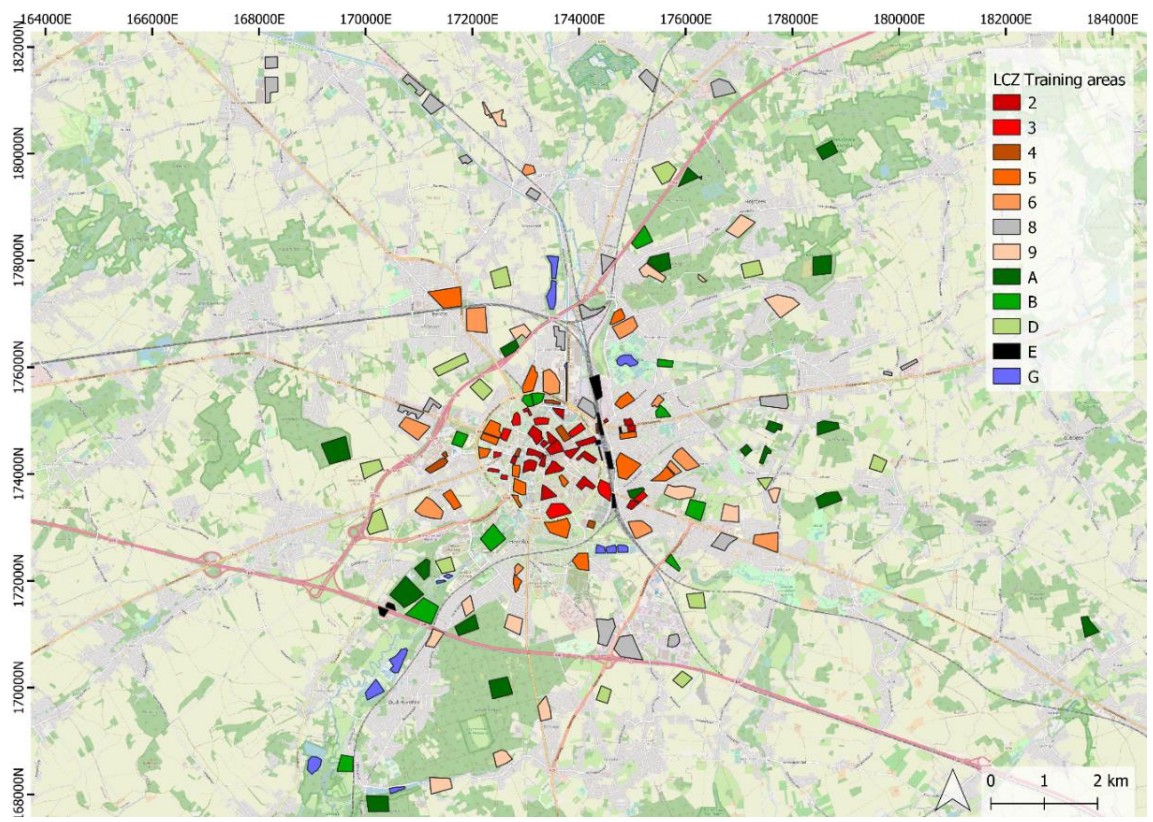

**Figure B2: Training areas used for the creation of the LCZ map. Delineation is based on land use, building height, building density and green ratio data**

**Table B2: Confusion matrix LCZ map. The matric summarises the number of cells wrongly or correctly classified for each LCZ class. The user accuracy (UA) and producer accuracy (PA) for each LCZ class and the overall accuracy are included as well.**

| | LCZ 2 | LCZ 3 | LCZ 4 | LCZ 5 | LCZ 6 | LCZ 8 | LCZ 9 | LCZ 11 | LCZ 12 | LCZ 14 | LCZ 15 | LCZ 17 | Total | UA (%) |
|---|---|---|---|---|---|---|---|---|---|---|---|---|---|---|
| LCZ 2 | **7.0** | 1.0 | 1.0 | 0.0 | 0.0 | 0.0 | 0.0 | 0.0 | 0.0 | 0.0 | 0.0 | 0.0 | 9.0 | 77.8 |
| LCZ 3 | 1.0 | **7.0** | 0.0 | 0.0 | 0.0 | 0.0 | 0.0 | 0.0 | 0.0 | 0.0 | 1.0 | 0.0 | 9.0 | 77.8 |
| LCZ 4 | 0.0 | 0.0 | **4.0** | 0.0 | 0.0 | 0.0 | 0.0 | 0.0 | 0.0 | 0.0 | 0.0 | 0.0 | 4.0 | 100.0 |
| LCZ 5 | 5.0 | 6.0 | 0.0 | **15.0** | 15.0 | 3.0 | 0.0 | 0.0 | 0.0 | 0.0 | 2.0 | 0.0 | 46.0 | 32.6 |
| LCZ 6 | 0.0 | 1.0 | 0.0 | 0.0 | **27.0** | 5.0 | 1.0 | 0.0 | 0.0 | 0.0 | 0.0 | 0.0 | 34.0 | 79.4 |
| LCZ 8 | 0.0 | 0.0 | 0.0 | 0.0 | 0.0 | **49.0** | 0.0 | 0.0 | 0.0 | 0.0 | 0.0 | 0.0 | 49.0 | 100.0 |
| LCZ 9 | 0.0 | 0.0 | 0.0 | 0.0 | 0.0 | 1.0 | **40.0** | 1.0 | 0.0 | 0.0 | 0.0 | 0.0 | 42.0 | 95.2 |
| LCZ 11 | 0.0 | 0.0 | 0.0 | 0.0 | 0.0 | 0.0 | 0.0 | **44.0** | 9.0 | 0.0 | 0.0 | 0.0 | 53.0 | 83.0 |
| LCZ 12 | 0.0 | 0.0 | 0.0 | 0.0 | 0.0 | 0.0 | 2.0 | 2.0 | **9.0** | 0.0 | 0.0 | 5.0 | 18.0 | 50.0 |
| LCZ 14 | 0.0 | 0.0 | 0.0 | 0.0 | 0.0 | 0.0 | 0.0 | 0.0 | 0.0 | **29.0** | 0.0 | 0.0 | 29.0 | 100.0 |
| LCZ 15 | 0.0 | 0.0 | 1.0 | 0.0 | 0.0 | 1.0 | 0.0 | 0.0 | 0.0 | 0.0 | **2.0** | 0.0 | 4.0 | 50.0 |
| LCZ 17 | 0.0 | 0.0 | 0.0 | 0.0 | 0.0 | 0.0 | 0.0 | 0.0 | 0.0 | 0.0 | 0.0 | **9.0** | 9.0 | 100.0 |
| Total | 13.0 | 15.0 | 6.0 | 15.0 | 42.0 | 59.0 | 43.0 | 47.0 | 18.0 | 29.0 | 5.0 | 14.0 | 306.0 | |
| PA (%) | 53.8 | 46.7 | 66.7 | 100.0 | 64.3 | 83.1 | 93.0 | 93.6 | 50.0 | 100.0 | 40.0 | 64.3 | | 79.1 |


## Appendix C. Quality control other Leuven.cool variables

A qualitative assessment of the data quality was performed by making scatterplots of these variables for the LC-R stations compared to the AWS stations (Figure C1).

Overall, the measured parameters are within the accuracy given by the manufacturer. There are, however, deviations in the wind and radiation measurement that are attributable to the small location differences (in the order of meters) between the LC-R stations and the official sensors. For wind, we compared the LC-R data with professional 2m wind measurements, but this height is in fact not the standard measurement height for wind since too many ground effects are still into play at this height. For radiation, we already mentioned the design flaw regarding the wind vane dropping a shadow, but also high nearby trees 780 can influence the measurements for low solar elevations (in the case of LC-R01 and LC-R02).

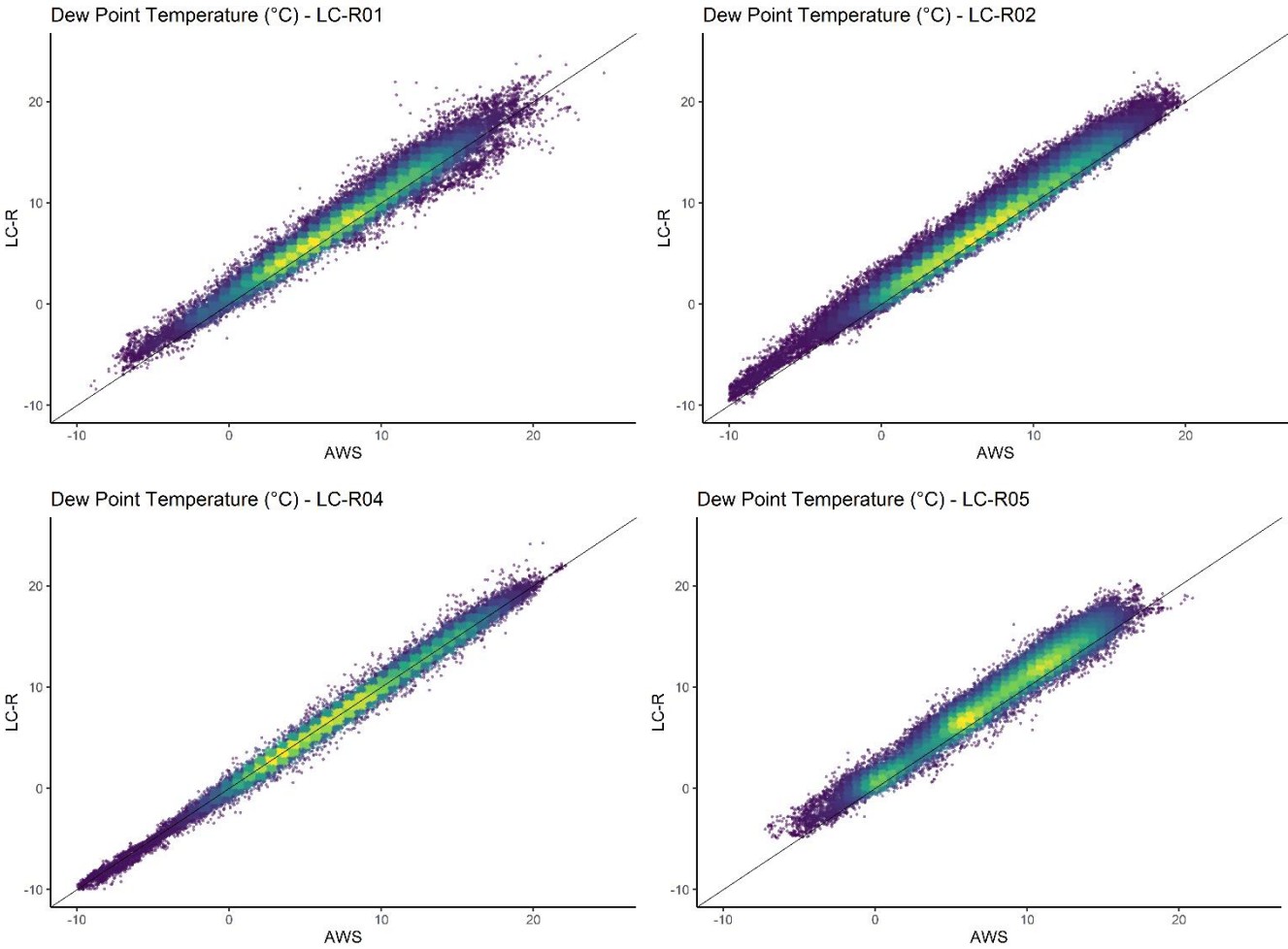

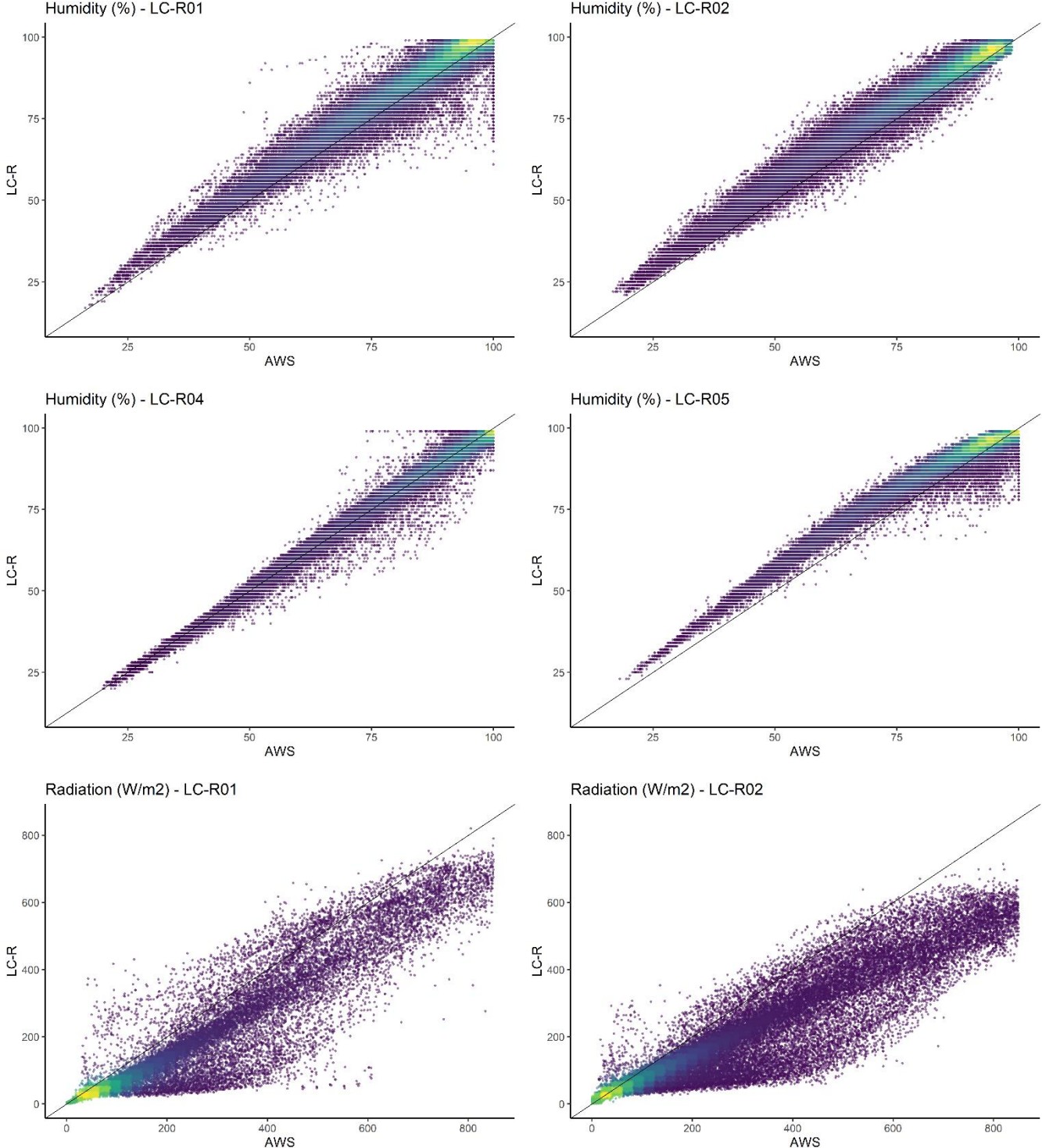


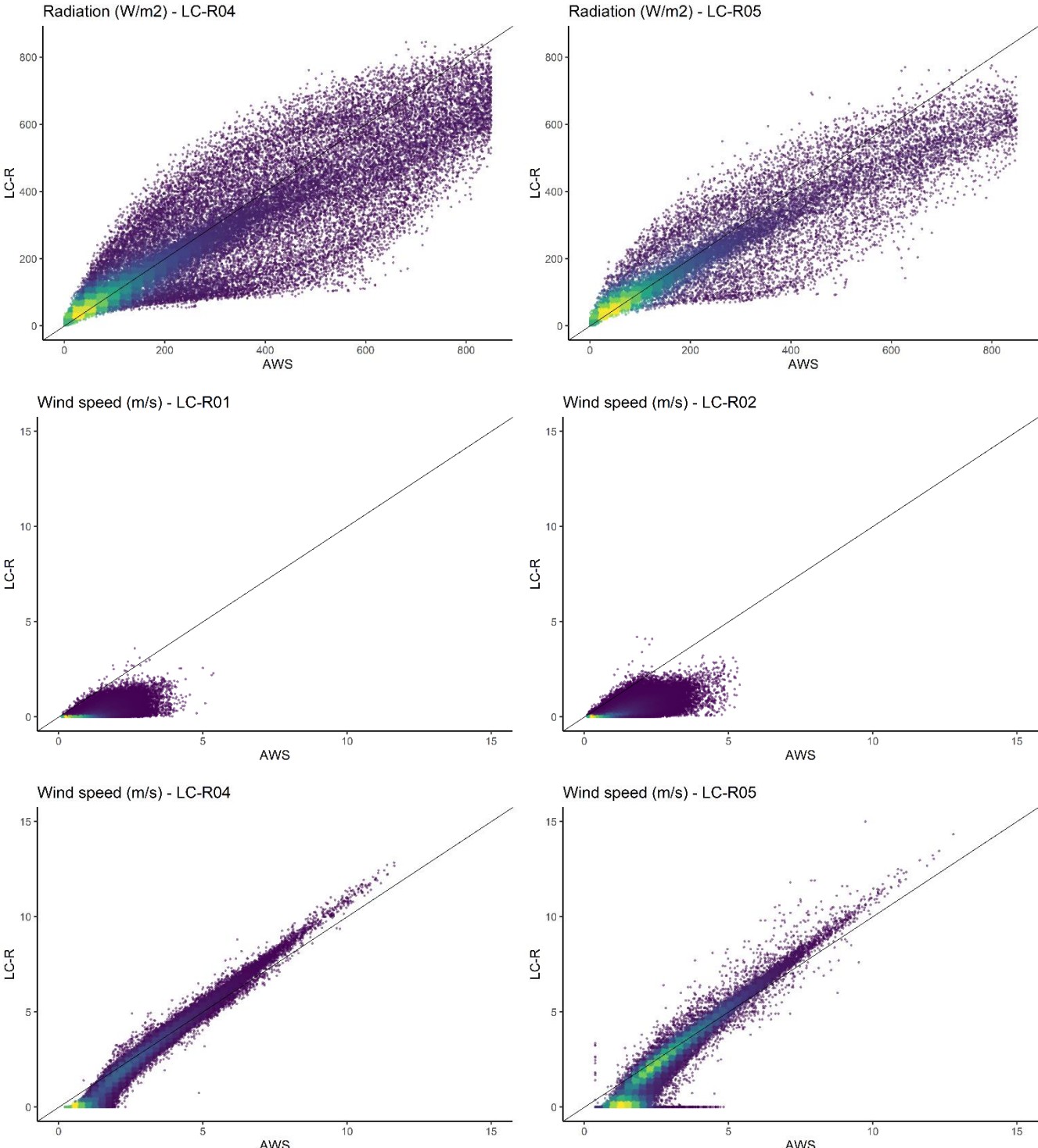

 **Figure C1: Scatterplots of LC-R versus AWS dew point temperature, humidity, radiation and wind speed for each reference station LC-R01 , LC-R02, LC-R04, LC-R05. The identity line is shown in black. The colour scale indicates the density of observations, yellow indicating the highest, purple the lowest. The scatterplots include all measurements between the installation data of each LC-R and December 2021. For each variable the same ranges of x- and y-axis are used.**

**Appendix D. Application RF model on LC-R dataset**

In section 5 the RF model is applied to the LC-R dataset to obtain a corrected temperature for each observation. These results could be biased since the RF model is trained on 60 % of the LC-R dataset, and then applied on the complete LC-R dataset. To account for this we did apply the prediction model to both the complete dataset and the test data set (which was not used for training the model). The outcome for each LC-R is listed in table D1. It can be noted that mean T difference remains equal

for both datasets, the standard deviation does slightly increase with 0.5 to 0.7 °C when using the test dataset only. The mean temperature difference and standard deviation for all LC-R stations increases from $0.00 \pm 0.22$°C to $0.00 \pm 0.28$ °C.

**Table D1: Comparison of the obtained T difference ($\Delta T = T_{LC-R} - T_{AWS}$) when using the complete LC-R dataset and the LC-R test dataset.**

| LC-R | ΔT for complete dataset | ΔT for test dataset |
| --- | --- | --- |
| LC-R01 | $0.01 \pm 0.24$°C | $0.01 \pm 0.30$°C |
| LC-R02 | $0.00 \pm 0.23$°C | $0.00 \pm 0.30$°C |
| LC-R04 | $0.00 \pm 0.20$°C | $0.00 \pm 0.26$°C |
| LC-R05 | $0.00 \pm 0.18$°C | $0.00 \pm 0.23$°C |
| Mean | $0.00 \pm 0.22$°C | $0.00 \pm 0.28$°C |


For comparison the histograms and heatmaps of the temperature difference ($\Delta T = T_{LC-R} - T_{AWS}$) using the complete and test dataset only have been added below in Figure D1, D2 and D3. Only slight differences smaller than 0.1°C exist between the the ΔT for complete dataset and ΔT for test dataset. The diurnal and seasonal pattern is still corrected for (Figure D2). Also, effects of wind speed and radiation are effectively eliminated (Figure D3).


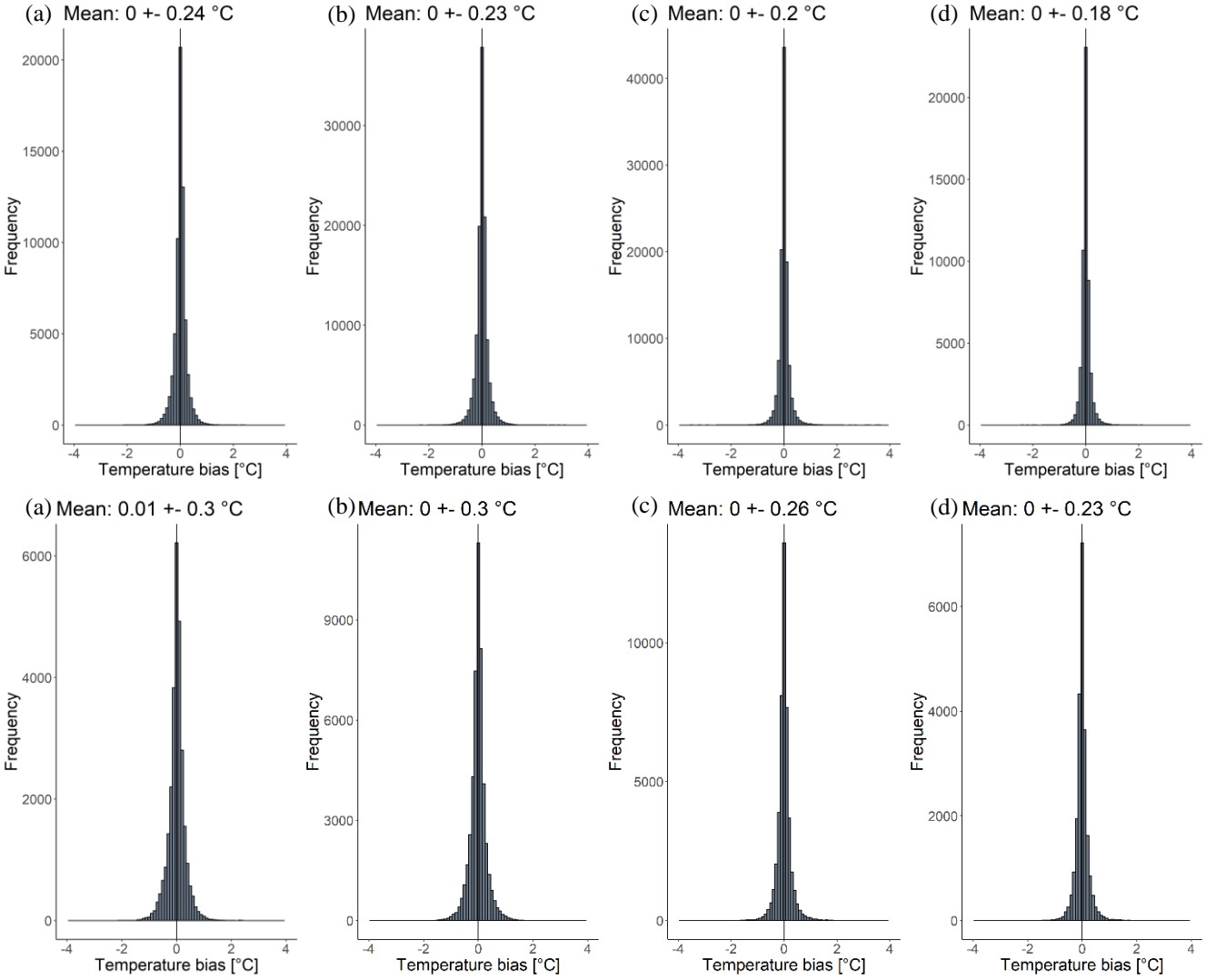

**Figure D1: Same as Figure 9 for QC level 3. Comparison between ΔT for complete dataset (upper row) and ΔT for test dataset (lower row)**


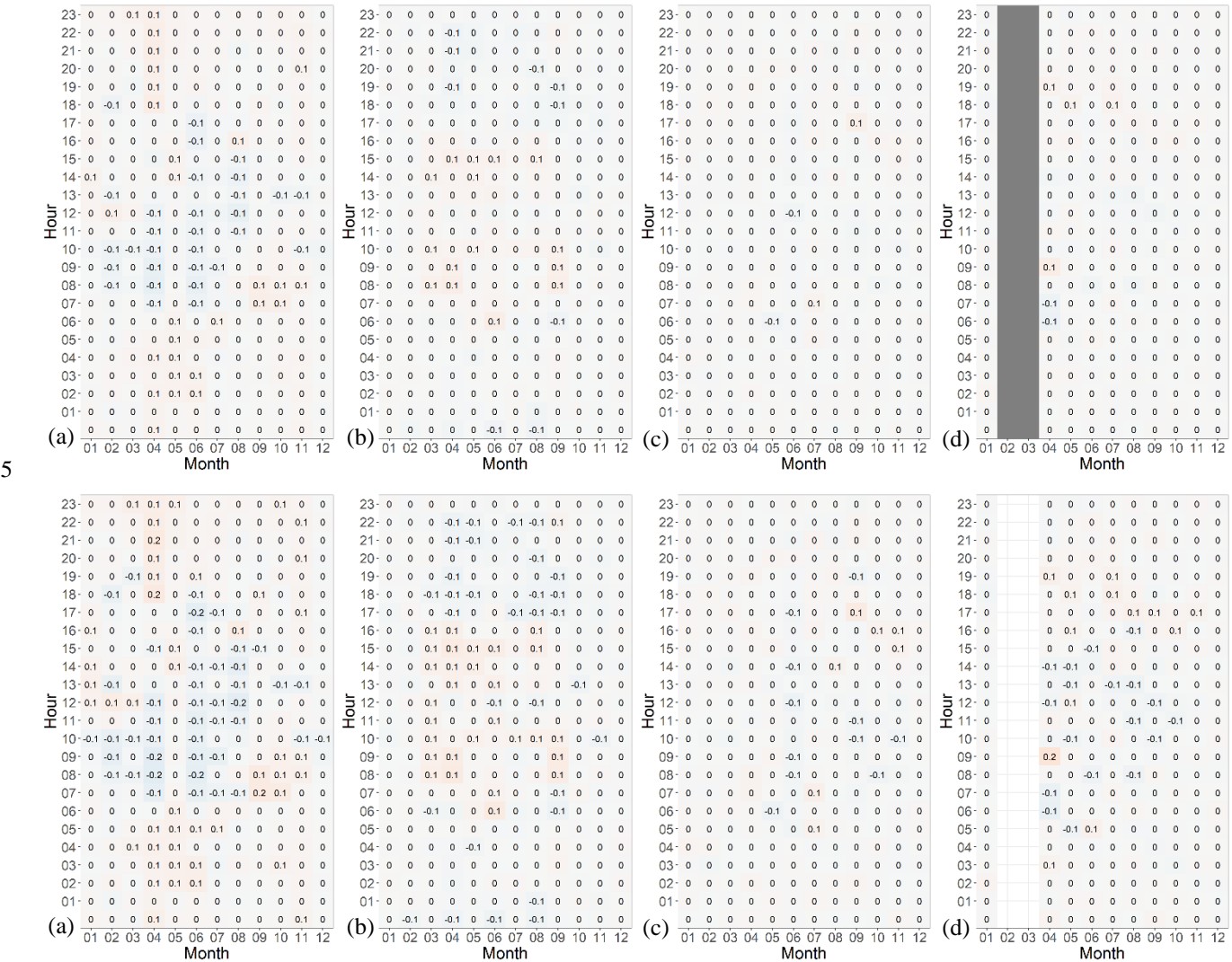

**Figure D2: Same as Figure 10 for QC level 3. Comparison between ΔT for complete dataset (upper row) and ΔT for test dataset (lower row)**

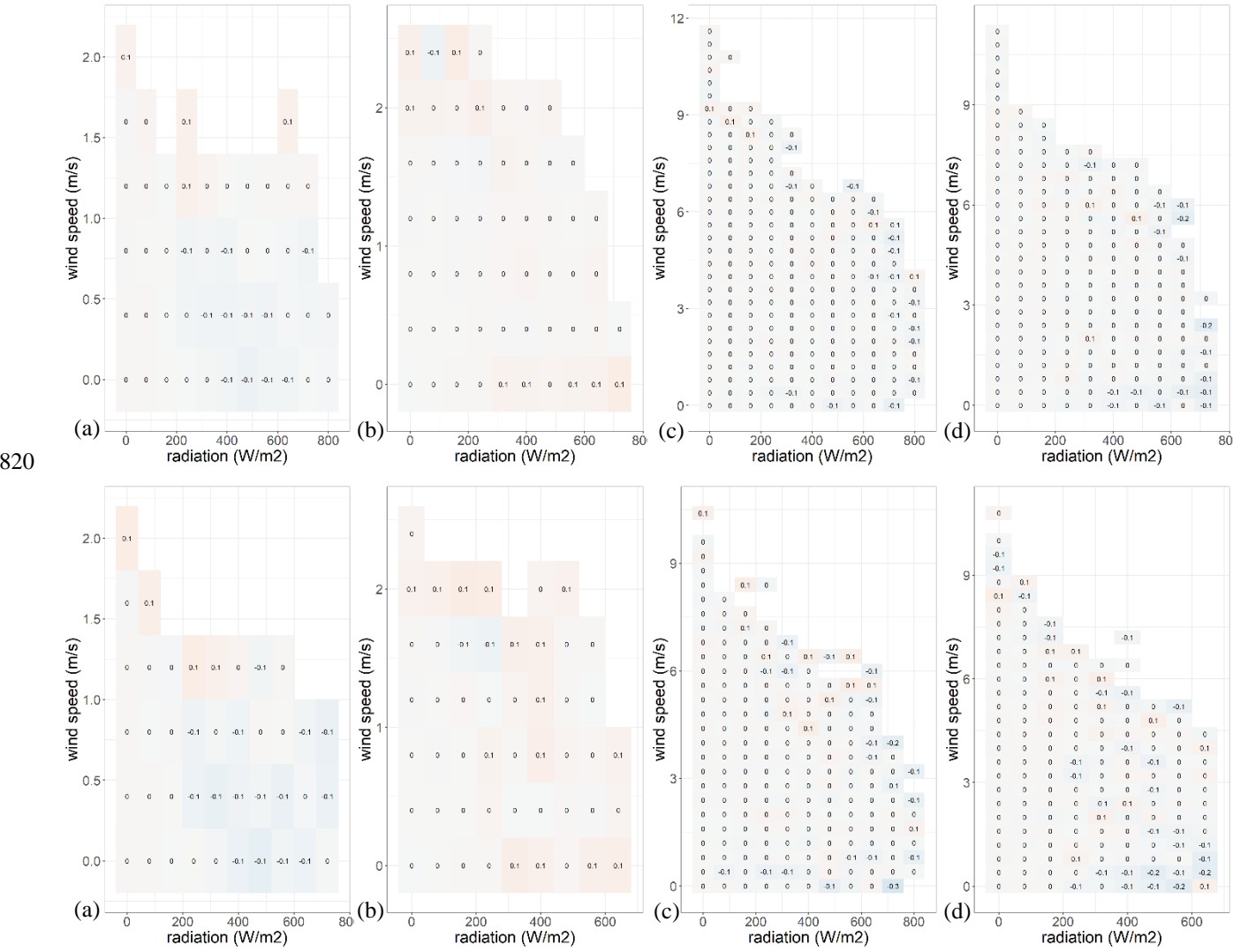


**Figure D3: Same as Figure 11 for QC level 3. Comparison between ΔT for complete dataset (upper row) and ΔT for test dataset (lower row)**

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
