# Peer review of "Quality control and correction method for air-temperature data from a citizen science weather station network in Leuven, Belgium"

_Earth System Science Data, 2022_

## Referee Comment (RC1)

Review for manuscript ESSD-2022-113 by Beele et al.: "Quality control and correction method for air temperature data from a citizen science weather station network in Leuven, Belgium"

**General comments:**

The manuscript deals with a data set of urban meteorological observations made by nearly 100 low-cost weather stations in the city of Leuven, Belgium. Further, the authors present a dedicated quality-control (QC) procedure for the air-temperature data. Though the data set comprises several variables (e.g. air temperature, humidity, wind speed, global radiation, precipitation), the QC only addresses the air-temperature measurements. The development and evaluation of the QC is the focus of the manuscript.

The manuscript and data set are of high relevance, as few cities have dedicated urban measurement networks. The data set can thus be of value to many researchers for various applications.
The manuscript is mostly well written and easily comprehensible. Quality of figures and tables, and their captions, should be improved (see details below). Number of figures and tables could potentially be reduced by combining them.

Overall, the quality of the research is good. My main remark is that the manuscript should highlight better the value of the data set itself for future research (which I think is high!), and that the developed and applied QC is relatively strictly focused on this type of data set and weather station. For example, one could not apply the QC to another crowdsourced data set of, e.g., Netatmo data (radiation data missing) or use it to quality control single/few stations (L1.3, L2).

My opinion is that the comprehensive data set and the fact that it is quality controlled to a high degree is the key selling point, not the developed QC itself. The fact that such a data set exists and that the network of stations will be maintained for some time (until when?) is great. I do not think that the QC is "easily transferable" to other cities, as you need a lot of measurement data for each station, which is not the case for most crowdsourced data sets. Thus I recommend to restructure the manuscript a bit to give more emphasis on the network (how set up, how maintained, by whom, ...) and the data (how collected, what is in the data set, …), and a bit less on the QC. The data set itself on the repository should then be enhanced with additional material regarding meta data of the stations, e.g., Gubler et al. 2021 Supplementary Material (https://doi.org/10.1016/j.uclim.2021.100817).

I hope my comments are helpful to the authors.

Daniel Fenner

**Specific comments:**

- Introduction: At some point when crowdsourced data is introduced, the authors should better highlight that this paper mainly deals with air-temperature data, especially regarding the QC.

- Introduction: In what way is the current data set a "crowdsourcing" data set? Please explain better and compare with the definition of crowdsourcing (e.g. Muller et al. 2015). Or is it more citizen science?

- l. 110-111: This is misleading in my opinion. It is true in your case, but a reader could get the impression that the QC could now (as it was developed and evaluated by you) be implemented in another city without other data. This is not true, as in L1.1 you also need data from an official station and since the thresholds were derived from your single network of stations and might need adjustment in other settings.

- l. 151: "stratified sampling". Please reword, as the number of stations does not represent the percentages (coverage) of the LCZs within the study area. As far as I understand it, the stations were installed to cover a large range of LCZ. But does the number represent the spatial coverage of the LCZ or was it mainly a choice of where a station could be (administration, finding owners, etc.)?

- Table 6: Please make sure that the given parameter names are used in the same way as in the text.

- QC L1.2: Why are the values different for TOathresmax and TOathresmin? Please explain.

- QC L1.2: You give a threshold of 0.05 °C, yet the specified resolution of the sensor is only 0.1 °C. Of course you can obtain more digits due to the averaging, yet this does not reflect the sensor resolution. Additionally, in l. 427 you write 0.1 °C. I suggest to use 0.1 °C.

- QC L1.3: Table 6 and text l. 275: SOthresmin and SOthresmax should be dimensionless, no?

- QC L1.3: With such a low minimum number of stations (nstat = 1) one could falsely remove values that are actually of good quality, if you check each station one by one. Consider three stations, located in a line, each 2 km apart. If the middle station produces false values, it will get flagged out because it deviates too much from the other two (good!). But similarly, checking either one of the stations on the sides will flag that station too, as it deviates too much from the middle station (which actually produces false values) (bad!). Have you investigated using a larger number of neighbours or have you solved this issue somehow differently?

- QC L2: Why was the resampling done to 2 hours? Have you tried different lengths here and determined somehow the optimal duration (if so, how)? Further, are the episodes you find station specific or do they have to overlap across all stations? Please elaborate a bit further.

- Training & validation & RF model: How did you split the data set of LC_R (e.g., randomly or by time)? Do you also train the RF on each LC_X station later on (l. 552: "In QC level 3 the random forest model is applied to each station in order to obtain a site-specific prediction ...")?
The outcome of QC L3 for LC_R is probably not surprising, since you train on the same data set (60 % of it) and then apply the RF model to the whole data set. Is the differentiation between day and night just for illustrative purposes or actually applied in the models?
This section needs better description what has been done.

- Figure 4 and text: What is the authors explanation for the fact that largest biases are found in the middle of the radiation range and not at the top? Please elaborate a bit in the text.

- l. 573: You argue that your data set (since only one type of station) is free of certain effects that would require station-specific QC methods. Yet, in QC L3 you do a station-specific QC, no? Why couldn't you apply the QC to another type of weather station that measures the same variables? Please reword the sentence and/or explain better what you mean.

- l. 585: Who controls the stations and does the maintenance? Please explain in more detail.

- Figure 20: I suggest to include the months January – June in the top sub figure, simply to have the same x range as the other two. Please also ensure that the y range is the same in all sub figures.

- Figures and tables in general:
(a) Maps should contain coordinate axis.
(b) In all figures the subplot labels a, b, c, etc. are missing. This makes it quite difficult to understand some of them and needs to be resolved.
(c) In many figures where different sub-figures are present the axes ranges differ. I propose to have common axes ranges per figure. Alternatively, should should at least give a note in the caption that the ranges differ.
(d) In all figures with coloured background (e.g. Figure 4): Please explain the colour coding and ensure that this is the same in all sub-figures, at best in all of those figures for comparison. Further, how did you group the data (binsize, hourly averages?, etc.)?
(e) The captions need more information regarding the data that is displayed, mainly regarding time period and data sets (this also applies to all tables).
(f) Number of figures: Several figures could be combined, e.g. Figures 9, 13, 15; Figures 10, 14, 16. Please check all figures and check (a) if they are all relevant and (b) if they could be combined. The amount of figures (20) and tables (12) is too much in my sense and should be reduced. For figures that basically display

the same but only differ in the QC level, you could also write in the caption: "Same as figure X but for QC level Y".

- The authors should somewhere better describe the actual data set on the repository (e.g. which variables, which resolution, what kind of meta data, etc.). This would be extremely helpful for other researchers to make use of the data. This section could, e.g., be put before the Conclusions.

- Conclusions: The authors could highlight more clearly the high value of the quality-controlled data set. The large amount of stations in a relatively small city could, e.g., enable detailed comparison with other crowdsourced data where no meta data are available and where the station setup is unknown (e.g. Netatmo). As such, the stations could act as "gatekeepers" for other stations (Chapman et al. 2017). The data set could also be of use to modellers (micro-scale modelling).

- Conclusions: The authors conclude that the method is "easily transferable to other urban regions not having an official weather station". I do not understand how the authors come to this conclusion when such information is need in QC L1.1. Also, "easily transferable" only in that sense, that quite a lot of other data are needed for the QC of air-temperature data. That is certainly not the case for the vast amount of citizen weather stations that exist, from which data could be crowdsourced. The authors should state more clearly what is needed for the QC to be transferred to another regions and reword the section.

- Meta data in the actual data set: I noticed that latitude and longitude values are given with only three digits. I highly recommend to give at least four digits for better precision and to subsequently allow more detailed spatial analyses on the micro scale.

- Do the authors have any idea about the quality of the other variables that are measured and used in the quality control? It could make sense to have relatively simple range and persistence tests (as you developed) implemented for all variables before the detailed QC of air temperature. Please comment on that in the text somewhere.

**Technical corrections:**

- Compound adjective (e.g. "air temperature" but "air-temperature data"). Please check the whole manuscript carefully
- l. 32: consider "… that both cities and their citizens are …"
- l. 50: "… in which air temperature is continuously modelled …"
- l. 101: consider "Thus, there is a need for …"
- l. 104: "Here we report on a …"
- l. 169: "… is limited due to the …"
- l. 178: "direct effect contact" ? Please reword.
- l. 182: "aggregated" – Do you mean temporally averaged?
- l. 194 (and other instances in the text, Table A1): There should be a space between the number and the unit, except for geographical degrees; cf. chapter 5.4.3 in https://www.bipm.org/documents/20126/41483022/SI-Brochure-9.pdf/fcf090b2-04e6-88cc-1149-c3e029ad8232).
- l. 266-267: Please check sentence and reword.
- l. 285: "non-negligible" – How did you determine that? What is the basis for the statement?
- l. 298: "i.e." is instead of "e.g."
- l. 300: Could be helpful to say "… across all ten selected episodes is considered …"
- Table 7: What is actually shown here? What kind of correlation coefficient? Per definition, a correlation coefficient has no units. Please remove them and specify more clearly in the caption what is shown.
- Figure 8: What is the black line? Please explain in the caption.
- l. 447 & 448: Where is the information regarding the summer day shown? Please specify or reword.

**References:**

Chapman, L., Bell, C. and Bell, S. (2017): Can the crowdsourcing data paradigm take atmospheric science to a new level? A case study of the urban heat island of London quantified using Netatmo weather stations. Int. J. Climatol. 37 (9): 3597-3605. DOI: 10.1002/joc.4940

Gubler, M., Christen, A., Remund, J. and Brönnimann, S. (2021): Evaluation and application of a low-cost measurement network to study intra-urban temperature differences during summer 2018 in Bern, Switzerland. Urban Clim. 37: 100817. DOI: 10.1016/j.uclim.2021.100817

Muller, C. L., Chapman, L., Johnston, S., Kidd, C., Illingworth, S., Foody, G., Overeem, A. and Leigh, R. R. (2015): Crowdsourcing for climate and atmospheric sciences: current status and future potential. Int. J. Climatol. 35 (11): 3185-3203. DOI: 10.1002/joc.4210

---

## Referee Comment (RC2)

Small typographic errors (missing punctuation or erroneous capitalization) that proofreaders should catch.

Data accessible, clearly described, easy to work with. This reviewer prefers .csv to .tab but that seems a matter of individual preference. I understand why authors might prefer CC-BY-NC but, due to proper ESSD cautions, have editors granted permission for -NC?

Overall: good well-organized description of QC levels and good summary of 'final' QC outcome, $0.15 \pm 0.56$ down to $0.00 \pm 0.22$ for temperature. In describing and summarizing these efforts and QC levels, the authors seem to have left out or passed over several issues.

Daytime vs nighttime much used but little described. Based on daily hours sunlight, calculated by day, month or season? Or calculated based on measured surface radiation values where night = low light = < some minimum value of W/m2?

Other parameters (RH, precip, wind) not corrected. Authors describe, correctly, the use of non-T parameters in various QC functions but otherwise give no hints about suitability, difficulty or desirability of addressing RH, precip, etc. Do readers await a future paper? Do we assume QC of T proved easier than other parameters? Authors give us no hint what to assume!

Authors should, technically, express offsets and SD in K rather than C. Not often done, I accept.

Authors have lumped 2 m data with 3-4 m data? Early mention but then no subsequent treatment. One suspects both sensor elevation and surrounding (mostly impervious) surfaces would have a large effect but never mentioned? Shadowing by anemometer mentioned occasionally but that occurs independently of short vs high poles and regardless of underlying surface?

UHI: This reader missed an overall assessment of data as QC'd here to address UHI. Do we need 0.1K? 1.0K? Have the authors come close with these corrections. In view of distance of reference AWS from Leuven and, for two RMIB stations at least, distance from true urban settings, do authors feel they have a network now suitable for addressing UHI. Not clear, perhaps needs/deserves further clarification. Given sensor height differences already mentioned plus apparent absence (or, avoidance) of the most urban land use categories, can users really trust this data for further UHI work? UHI mentioned frequently in introduction but not at all in conclusion paragraph. I agree with summary sentences but, if they could not in the end address UHI - which, in many indices, includes high net radiation, low RH and low wind - should they have given so much attention in introduction?

---

## Author Comment (AC1)

**Authors' response to RC1 - manuscript ESSD-2022-113**

Responses to Reviewer comments: The text in blue represents the authors' reply. Text in italic represents text that was changed in the manuscript, with line numbers referring to those in the revised manuscript. We numbered the comments for easier referral.

**General comments:**

My main remark is that the manuscript should highlight better the value of the data set itself for future research (which I think is high!), and that the developed and applied QC is relatively strictly focused on this type of data set and weather station. For example, one could not apply the QC to another crowdsourced data set of, e.g., Netatmo data (radiation data missing) or use it to quality control single/few stations (L1.3, L2).

My opinion is that the comprehensive data set and the fact that it is quality controlled to a high degree is the key selling point, not the developed QC itself. The fact that such a data set exists and that the network of stations will be maintained for some time (until when?) is great. I do not think that the QC is "easily transferable" to other cities, as you need a lot of measurement data for each station, which is not the case for most crowdsourced data sets. Thus I recommend to restructure the manuscript a bit to give more emphasis on the network (how set up, how maintained, by whom, ...) and the data (how collected, what is in the data set, …), and a bit less on the QC. The data set itself on the repository should then be enhanced with additional material regarding meta data of the stations, e.g., Gubler et al. 2021 Supplementary Material (https://doi.org/10.1016/j.uclim.2021.100817).

**Reply:** We thank the reviewer for the thorough review and useful comments and suggestions that improved our manuscript. We have considered all your comments and revised the manuscript accordingly. Please see our point-by-point responses on the specific comments below.

We agree that the value of this dataset is of high relevance and has many useful applications. Those applications were currently not introduced in the manuscript. We incorporated them in an additional section; Section 6. Application potential of the quality controlled and corrected Leuven.cool dataset. The main applications are further summarised in the conclusion.

*'6. Application potential of the quality controlled and corrected Leuven.cool dataset*

*A validation of the proposed QC method showed that it can reduce the mean temperature difference and standard deviation from 0.15 ± 0.56 °C to 0.00 ± 0.28 °C. The QC method can correct the temperature difference equally across different hours of the day and months of the year (Figure 16) as well and under different radiation and windspeed conditions (Figure 17).*

*The quality-controlled Leuven.cool dataset enables a detailed comparison with other crowdsourced datasets for which less or even no metadata is available. As such, the Leuven.cool stations can serve as gatekeepers for other crowdsourced observations. In the past this role has been limited to standard weather station network which mostly only have a limited number of observations available (Chapman et al., 2017).*

*Numerous studies have shown that the UHI effect causes night time temperature differences up 6 to 9 °C during clear nights (Chapman et al., 2017; Feichtinger et al., 2020; Napoly et al., 2018; Stewart, 2011; Venter et al., 2021). These thresholds are much higher than mean bias obtained after correction. The dense quality controlled Leuven.cool dataset thus allows for microscale modelling of urban weather patterns, including the urban heat island (Chapman et al., 2017; de Vos et al., 2020; Feichtinger et al., 2020; Napoly et al., 2018). Since such high-quality datasets contain measurements*

*with both high spatial and temporal resolution, they can easily be used to obtain spatially continuous temperature patterns across a region (e.g. Feichtinger et al., 2020; Napoly et al., 2018). Interpolation methods based on single pair stations or mobile transect methods are much less trustworthy (Napoly et al., 2018). Dense weather station networks can be used to investigate the inter- and intra LCZ variability within a city (Fenner et al., 2017; Verdonck et al., 2018). The dataset can further help investigate the relation between temperature and human and ecosystem health (e.g. Aerts et al., 2022; Troeyer et al., 2020) and their effect on evolutionary processes (e.g. Brans et al., 2022).*

*The dataset can also help refine existing weather forecast models which are currently mostly based on official rural observations (Sgoff et al., 2022). Nipen et al. (2020) showed that the inclusion of citizen observations improves the accuracy of short-term temperature forecasts in regions where official stations are sparse. Mandement & Caumont (2020) used crowdsourced weather stations to improve the observation and prediction of near convection. After quality control and correction, also the wind (Chen et al., 2021) and precipitation measurements (de Vos et al., 2019) can be useful to improve detection and forecasting. Further, the Leuven.cool dataset could be a useful input in air pollution prediction models (e.g. IFD-model (Lefebvre et al., 2011)).'* (Lines 673-701)

*'The quality-controlled Leuven.cool dataset enables a detailed comparison with other crowdsourced datasets for which less or even no metadata is available. The dense dataset further allows for microscale modelling of urban weather patterns, such as the urban heat island, and can help identify the relation between temperature and human and ecosystem and their effect on evolutionary processes. Lastly the dataset could be used to refine existing forecast models which are currently mostly based on official rural observations.'* (Lines 723-727)

**Reply:** We further agree that the QC method was built for this specific data set but believe that it can be adapted to similar crowdsourced datasets and regions. When no climate thresholds are directly available, thresholds can be based on existing climate classification maps (QC L1.1). If no estimation can be made, QC L1.1 can be removed. The QC method was built for weather station networks is thus less suitable to quality control single or few stations (QC L1.3, QC L2). Lastly, the random forest model (QC L3) is easily adaptable to the parameters that are available (we do acknowledge, however, that radiation is the most important parameter in the correction procedure). As an example, we could not use atmospheric pressure measurements in our case, since these were not correctly transferred by the receiver for many of the stations within the Leuven.cool network. In the conclusion paragraph, we summarised what is needed for the QC to be transferred to other regions

*'Afterwards the method can be applied independently of the official network that was used in the development phase. Transferring the method to other networks or regions would require the recalibration of the QC parameters. Specifically, for QC L1.1 some indication of climate thresholds is needed. QC L1.3 and QC L2 require a dense weather station network, the QC method is thus less suitable for single or few stations. For QC L3 quite some other data (e.g., radiation, windspeed, ...) are needed. The random forest model is however easily adaptable to the parameters that are available.'* (Lines 714-719)

**Reply:** To improve the usability of the dataset, the metadata on the data repository will be enhanced with the following parameters: installation height, LCZ, dominant landcover, mean SVF and mean building height in buffer of 10m around weather station.

**Specific comments:**

1. Introduction: At some point when crowdsourced data is introduced, the authors should better highlight that this paper mainly deals with air-temperature data, especially regarding the QC.

**Reply:** We agree that the link to air temperature might not have been very clearly stated in the introduction. Within this introduction crowdsourced data is introduced as a promising solution to measure the urban heat island. The connection to air-temperature data seemed logic but was indeed not specified. The overview of currently existing QC methods did mostly focus on air-temperature data. We changed the introduction accordingly.

*'The rise of crowdsourced air-temperature data, especially in urban areas, could be a promising solution to bridge this knowledge gap (Muller et al., 2015).'* (Lines 56-57)

*Recent studies have therefore highlighted the importance of performing a data quality control in data processing applications (Båserud et al., 2020; Longman et al., 2018), especially before analysing crowdsourced air-temperature data (Bell et al., 2015; Chapman et al., 2017; Cornes et al., 2020; Feichtinger et al., 2020; Jenkins, 2014; Meier et al., 2017; Napoly et al., 2018; Nipen et al., 2020).* (Lines 69-72)

*'Current QC studies mostly identify and remove implausible temperature measurements (Chapman et al., 2017; Meier et al., 2017; Napoly et al., 2018), instead of correcting for known temperature biases (Cornes et al., 2020).'* (Lines 95-96)

2. Introduction: In what way is the current data set a "crowdsourcing" data set? Please explain better and compare with the definition of crowdsourcing (e.g. Muller et al. 2015). Or is it more citizen science?

**Reply:** We understand the confusion between both terms. The definition "crowdsourcing" and "citizen science" is still somehow fluid, and certainly not settled at this time.

Muller et al. (2015) states that crowdsourcing was traditionally defined as "obtaining data or information by enlisting the services of a (potentially large) number of people" but highlights that the definition should be expanded to include "and/or from a range of public sensors, typically connected via the Internet". In the past, crowdsourcing was merely understood as the outsourcing of a certain task to the general public but in more recent years it is seen as the "lowest level" of citizen science, i.e., citizens contributing to sensing (see, e.g., the review article of Zheng et al. (2018)). Citizen science itself is then more broadly defined as "the involvement of citizens in science".

In this definition, the Leuven.cool weather station network is crowdsourcing in the first place, and hence also citizen science. The weather station of Leuven.cool are installed in both private gardens of citizens and semi-public locations in Leuven. The volunteering citizens provided us with a location for the weather station and are responsible for simple maintenance tasks. It is this terminology we had in mind when drafting the manuscript.

*'Here we report on a statistically-based QC method for the crowdsourced air-temperature data of the Leuven.cool network, a citizens science network of almost 100 weather stations distributed across private gardens and (semi-) public locations in Leuven, Belgium.'* (Lines 105-107)

3. l. 110-111: This is misleading in my opinion. It is true in your case, but a reader could get the impression that the QC could now (as it was developed and evaluated by you) be implemented in another city without other data. This is not true, as in L1.1 you also need data from an official station and since the thresholds were derived from your single network of stations and might need adjustment in other settings.

**Reply:** Thanks for this good remark. This is (unintentionally) indeed misleading. Transferring the method to other networks would indeed require the recalibration of the QC parameters. These lines are rephrased in the manuscript. In the conclusion we give a complete overview of what is needed for the QC to be transferred to another regions.

*'The QC method only needs an official network during its development and evaluation stage. Afterwards the method can be applied independently of the official network that was used in the development phase. Transferring the method to other networks or regions would require the recalibration of the QC parameters.' (*Lines 112- 117)

*'Afterwards the method can be applied independently of the official network that was used in the development phase. Transferring the method to other networks or regions would require the recalibration of the QC parameters. Specifically, for QC L1.1 some indication of climate thresholds is needed. QC L1.3 and QC L2 require a dense weather station network, the QC method is thus less suitable for single or few stations. For QC L3 quite some other data (e.g., radiation, windspeed, ...) are needed. The random forest model is however easily adaptable to the parameters that are available.'* (Lines 714-719)

4. l. 151: "stratified sampling". Please reword, as the number of stations does not represent the percentages (coverage) of the LCZs within the study area. As far as I understand it, the stations were installed to cover a large range of LCZ. But does the number represent the spatial coverage of the LCZ or was it mainly a choice of where a station could be (administration, finding owners, etc.)?

**Reply:** We agree that the term stratified sampling is not completely correct. The selection procedure started as a stratified sampling based on the concept of LCZs (by looking at building height, building density and vegetation cover). The goal was to divide the stations across the LCZ classes so that they would represent the spatial coverage of each LCZ. We should state that the network was implemented with the intention of gaining knowledge on the mitigating effect of green and blue infrastructures within urban settings. The initial study area thus mainly included the city centre of Leuven. As a result, the network thus has a bias towards urban classes.

Figure 1 shows the distribution of the weather stations across the LCZ classes (left panel) and proportion of LCZ classes across the initial study area (right panel). Due to the complex urban settings in which the network is deployed, practical limitations apply to the eligible locations for installation. We rely on volunteering citizens, private companies and government institutions giving permission to install a weather station on their property. Due to a technical limitation of the weather station, it cannot be installed in natural environments without a LAN connection within 50 to 100 meters of the weather station.

[Figure]

*Figure 1: Distribution of weather stations across LCZ classes (left panel) and proportion of LCZ classes across study area (right panel).*

*'From July 2019 onwards the weather stations were distributed along an urban gradient from green (private) gardens to public grey locations following a sampling design based on the concept of Local Climate Zones (LCZ) (Figure 2) (Stewart & Oke, 2012).'* (Lines 158-160)

*'It can be noted that the weather stations are not evenly distributed across the different LCZ classes, also the number of weather station within a LCZ class does not represent the spatial coverage of this LCZ class. Due to the complex urban settings in which the network is deployed, practical limitations apply to the eligible locations for installation. We rely on volunteering citizens, private companies and government institutions giving permission to install a weather station on their property. Further, the middle-sized city of Leuven does not contain all available LCZ classes. In the urban context compact high-rise, lightweight low-rise and heavy industry is missing. In the natural context, brush or shrub vegetation, bare rock or paved and bare soil or sand are not present in sufficiently large areas (Table 2). Furthermore, the number of stations within more natural settings is limited due to the technical limitations of the weather stations; each outdoor unit needs a base station, with both power and LAN connection, within 50 to 100 meters in order to transmit its data. Lastly, the network was implemented with the intention of gaining knowledge on the mitigating effect of green and blue infrastructures within urban settings. The weather station network thus mostly focusses on urban classes.* (Lines 173-183)

**Additional note:** After submitting the manuscript in May/April 2022, we noticed that a LCZ map based on worldwide datasets, as the methodology proposed by Demuzere et al. (2021), is not optimal for fine-scale applications as the one presented here (Verdonck et al., 2017). Behind the scenes we have developed a new LCZ map based on fine-scale land use, building height, building density and green ratio data. The methodology was performed using a supervised random forest classification approach in R. All details on the LCZ map are summarised under appendix B.

Since the review process of this manuscript took longer than originally expected, it seemed more straightforward to directly update the old LCZ map with the new LCZ map. In the meantime, additional stations have been installed. The Leuven.cool networks currently counts 106 weather stations.

*'Stewart & Oke (2012) define 17 LCZ classes, divided into 10 urban LCZs (1-10) and 7 natural LCZs (A-G). A LCZ map for Leuven was developed using a supervised random forest classification approach based on fine-scale land use, building height, building density and green ratio data. Details on this LCZ map are available in Appendix B. . Table 2 summarises the LCZs present in Leuven and the number of weather stations in each LCZ class.*

*Table 2: LCZs present in Leuven and the number of weather stations in each LCZ class.*

| LCZ ID | LCZ description | # Stations (106) |
|---|---|---|
| LCZ 2 | Compact midrise | 20 |
| LCZ 3 | Compact low-rise | 7 |
| LCZ 4 | Open high-rise | 3 |
| LCZ 5 | Open midrise | 24 |
| LCZ 6 | Open low-rise | 19 |
| LCZ 8 | Large low-rise | 2 |
| LCZ 9 | Sparsely built | 16 |
| LCZ A | Dense trees | 0 |
| LCZ B | Scattered trees | 10 |
| LCZ D | Low plants | 4 |
| LCZ E | Bare rock or paved | 0 |
| LCZ G | Water | 1 |

(Lines 165-172)

5.   Table 6: Please make sure that the given parameter names are used in the same way as in the text.

**Reply:** Thanks for noticing this mistake. Adjustments were made in the manuscript.

*Table 6: Parameter settings for QC level 1 - Outlier detection.*

| Outlier parameter | Value (unit) | Description |
|---|---|---|
| **Range outliers (RO)** | | |
| dev_reference | 1 (°C) | Max allowed deviation between climatological min and max temperature of AWS stations in Uccle/Diepenbeek/Humain and LC-R in Uccle/Diepenbeek/Humain |
| dev | 5 (°C) | Max allowed deviation between climatological min and max temperature of AWS station in Uccle and LC-X in Leuven |
| **Temporal outliers (TO)** | | |
| $TOa_{ThresMin}$ | -3 (°C) | Min allowed difference between sequential 10 min observations |
| $TOa_{ThresMax}$ | 2.5 (°C) | Max allowed difference between sequential 10 min observations |
| $TOb_{ThresMin}$ | 0.05 (°C) | Min difference that should be noted in $TOb_{Timespan}$ |
| $TOb_{Timespan}$ | 19 (-) | Number of consecutive 10 min observations in which temperature should change with $TOb_{ThresMin}$ |
| **Spatial outliers (SO)** | | |
| range | 2500 (m) | Range used to define neighbouring stations |
| $SO_{ThresMin}$ | -3 (-) | Min allowed Z-score |
| $SO_{ThresMax}$ | 3 (-) | Max allowed Z-score |

| | | |
|---|---|---|
| *nstat* | *1 (-)* | *Minimum requirement of measurements in range* |

(Lines 262-264)

6.  QC L1.2: Why are the values different for TOathresmax and TOathresmin? Please explain.

**Reply:** The values of TOa$_{ThresMin}$ and TOa$_{ThresMax}$ differ for meteorological reasons. The cooling down of air temperatures is meteorologically seen faster (e.g. through the passing of a cold front or thunderstorm) compared to the heating up of air temperatures. Since cold air is less dense than warm air, it will rapidly replace warmer air preceding the front's edge. Cold fronts are thus known to move faster than warm fronts hereby producing sharp changes in weather. (Ahrens, 2009).

*'The values of TOa$_{ThresMin}$ and TOa$_{ThresMax}$ differ for meteorological reasons. The cooling down of air temperatures will, from a meteorological point of view, occur faster (e.g. through the passing of a cold front or thunderstorm) compared to the heating up of air temperatures (Ahrens, 2009).'* (Lines 280-282)

7.  QC L1.2: You give a threshold of 0.05 °C, yet the specified resolution of the sensor is only 0.1 °C. Of course you can obtain more digits due to the averaging, yet this does not reflect the sensor resolution. Additionally, in l. 427 you write 0.1 °C. I suggest to use 0.1 °C.

**Reply:** We understand the concern of the reviewer, but we should state that defining the temperature threshold of the persistence test has been quite time consuming. The goal of this test is to identify stations that repeatedly send out the same observation.

When setting the threshold equal to 0.1°C too many good observations were thrown away. To identify the most ideal threshold, an iterative procedure on the AWS data was performed. Here we assumed that the AWS data is correct, the persistence test should thus not return any flagged observation (or only return a limited number of flagged observations). Setting TOb$_{ThresMin}$ and TOb$_{Timespan}$ equal to 0.05°C and 19 respectively returned the best results.

*'The results show only a few range outlier and even no spatial outliers for the LC-R reference stations. The procedure does however highlight 180 observations as temporal outliers (Table 10). These observations were highlighted during the persistence test, 180 observations change less than 0.05 °C within 2 hours.'* (Lines 459-461)

8.  *QC L1.3: Table 6 and text l. 275: SOthresmin and SOthresmax should be dimensionless, no?*

**Reply:** Thanks for the good remark. SOthresmin and SOthresmax are indeed dimensionless. This has been adjusted in the manuscript.

*Table 6: Parameter settings for QC level 1 - Outlier detection.*

| Outlier parameter | Value (unit) | Description |
|---|---|---|
| **Range outliers (RO)** | | |
| *dev_reference* | *1 (°C)* | *Max allowed deviation between climatological min and max temperature of AWS stations in Uccle/Diepenbeek/Humain and LC-R in Uccle/Diepenbeek/Humain* |

| | | |
|---|---|---|
| *dev* | *5 (°C)* | *Max allowed deviation between climatological min and max temperature of AWS station in Uccle and LC-X in Leuven* |
| ***Temporal outliers (TO)*** | | |
| *TOa$_{ThresMin}$* | *-3 (°C)* | *Min allowed difference between sequential 10 min observations* |
| *TOa$_{ThresMax}$* | *2.5 (°C)* | *Max allowed difference between sequential 10 min observations* |
| *TOb$_{ThresMin}$* | *0.05 (°C)* | *Min difference that should be noted in TOb$_{Timespan}$* |
| *TOb$_{Timespan}$* | *19 (-)* | *Number of consecutive 10 min observations in which temperature should change with TOb$_{ThresMin}$* |
| ***Spatial outliers (SO)*** | | |
| *range* | *2500 (m)* | *Range used to define neighbouring stations* |
| *SO$_{ThresMin}$* | *-3 (-)* | *Min allowed Z-score* |
| *SO$_{ThresMax}$* | *3 (-)* | *Max allowed Z-score* |
| *nstat* | *1 (-)* | *Minimum requirement of measurements in range* |

(Lines 262-264)

*'Whenever the Z-score is lower than -3 (SOThresMin) or higher than 3 (SOThresMax) the observation is seen as a spatial outlier and receives a flag equal to 1.'* (Lines 296-297)

9. QC L1.3: With such a low minimum number of stations (nstat = 1) one could falsely remove values that are actually of good quality, if you check each station one by one. Consider three stations, located in a line, each 2 km apart. If the middle station produces false values, it will get flagged out because it deviates too much from the other two (good!). But similarly, checking either one of the stations on the sides will flag that station too, as it deviates too much from the middle station (which actually produces false values) (bad!). Have you investigated using a larger number of neighbours or have you solved this issue somehow differently?

**Reply:** Thanks for the comment. We have investigated this using a larger minimal number of neighbours, but the results stayed very similar for our dataset. Within our weather station network there is only 1 weather station (LC-064), situated in Lovenjoel south-east of Leuven, which only has one neighbour (LC-002). For both stations no spatial outliers were found. If the minimal number of neighbours is increased, no spatial outliers could have been calculated for LC-064.

This is however a very good remark that will be added to the Outlier settings document on the data repository.

10. QC L2: Why was the resampling done to 2 hours? Have you tried different lengths here and determined somehow the optimal duration (if so, how)? Further, are the episodes you find station specific or do they have to overlap across all stations? Please elaborate a bit further.

**Reply:** A qualitative assessment was used to define the length of the episode. If we would have taken a shorter period (≤ 1h), there is a possibility to select a period without continuous high windspeed conditions. This could be the start of the passing of a cold front or thunderstorm or a dynamic episode during night time. These episodes should be excluded since UHI effects and other local differences will

still be present. For longer periods (> 2h), the diurnal cycle will influence the temperature measurements while we aim to select episodes with a constant temperature across Leuven.

Episodes are found by resampling the meteorological variables across all weather stations for each 2 hour interval. These intervals are filtered based on different conditions for rainfall, radiation and wind speed. The episodes are thus not station specific.

11. Training & validation & RF model: How did you split the data set of LC_R (e.g., randomly or by time)? Do you also train the RF on each LC_X station later on (l. 552: "In QC level 3 the random forest model is applied to each station in order to obtain a site-specific prediction ...")? The outcome of QC L3 for LC_R is probably not surprising, since you train on the same data set (60 % of it) and then apply the RF model to the whole data set. Is the differentiation between day and night just for illustrative purposes or actually applied in the models? This section needs better description what has been done.

**Reply:** The LC_R dataset was randomly split in training (0.60) and validation data (0.40).

*'For the construction of a predictor model, the dataset was randomly split in training (0.60) and validation (0.40) data.'* (Line 335)

**Reply:** Thanks for the question. No, we do not train the RF model on each LC_X. Once the RF model is built (based on the LC_R), it is simply applied to each observation of the LC_X dataset. The RF model cannot be trained on each LC-X station since no official measurement equipment is available in or nearby Leuven. Consequently, a temperature bias cannot be calculated for the LC_X stations.

*'In QC level 3 the random forest model is applied to each temperature observation of all LC-X stations in order to obtain a site-and time-specific prediction for its temperature bias.'* (Line 612-613)

**Reply:** The outcome of QC L3 for the LC_R dataset is indeed not surprising. For the creation and validation of the random forest model, the reference dataset (R01, R02, R43 & R05) is split in to training (0.6) and test data (0.4). The histograms in figure 5 and figure 6 are thus based on the test data only. However, in order to obtain a corrected temperature for each observation, during the evaluation step the random forest model is applied to the complete dataset.

We understand that this could potentially underestimate the obtained T difference ($\Delta T = T_{LC-R} - T_{AWS}$). To account for this we did apply the prediction model to both the complete dataset and the test data set (which was not used for training the model). The outcome for each LC-R is listed in Table 1. It can be noted that mean T difference remains equal, while the standard deviation slightly increases with 0.5 to 0.7°C. We have added appendix D to the manuscript explaining the difference between using the complete dataset versus the test dataset. The results in the manuscript have been adjusted and are now based on the test dataset only.

*Table 1: Comparison of the obtained T difference ($\Delta T = T_{LC-R} - T_{AWS}$) when using the complete LC-R dataset and the LC-R test dataset.*

| LC-R | ΔT for complete dataset | ΔT for test dataset |
|------|-------------------------|---------------------|
| LC-R01 | $0.01 \pm 0.24$°C | $0.01 \pm 0.30$°C |
| LC-R02 | $0.00 \pm 0.23$°C | $0.00 \pm 0.30$°C |

| | | |
|---|---|---|
| LC-R04 | 0.00 ± 0.20°C | 0.00 ± 0.26°C |
| LC-R05 | 0.00 ± 0.18°C | 0.00 ± 0.23°C |
| Mean | 0.00 ± 0.22°C | 0.00 ± 0.28°C |

*'From Section 3, we recall that the random forest prediction of temperature bias showed the best results After applying this prediction model on the reference dataset (LC-R), a level 3 corrected temperature is obtained for each LC-R station. These results could be biased since the RF model is trained on 60 % of the LC-R dataset, and then applied on the complete LC-R dataset. To account for this we did apply the prediction model to both the complete dataset and the test data set (which was not used for training the model). The outcome for each LC-R is listed in Appendix D. It can be noted that mean T difference remains equal for both datasets, the standard deviation does slightly increase with 0.5 to 0.7 °C when using the test dataset. The results below are based on the test dataset only. (*Lines 544-550)

*'Appendix D. Application RF model on LC-R dataset*

*In section 5.the RF model is applied to the LC-R dataset to obtain a corrected temperature for each observation. These results could be biased since the RF model is trained on 60 % of the LC-R dataset, and then applied on the complete LC-R dataset. To account for this we did apply the prediction model to both the complete dataset and the test data set (which was not used for training the model). The outcome for each LC-R is listed in table D1. It can be noted that mean T difference remains equal for both datasets, the standard deviation does slightly increase with 0.5 to 0.7 °C when using the test dataset only. The mean temperature difference and standard deviation for all LC-R stations increases from 0.00 ± 0.22°C to 0.00 ± 0.28 °C.*

*Table D1: Comparison of the obtained T difference (ΔT = TLC-R - TAWS) when using the complete LC-R dataset and the LC-R test dataset.*

| LC-R | ΔT for complete dataset | ΔT for test dataset |
|---|---|---|
| LC-R01 | 0.01 ± 0.24°C | 0.01 ± 0.30°C |
| LC-R02 | 0.00 ± 0.23°C | 0.00 ± 0.30°C |
| LC-R04 | 0.00 ± 0.20°C | 0.00 ± 0.26°C |
| LC-R05 | 0.00 ± 0.18°C | 0.00 ± 0.23°C |
| Mean | 0.00 ± 0.22°C | 0.00 ± 0.28°C |

*For comparison the histograms and heatmaps of the temperature difference ($\Delta T = T_{LC-R} - T_{AWS}$) using the complete and test dataset only have been added below in Figure D1, D2 and D3. Only slight differences smaller than 0.1°C exist between the the ΔT for complete dataset and ΔT for test dataset. The diurnal and seasonal pattern is still corrected for (Figure D2). Also, effects of wind speed and radiation are effectively eliminated (Figure D3).*

[Figure]

*Figure D1: Same as Figure 9 for QC level 3. Comparison between ΔT for complete dataset (upper row) and ΔT for test dataset (lower row)*

[Figure]

*Figure D2: Same as Figure 10 for QC level 3. Comparison between ΔT for complete dataset (upper row) and ΔT for test dataset (lower row)*

[Figure]

*Figure D3: Same as Figure 11 for QC level 3. Comparison between ΔT for complete dataset (upper row) and ΔT for test dataset (lower row)'*

(Lines 822-850)

**Reply:** The differentiation between night and day is only for illustrative purposes. Only one RF model has been made for both day and night, the differentiation between day and night is however indirectly present in the model through the input variables "Hour", "Radiation" and "Radiation60".

*'The random forest prediction of temperature bias showed the best results. By splitting up the results for day (radiation > 0 W/m2) and night (radiation = 0 W/m2) (Figure 6), a smaller standard deviation of the bias during night time (0.25) compared to daytime (0.31) is obtained. This differentiation between night and day is only for illustrative purposes, only one RF was built for both day and night. The statistical details of the random forest model are further summarized in Table 9.'* (Lines 425-428)

12. Figure 4 and text: What is the authors explanation for the fact that largest biases are found in the middle of the radiation range and not at the top? Please elaborate a bit in the text.

**Reply:** We agree that this is indeed a bit unexpected, but our hypothesis is that it is related to the station design itself. Two effects are at play here: (1) the placement of the radiation sensor and (2) the placement of the temperature sensor. For (1), certainly for lower solar elevations (winter), the wind vane drops its

shadow at the radiation sensor for a short time of the day. For (2), which we might consider more important, the temperature sensor is more shaded around midday (highest radiation) by the body of the station (all seasons). This explains in our opinion the local minimum around noon for the summer months in the temperature bias. Note that our random forest model is capable of taking these station design flaws into account.

*'The shallow local minimum seen around noon during the summer months (Figure 4a) and the fact that the largest biases are found in the middle of the radiation range rather than at the top (Figure 4d) are probably related to the station design itself. Two effects are at play here: (1) the placement of the radiation sensor and (2) the placement of the temperature sensor. For (1), certainly for lower solar elevations (during winter), the wind vane drops its shadow at the radiation sensor for a short time of the day. For (2), which we might consider more important, the temperature sensor is more shaded around midday (highest radiation) by the body of the station (during all seasons).'* (Lines 372-377)

13. l. 573: You argue that your data set (since only one type of station) is free of certain effects that would require station-specific QC methods. Yet, in QC L3 you do a station-specific QC, no? Why couldn't you apply the QC to another type of weather station that measures the same variables? Please reword the sentence and/or explain better what you mean.

**Reply:** Thanks for this good remark. We did indeed correct for intrinsic differences between different weather station units during QC level 2. The sentence has been reworded.

*'Cornes et al. (2020) further highlights the need for station specific quality controls in order to remove the confounding effect of different instrument types. With the use of a unique station type, we aimed at minimizing such effects in our dataset. QC level 2 (Sect. 5.2) showed that these effects were indeed limited.'* (Lines 635-638)

14. l. 585: Who controls the stations and does the maintenance? Please explain in more detail.

**Reply:** The maintenance of the network is controlled by PhD students and the technical staff at the Division of Forest, Nature & Landscape of the KU Leuven and with support of the RMIB. Since most of the weather stations are installed in private gardens, our volunteers keep an eye out for generic problems as well (e.g. leaves in the rain gauge, …).

*'The maintenance of the network is controlled by PhD students and the technical staff at the Division of Forest, Nature & Landscape of the KU Leuven and with support of the RMIB. Since most of the weather stations are installed in private gardens, our volunteers keep an eye out for generic problems as well (e.g. leaves in the rain gauge, …).'* (Lines 206-208)

15. Figure 20: I suggest to include the months January – June in the top sub figure, simply to have the same x range as the other two. Please also ensure that the y range is the same in all sub figures.

**Reply:** Thanks for the nice suggestion. Changes have been made in the manuscript: see figure 20

16. Figures and tables in general:

(a) Maps should contain coordinate axis.

**Reply:** Thanks for noticing. The maps have been updates with a coordinate axis: see figure 2

(b) In all figures the subplot labels a, b, c, etc. are missing. This makes it quite difficult to understand some of them and needs to be resolved.

**Reply:** Thanks for noticing. All subplots have been labelled.

(c) In many figures where different sub-figures are present the axes ranges differ. I propose to have common axes ranges per figure. Alternatively, should at least give a note in the caption that the ranges differ.

**Reply:** Thanks for the comment. The axes ranges have been changed where possible. If not possible, a note was added to the caption.

(d) In all figures with coloured background (e.g. Figure 4): Please explain the colour coding and ensure that this is the same in all sub-figures, at best in all of those figures for comparison. Further, how did you group the data (binsize, hourly averages?, etc.)?

**Reply:** The colour coding represents the temperature difference or temperature bias. The range was indeed different in different subplots. This has now been adjusted; the temperature difference/bias ranges in all figures from -1.7°C (blue) to 1.7°C (red).

Plots of the temperature difference/bias as a function of hour of the day and month of the year were made by averaging the temperature bias/difference for every hour of every month. Plots of the temperature difference/bias as a function of radiation and wind speed were made by rounding all radiation values in bins of 80 W/m2 and all windspeed values in bins of 0.4 m/s. The temperature bias/difference was then averaged for each bin combination.

(e) The captions need more information regarding the data that is displayed, mainly regarding time period and data sets (this also applies to all tables).

**Reply:** Thanks for the comment. The heatmaps and histograms in section 3 and 4 are calculated from all LC-R measurements between the installation date of each LC-R and December 2021. All figures in section 5 are based on the data of all LC-X station between July 2019 and December 2021. All captions are updated with additional information on the time period and dataset.

(f) Number of figures: Several figures could be combined, e.g. Figures 9, 13, 15; Figures 10, 14, 16. Please check all figures and check (a) if they are all relevant and (b) if they could be combined. The amount of figures (20) and tables (12) is too much in my sense and should be reduced. For figures that basically display the same but only differ in the QC level, you could also write in the caption: "Same as figure X but for QC level Y".

**Reply:** We understand that 20 figures might seem like a lot, but each figure is relevant for this manuscript. Combining all figures result a huge block of figures and would deteriorate the structure of the manuscript. Currently the figures are ordered by quality control level. We have minimalized the captions of Figures 13; 14, 15, 16 and 17 since they display the same output only for a different QC level. If the reviewer would like us to move some figures to the appendix, we will happily do so.

17. The authors should somewhere better describe the actual data set on the repository (e.g. which variables, which resolution, what kind of meta data, etc.). This would be extremely helpful for other researchers to make use of the data. This section could, e.g., be put before the Conclusions.

**Reply:** Thanks for the comment. On the data repository, the dataset is accompanied with an extensive README file explaining all these details. To make use of the data more toilless, a few details have been added to data and code availability section in the manuscript

*'The dataset is accompanied by an extensive README file explaining the content of the dataset. The dataset includes a metadata file (01_Metadata.csv), the actual observations per 3 months/one quarter (LC_YYYYQX.csv) and the R scripts needed to build and apply the QC method (LC-R_QCL1-2-3.Rmd and LC-X_QCL1-2-3.Rmd). The metadata file contains info on the weather station's coordinates, altitude above sea level, installation height, LCZ class, dominant landcover, mean SVF and mean building height in a buffer of 10m around each weather station. Coordinates have rounded to 3 decimals for privacy reasons. The actual observations are aggregated to 10 minutes and include Relative humidity [%], Dew point temperature [°C], Number of 16 second observations in 10 minutes aggregate, Solar radiation [W/m2], Rain intensity [mm/h], Daily rain sum [mm], Wind direction [°], Wind speed [m/s], Date in YYYY-MM-DD, Year in YYYY, Month in MM, Day in DD, Hour in HH, Minute in MM, Weighted radiation during last 60 minutes [W/m2], Temperature at QCL0 [°C], Temperature at QCL1 [°C], Temperature at QCL2 [°C], Temperature at QCL3 [°C].'* (Lines 734-743)

18. Conclusions: The authors could highlight more clearly the high value of the quality-controlled data set. The large amount of stations in a relatively small city could, e.g., enable detailed comparison with other crowdsourced data where no meta data are available and where the station setup is unknown (e.g. Netatmo). As such, the stations could act as "gatekeepers" for other stations (Chapman et al. 2017). The data set could also be of use to modellers (micro-scale modelling).

**Reply:** Thanks for the good suggestion. We have added a list of possible applications to the manuscript.

*'6. Application potential of the quality controlled and corrected Leuven.cool dataset*

*A validation of the proposed QC method showed that it is able to reduce the mean temperature difference and standard deviation from 0.15 ± 0.56 °C to 0.00 ± 0.22 °C. The QC method can correct the temperature difference equally across different hours of the day and months of the year (Figure 16) as well and under different radiation and windspeed conditions (Figure 17).*

*The quality-controlled Leuven.cool dataset enables a detailed comparison with other crowdsourced datasets for which less or even no metadata is available. As such, the Leuven.cool stations can serve as gatekeepers for other crowdsourced observations. In the past this role has been limited to standard weather station network which mostly only have a limited number of observations available (Chapman et al., 2017).*

*The dense dataset further allows for microscale modelling of urban weather patterns, including the urban heat island (Chapman et al., 2017; de Vos et al., 2020; Feichtinger et al., 2020; Napoly et al., 2018). Since such high-quality dataset contain measurements with both high spatial and temporal resolution, they can easily be used to obtain spatially continuous temperature patterns across a region (e.g. Feichtinger et al., 2020; Napoly et al., 2018). Interpolation methods based on single pair stations or mobile transect methods are much less trustworthy (Napoly et al., 2018). Dense weather station networks can be used to investigate the inter- and intra LCZ variability within a city (Fenner et al., 2017; Verdonck et al., 2018). The dataset can further help investigate the relation between temperature*

*and human and ecosystem health (e.g. Aerts et al., 2022; Troeyer et al., 2020) and their effect on evolutionary processes (e.g. Brans et al., 2022).*

*The dataset can also help refine existing weather forecast models which are currently mostly based on official rural observations (Sgoff et al., 2022). Nipen et al. (2020) showed that the inclusion of citizen observations improves the accuracy of short-term temperature forecasts in regions where official stations are sparse. Mandement & Caumont (2020) used crowdsourced weather stations to improve the observation and prediction of near convection. After quality control and correction, also the wind (Chen et al., 2021) and precipitation measurements (de Vos et al., 2019)can be useful to improve detection and forecasting. Further, the Leuven.cool dataset could be a useful input in air pollution prediction models (e.g. IFD-model (Lefebvre et al., 2011))'* (Lines 673-701)

19. Conclusions: The authors conclude that the method is "easily transferable to other urban regions not having an official weather station". I do not understand how the authors come to this conclusion when such information is need in QC L1.1. Also, "easily transferable" only in that sense, that quite a lot of other data are needed for the QC of air-temperature data. That is certainly not the case for the vast amount of citizen weather stations that exist, from which data could be crowdsourced. The authors should state more clearly what is needed for the QC to be transferred to another regions and reword the section.

**Reply:** Thanks for the valid comment. This is (unintentionally) indeed misleading. Transferring the method to other networks would indeed require the recalibration of the QC parameters. These lines are rephrased in the manuscript. In the conclusion we added a complete overview of what is needed for the QC to be transferred to another regions.

*'Afterwards the method can be applied independently of the official network that was used in the development phase. Transferring the method to other networks or regions would require the recalibration of the QC parameters. Specifically, for QC L1.1 some indication of climate thresholds is needed. QC L1.3 and QC L2 require a dense weather station network, the QC method is thus less suitable for single or few stations. For QC L3 quite some other data (e.g. radiation, windspeed, ...) are needed. The random forest model is however easily adaptable to the parameters that are available.'* (Lines 714-719)

20. Meta data in the actual data set: I noticed that latitude and longitude values are given with only three digits. I highly recommend to give at least four digits for better precision and to subsequently allow more detailed spatial analyses on the micro scale.

**Reply:** Thanks for the suggestion. Due to privacy issues and as agreed in the informed consent with our volunteers, we are however not allowed to give more than 3 digits. Most of our weather stations are installed in private gardens. We assured our volunteers that their individual gardens would not be detectable, and that the data of each weather station would be shown on a resolution of 'street level'. When using 4 digits, the individual gardens are easily detectable.

21. Do the authors have any idea about the quality of the other variables that are measured and used in the quality control? It could make sense to have relatively simple range and persistence tests (as

you developed) implemented for all variables before the detailed QC of air temperature. Please comment on that in the text somewhere.

**Reply:** We have made a qualitative assessment of the data quality by making scatterplots of these variables for the LC-R stations, compared to the AWS data. Overall, the measured parameters are within the accuracy given by the manufacturer.

There are, however, deviations in the wind and radiation measurement that are attributable to the small location differences (in the order of meters) between the LC-R stations and the official sensors. For wind, we compared the LC-R data with professional 2m wind measurements, but this height is in fact not the standard measurement height for wind since too many ground effects are still into play at this height. For radiation, we already mentioned the design flaw regarding the wind vane dropping a shadow, but also high nearby trees can influence the measurements for low solar elevations (in the case of LC-R01 and LC-R02).

'*A qualitative assessment of the data quality of these variables in included in Appendix C.*' (Line 204)

'*Appendix C. Quality control other Leuven.cool variables*

*A qualitative assessment of the data quality was performed by making scatterplots of these variables for the LC-R stations compared to the AWS stations (Figure C1).*

*Overall, the measured parameters are within the accuracy given by the manufacturer. There are, however, deviations in the wind and radiation measurement that are attributable to the small location differences (in the order of meters) between the LC-R stations and the official sensors. For wind, we compared the LC-R data with professional 2m wind measurements, but this height is in fact not the standard measurement height for wind since too many ground effects are still into play at this height. For radiation, we already mentioned the design flaw regarding the wind vane dropping a shadow, but also high nearby trees can influence the measurements for low solar elevations (in the case of LC-R01 and LC-R02).*

[Figure]

[Figure]

[Figure]

*Figure C1: Scatterplots of LC-R versus AWS dew point temperature, humidity, radiation and wind speed for each reference station LC-R01, LC-R02, LC-R04, LC-R05. The identity line is shown in black. The colour scale indicates the density of observations, yellow indicating the highest, purple the lowest. The scatterplots include all measurements between the installation data of each LC-R and December 2021. For each variable the same ranges of x- and y-axis are used.'*

(Lines 798-820)

**Technical corrections:**

- Compound adjective (e.g. "air temperature" but "air-temperature data"). Please check the whole manuscript carefully

**Reply:** Thanks for the correction. This has been adjusted in the manuscript

- l. 32: consider "… that both cities and their citizens are …"

**Reply:** Thanks for the suggestion. This has been adjusted in the manuscript.

- l. 50: "… in which air temperature is continuously modelled …"

**Reply:** Thanks for the correction. This has been adjusted in the manuscript

- l. 101: consider "Thus, there is a need for …"

**Reply:** Thanks for the suggestion. This has been adjusted in the manuscript.

- l. 104: "Here we report on a …"

**Reply:** Thanks for the suggestion. This has been adjusted in the manuscript.

- l. 169: "… is limited due to the …"

**Reply:** Thanks for the suggestion. This has been adjusted in the manuscript.

- l. 178: "direct effect contact" ? Please reword.

**Reply:** Thanks for the correction. This has been adjusted in the manuscript.

*'Weather stations located on public impervious surfaces were installed on available light poles using specially designed L-structures to avoid direct contact with the pole.'* (Lines190-192)

- l. 182: "aggregated" – Do you mean temporally averaged?

**Reply:** Thanks for the comments. We indeed mean temporally averaged. This has been adjusted in the manuscript.

*'The raw 16 seconds measurements are aggregated (temporally averaged) to 10 minutes observations.'* (Lines 195-196)

- l. 194 (and other instances in the text, Table A1): There should be a space between the number and the unit, except for geographical degrees; cf. chapter 5.4.3 in https://www.bipm.org/documents/20126/41483022/SIBrochure-9.pdf/fcf090b2-04e6-88cc-1149-c3e029ad8232).

**Reply:** Thanks for the correction. This has been adjusted in the manuscript.

- l. 266-267: Please check sentence and reword.

**Reply:** Thanks for the comment. This has been adjusted in the manuscript.

*'Whenever the difference between sequential observations cannot be calculated an observation gets a flag equal to -1.'* (Lines 287-288)

- l. 285: "non-negligible" – How did you determine that? What is the basis for the statement?

**Reply:** A temperature difference of 0.2°C was found. Such a difference cannot be explained by the resolution of the temperature sensor (0.1°C) and is thus non negligible.

*'A temperature difference of 0.2°C was found, which cannot be explained by the resolution of the temperature sensor (0.1°C).'* (Lines 306-307)

- l. 298: "i.e." is instead of "e.g."
**Reply:** Thanks for the correction. This has been adjusted in the manuscript.

- l. 300: Could be helpful to say "… across all ten selected episodes is considered …"
**Reply:** Thanks for the suggestion but that would be incorrect. First, the query is performed for every 6-months period, resulting in 10 episodes times the number of 6-month period assessed. The analysis is currently done from 2019S2 until 2021S2, resulting in 5 6-month periods x 10 episodes = 50 episodes. Afterwards temperature is regressed versus altitude and only the episodes with a high correlation between temperature and altitude ($> 0.7$) are kept.

- Table 7: What is actually shown here? What kind of correlation coefficient? Per definition, a correlation coefficient has no units. Please remove them and specify more clearly in the caption what is shown.
**Reply:** Thanks for the valid remark. Table 7 shows the Pearson correlation coefficient between the observed temperature bias and other variables measured by the weather station. The units are indeed incorrect, they have been removed.

*Table 7: Pearson correlation matrix of temperature bias with other meteorological variables measured by the low-cost station.*

| Temperature | Dew point temperature | Humidity | Radiation | Radiation60 | Wind speed |
|---|---|---|---|---|---|
| 0.41 | 0.18 | -0.48 | 0.49 | 0.56 | -0.01 |

(Lines 391-393)

- Figure 8: What is the black line? Please explain in the caption.
**Reply:** the black line indicates the identity line or line of equality. This has been further explained in the caption

*'Figure 8: Scatterplots of LC-R versus AWS temperature for each reference station LC-R01 (a), LC-R02 (b), LC-R04 (c), LC-R05 (d) at QC level 0. Observations defined as range outliers are symbolized by a red circle, temporal outliers by a red square. The identity line is shown in black. The colour scale indicates the density of observations, white indicating the highest, black the lowest. The scatterplots include all measurements between the installation data of each LC-R and December 2021.'* (Lines 475-479)

- l. 447 & 448: Where is the information regarding the summer day shown? Please specify or reword.

**Reply:** Thanks for the valid comment. The information for one summer day is indeed not shown individually. From figures 10 and 11 we can deduce that a higher difference is obtained during the summer months (Figure 10) under low cloud and low windspeed conditions (Figure 11).

*'As expected, the temperature difference between the LC-R and AWS stations is not constant and is correlated with other variables. A higher difference is obtained during the summer months (Figure 10) under low cloud and low windspeed conditions (Figure 11).'* (Lines 483-485)

**References:**

Aerts, R., Vanlessen, N., Dujardin, S., Nemery, B., Van Nieuwenhuyse, A., Bauwelinck, M., Casas, L., Demoury, C., Plusquin, M., & Nawrot, T. S. (2022). Residential green space and mental health-related prescription medication sales: An ecological study in Belgium. *Environmental Research*, *211*. https://doi.org/10.1016/j.envres.2022.113056

Ahrens, C. D. (2009). *Meteorology Today: An Introduction to Weather, Climate, and the Environment* (Ninth Edit). Brooks/Cole.

Båserud, L., Lussana, C., Nipen, T. N., Seierstad, I. A., Oram, L., & Aspelien, T. (2020). TITAN automatic spatial quality control of meteorological in-situ observations. *Advances in Science and Research*, *17*, 153–163. https://doi.org/10.5194/asr-17-153-2020

Bell, S., Cornford, D., & Bastin, L. (2015). How good are citizen weather stations? Addressing a biased opinion. *Weather*, *70*(3), 75–84. https://doi.org/10.1002/wea.2316

Brans, K. I., Tüzün, N., Sentis, A., De Meester, L., & Stoks, R. (2022). Cryptic eco-evolutionary feedback in the city: Urban evolution of prey dampens the effect of urban evolution of the predator. *Journal of Animal Ecology*, *91*(3), 514–526. https://doi.org/10.1111/1365-2656.13601

Chapman, L., Bell, C., & Bell, S. (2017). Can the crowdsourcing data paradigm take atmospheric science to a new level? A case study of the urban heat island of London quantified using Netatmo weather stations. *International Journal of Climatology*, *37*(9), 3597–3605. https://doi.org/10.1002/joc.4940

Chen, Y., Weng, Q., Tang, L., Liu, Q., Zhang, X., & Bilal, M. (2021). Automatic mapping of urban green spaces using a geospatial neural network. *GIScience & Remote Sensing*, *00*(00), 1–19. https://doi.org/10.1080/15481603.2021.1933367

Cornes, R. C., Dirksen, M., & Sluiter, R. (2020). Correcting citizen-science air temperature measurements across the Netherlands for short wave radiation bias. *Meteorological Applications*, *27*(1), 1–16. https://doi.org/10.1002/met.1814

de Vos, Droste, A. M., Zander, M. J., Overeem, A., Leijnse, H., Heusinkveld, B. G., Steeneveld, G. J., & Uijlenhoet, R. (2020). Hydrometeorological monitoring using opportunistic sensing networks in the Amsterdam metropolitan area. *Bulletin of the American Meteorological Society*, *101*(2), E167–E185. https://doi.org/10.1175/BAMS-D-19-0091.1

de Vos, L., Leijnse, H., Overeem, A., & Uijlenhoet, R. (2019). Quality Control for Crowdsourced Personal Weather Stations to Enable Operational Rainfall Monitoring. *Geophysical Research Letters*, *46*(15), 8820–8829. https://doi.org/10.1029/2019GL083731

Feichtinger, M., Wit, R. De, Goldenits, G., Kolejka, T., & Hollósi, B. (2020). Urban Climate Case-study of neighborhood-scale summertime urban air temperature for the City of Vienna using crowd-sourced data. *Urban Climate*, *32*(August 2019), 1–12. https://doi.org/10.1016/j.uclim.2020.100597

Fenner, D., Meier, F., Bechtel, B., Otto, M., & Scherer, D. (2017). Intra and inter "local climate zone" variability of air temperature as observed by crowdsourced citizen weather stations in Berlin, Germany. *Meteorologische Zeitschrift*, *26*(5), 525–547. https://doi.org/10.1127/metz/2017/0861

Jenkins, G. (2014). A comparison between two types of widely used weather stations. *Weather*, *69*(4), 105–110. https://doi.org/10.1002/wea.2158

Lefebvre, W., Vercauteren, J., Schrooten, L., Janssen, S., Degraeuwe, B., Maenhaut, W., de Vlieger, I., Vankerkom, J., Cosemans, G., Mensink, C., Veldeman, N., Deutsch, F., Van Looy, S., Peelaerts, W., & Lefebre, F. (2011). Validation of the MIMOSA-AURORA-IFDM model chain for policy support: Modeling concentrations of elemental carbon in Flanders. *Atmospheric Environment*, *45*(37), 6705–6713. https://doi.org/10.1016/j.atmosenv.2011.08.033

Longman, R. J., Giambelluca, T. W., Nullet, M. A., Frazier, A. G., Kodama, K., Crausbay, S. D., Krushelnycky, P. D., Cordell, S., Clark, M. P., Newman, A. J., & Arnold, J. R. (2018). Compilation of climate data from heterogeneous networks across the Hawaiian Islands. *Scientific Data*, *5*(1), 180012. https://doi.org/10.1038/sdata.2018.12

Mandement, M., & Caumont, O. (2020). Contribution of personal weather stations to the observation of deep-convection features near the ground. *Natural Hazards and Earth System Sciences*, *20*(1), 299–322. https://doi.org/10.5194/nhess-20-299-2020

Meier, F., Fenner, D., Grassmann, T., Otto, M., & Scherer, D. (2017). Crowdsourcing air temperature from citizen weather stations for urban climate research. *Urban Climate*, *19*(February), 170–191. https://doi.org/10.1016/j.uclim.2017.01.006

Muller, C. L., Chapman, L., Johnston, S., Kidd, C., Illingworth, S., Foody, G., Overeem, A., & Leigh, R. R. (2015). Crowdsourcing for climate and atmospheric sciences: Current status and future potential. *International Journal of Climatology*, *35*(11), 3185–3203. https://doi.org/10.1002/joc.4210

Napoly, A., Grassmann, T., Meier, F., & Fenner, D. (2018). Development and Application of a Statistically-Based Quality Control for Crowdsourced Air Temperature Data. *Frontiers in Earth Science*, *6*(August), 1–16. https://doi.org/10.3389/feart.2018.00118

Nipen, T. N., Seierstad, I. A., Lussana, C., Kristiansen, J., & Hov, Ø. (2020). Adopting citizen observations in operational weather prediction. *Bulletin of the American Meteorological Society*, *101*(1), E43–E57. https://doi.org/10.1175/BAMS-D-18-0237.1

Sgoff, C., Acevedo, W., Paschalidi, Z., Ulbrich, S., Bauernschubert, E., Kratzsch, T., & Potthast, R. (2022). Assimilation of crowd-sourced surface observations over Germany in a regional weather prediction system. *Quarterly Journal of the Royal Meteorological Society*, *148*(745), 1752–1767. https://doi.org/10.1002/qj.4276

Stewart, I. D. (2011). A systematic review and scientific critique of methodology in modern urban heat island literature. *International Journal of Climatology*, *31*, 200–217. https://doi.org/10.1002/joc.2141

Stewart, I. D., & Oke, T. R. (2012). Local climate zones for urban temperature studies. *Bulletin of the American Meteorological Society*, *93*(12), 1879–1900. https://doi.org/10.1175/BAMS-D-11-00019.1

Troeyer, K. De, Bauwelinck, M., Aerts, R., Profer, D., Berckmans, J., Delcloo, A., Hamdi, R., Schaeybroeck, B. Van, Hooyberghs, H., Lauwaet, D., Demoury, C., & Nieuwenhuyse, A. Van. (2020). Heat related mortality in the two largest Belgian urban areas : A time series analysis. *Environmental Research*, *188*(February), 109848. https://doi.org/10.1016/j.envres.2020.109848

Venter, Z. S., Chakraborty, T., & Lee, X. (2021). Crowdsourced air temperatures contrast satellite measures of the urban heat island and its mechanisms. *Science Advances*, *7*(22), 1–9. https://doi.org/https://dx.doi.org/10.1126/sciadv.abb9569

Verdonck, M. L., Demuzere, M., Hooyberghs, H., Beck, C., Cyrys, J., Schneider, A., Dewulf, R., & Van Coillie, F. (2018). The potential of local climate zones maps as a heat stress assessment tool, supported by simulated air temperature data. *Landscape and Urban Planning*, *178*(July 2017), 183–197. https://doi.org/10.1016/j.landurbplan.2018.06.004

Verdonck, M. L., Okujeni, A., van der Linden, S., Demuzere, M., De Wulf, R., & Van Coillie, F. (2017). Influence of neighbourhood information on 'Local Climate Zone' mapping in heterogeneous cities. *International Journal of Applied Earth Observation and Geoinformation*, *62*(September 2016), 102–113. https://doi.org/10.1016/j.jag.2017.05.017

Zheng, F., Tao, R., Maier, H. R., See, L., Savic, D., Zhang, T., Chen, Q., Assumpção, T. H., Yang, P.,

Heidari, B., Rieckermann, J., Minsker, B., Bi, W., Cai, X., Solomatine, D., & Popescu, I. (2018). Crowdsourcing Methods for Data Collection in Geophysics: State of the Art, Issues, and Future Directions. *Reviews of Geophysics*, *56*(4), 698–740. https://doi.org/10.1029/2018RG000616

---

## Author Comment (AC2)

**Authors' response to RC2 - manuscript ESSD-2022-113**

Responses to Reviewer comments: The text in blue represents the authors' reply. Text in italic represents text that was changed in the manuscript, with line numbers referring to those in the revised manuscript. We numbered the comments for easier referral.

**General comments:**

Small typographic errors (missing punctuation or erroneous capitalization) that proofreaders should catch.

Data accessible, clearly described, easy to work with. This reviewer prefers .csv to .tab but that seems a matter of individual preference.

I understand why authors might prefer CC-BY-NC but, due to proper ESSD cautions, have editors granted permission for -NC?

Overall: good well-organized description of QC levels and good summary of 'final' QC outcome, 0.15 + 0.56 down to 0.00 + 0.22 for temperature. In describing and summarizing these efforts and QC levels, the authors seem to have left out or passed over several issues.

**Reply:** We thank the reviewer for the thorough review. We have considered all your comments and revised the manuscript accordingly. We have proofread the manuscript to correct for additional typographic errors. We also like to mention that the data depository automatically transforms .csv files to .tab files. There is, however, a possibility to download the files as .csv. Regarding the CC-BY-NC license, this was not raised as a problem during the initial submission phase. If the editors would not grant permission for –NC, we can change this. We tried to rephrase and further explain the QC methodology in order to solve the issues mentioned below. Please see our point-by-point responses on the specific comments below.

**Specific comments:**

1. Daytime vs nighttime much used but little described. Based on daily hours sunlight, calculated by day, month or season? Or calculated based on measured surface radiation values where night = low light = < some minimum value of W/m2?

**Reply:** Thanks for the valid comment. We agree that the differentiation between night- and daytime was not clearly described. Currently it was only mentioned in the caption of figure 3. The distinction between day- and nighttime is based on the radiation measurements of each station; daytime is defined as $LC\_RAD > 0$ W/m2, nighttime is defined as $LC\_RAD = 0$ W/m2. This has been further clarified in the manuscript. We should further state that the differentiation between night and day is only for illustrative purposes. Only one RF model has been made for both day and night, the differentiation between day and night is however indirectly present in the model through the input variables "Hour", "Radiation" and "Radiation60".

*'The overall temperature bias (i.e., all LC-R stations together) between the LC-R and the AWS data has a mean value of 0.10°C and a standard deviation of 0.55 °C (Figure 3). By splitting up the temperature bias for day (radiation > 0 W/m2) and night (radiation = 0 W/m2), a positive mean temperature bias during daytime (0.32 °C) and a negative mean temperature bias during night time (-0.10 °C) is obtained.'* (Lines 348-351)

*'The random forest prediction of temperature bias showed the best results. By splitting up the results for day (radiation > 0 W/m2) and night (radiation = 0 W/m2) (Figure 6), a smaller standard deviation of the bias during night time (0.25) compared to daytime (0.31) is obtained. This differentiation between night and day is only for illustrative purposes, only one RF was built for both day and night. The statistical details of the random forest model are further summarized in Table 9.'* (Lines 425-428)

2. Other parameters (RH, precip, wind) not corrected. Authors describe, correctly, the use of non-T parameters in various QC functions but otherwise give no hints about suitability, difficulty or desirability of addressing RH, precip, etc. Do readers await a future paper? Do we assume QC of T proved easier than other parameters? Authors give us no hint what to assume!

**Reply:** We understand the question of the reviewer. Before using the non-T parameters in various steps of the QC, we have made a qualitative assessment of the data quality by making scatterplots of the non-T parameters for the LC-R stations, compared to the AWS data. Overall, the measured parameters are within the accuracy given by the manufacturer. These scatterplots have been added as an Appendix (Appendix C) to the manuscript.

There are, however, deviations in the wind and radiation measurement that are attributable to the small location differences (in the order of meters) between the LC-R stations and the official sensors. For wind, we compared the LC-R data with professional 2 m wind measurements, but this height is in fact not the standard measurement height for wind since too many ground effects are still into play at this height. For radiation, we already mentioned the design flaw regarding the wind vane dropping a shadow, but also high nearby trees can influence the measurements for low solar elevations (in the case of LC-R01 and LC-R02).

The main focus of the authors research is related to urban temperatures and the UHI effect. Currently no QC for the other parameters are planned.

*'A qualitative assessment of the data quality of these variables in included in Appendix C.'* (Line 204)

*'**Appendix C. Quality control other Leuven.cool variables***

*A qualitative assessment of the data quality was performed by making scatterplots of these variables for the LC-R stations compared to the AWS stations (Figure C1).*

*Overall, the measured parameters are within the accuracy given by the manufacturer. There are, however, deviations in the wind and radiation measurement that are attributable to the small location differences (in the order of meters) between the LC-R stations and the official sensors. For wind, we compared the LC-R data with professional 2m wind measurements, but this height is in fact not the standard measurement height for wind since too many ground effects are still into play at this height. For radiation, we already mentioned the design flaw regarding the wind vane dropping a shadow, but also high nearby trees can influence the measurements for low solar elevations (in the case of LC-R01 and LC-R02).*

[Figure]

[Figure]

[Figure]

*Figure C1: Scatterplots of LC-R versus AWS dew point temperature, humidity, radiation and wind speed for each reference station LC-R01 , LC-R02, LC-R04, LC-R05. The identity line is shown in black. The colour scale indicates the density of observations, yellow indicating the highest, purple the lowest. The scatterplots include all measurements between the installation data of each LC-R and December 2021. For each variable the same ranges of x- and y-axis are used.'*

(Lines 798-820)

3. Authors should, technically, express offsets and SD in K rather than C. Not often done, I accept.

**Reply:** Thanks for the comment. We have followed related literature on air temperature quality control and correction methods, which choose to express temperature offsets and SD in °C. A small clarification is however added to the manuscript.

*'The outdoor sensor array measures temperature (°C, add 273.15 for K), humidity (%), precipitation (mm), wind speed (m/s), wind direction (°), solar radiation (W/m2), and UV (-) every 16 seconds.'* (Lines 145-147)

4. Authors have lumped 2 m data with 3-4 m data? Early mention but then no subsequent treatment. One suspects both sensor elevation and surrounding (mostly impervious) surfaces would have a large effect but never mentioned? Shadowing by anemometer mentioned occasionally but that occurs independently of short vs high poles and regardless of underlying surface?

**Reply:** The installation height of the weather station is indeed not directly added as a parameter in the RF model (QC L3) in order to correct the T observations. This was not possible since all the LC-R stations were installed at a height of 2 m. We do however believe that the installation height is indirectly present in the RF model through the radiation and wind speed parameters. Furter, the altitude (above sea level) of the weather station will more clearly influence the measured temperature and is added as a parameter in the RF model.

We do acknowledge that nocturnal temperature inversions can appear under clear calm nights (under low radiation and low windspeed conditions). During such a nocturnal inversion we normally see an increase in temperature with height due to radiational cooling of the earth's surface (Ahrens, 2009). Such inversions however often occur at ground level, and thus below 2 m (Ahrens, 2009).

To quantify the temperature difference between 2 m and 3 m, a small test was performed using multiple AWS stations having T observations at both 2 m and 10 m. By interpolating the temperature at 2 m and

10 m, a temperature at 3 m is obtained. Next the average temperature difference and standard deviation between 3 m and 2 m was calculated for the whole of 2021 and for stable conditions ($T_{10\,m} > T_{2\,m}$) in which we expect the highest difference between 2 m and 3 m. For the whole of 2021 the mean difference equals $0 \pm 0.09$ °C, for the stable conditions we obtain a small difference of $0.08 \pm 0.09$ °C. We should thus state that stations installed at 3 m – 4 m can have an additional offset up to $0.08 \pm 0.09$ °C under stable conditions. This offset is much smaller than the night time temperature difference caused by the UHI effect (Chapman et al., 2017; Stewart, 2011; Venter et al., 2021).

We further stress that stations have only been installed at a height of 3 m to 4 m where it was needed due safety reasons.

**Reply:** The environment and surrounding surfaces of each sensor will indeed further influence the temperature measurements (Logan et al., 2020; J. Wang et al., 2022; Q. Wang et al., 2022; Ziter et al., 2019). The weather stations were however installed following a strict protocol, at least 1 m from interfering objects, making sure no direct effects of the environment incorrectly influence the T observations. On the other hand, the goal of this weather station network is to measure the micro-climate in Leuven, namely the difference between impervious and greener locations within and outside the city center. In other words, our stations are installed at both impervious and green locations within and outside the city center in order to measure (and in the future also explain) these temperature differences.

**Reply:** The shadowing of the anemometer is intrinsic to the station design and hence the same for every station, also the LC-R stations. As a consequence, the effect is included in our RF model.

5. UHI: This reader missed an overall assessment of data as QC'd here to address UHI. Do we need 0.1K? 1.0K? Have the authors come close with these corrections. In view of distance of reference AWS from Leuven and, for two RMIB stations at least, distance from true urban settings, do authors feel they have a network now suitable for addressing UHI. Not clear, perhaps needs/deserves further clarification. Given sensor height differences already mentioned plus apparent absence (or, avoidance) of the most urban land use categories, can users really trust this data for further UHI work? UHI mentioned frequently in introduction but not at all in conclusion paragraph. I agree with summary sentences but, if they could not in the end address UHI - which, in many indices, includes high net radiation, low RH and low wind - should they have given so much attention in introduction

**Reply:** We thank the reviewer for the valid comment. The resolution on which the UHI should be investigated is highly dependent on the application. For human thermal comfort studies, a resolution of 1°C would however be sufficient (Epstein & Moran, 2006; Georgi & Zafiriadis, 2006). Before the QC method, a mean temperature bias up to $0.15 \pm 0.56$ °C is noted. After correction the mean bias has been diminished to $0.00 \pm 0.28$ °C. Numerous studies have however showed that the UHI effect causes nighttime temperature differences up 6 to 9 °C during clear nights. These thresholds are much higher than mean bias obtained after correction, the quality controlled Leuven.cool dataset is thus suitable to study the UHI (Chapman et al., 2017; Stewart, 2011; Venter et al., 2021).

We further showed that the QC method can correct the temperature bias equally across different hours of the day and months of the year (Figure 16) as well as under different radiation and windspeed conditions (Figure 17). Most UHI indicators are based on either maximum daytime or minimum nighttime temperatures during extreme heat events (high radiation + low windspeed), while these time

periods show the highest temperature bias (Figure 10 and Figure 11). Before correction these indicators would have either a positive (daytime) or negative (nighttime) bias, the ideal meteorological conditions would result in a positive bias. This QC method ensures that day -and nighttime effects as well as effects due to certain meteorological conditions have been corrected for. As a result, we can trust UHI indicators calculated for specific day- and night time events.

*'Numerous studies have shown that the UHI effect causes night time temperature differences up 6 to 9 °C during clear nights (Chapman et al., 2017; Venter et al., 2021; Stewart, 2011; Napoly et al., 2018; Feichtinger et al., 2020). These thresholds are much higher than mean bias obtained after correction; the dense quality controlled Leuven.cool dataset thus allows for microscale modelling of urban weather patterns, including the urban heat island (Chapman et al., 2017; de Vos et al., 2020; Napoly et al., 2018; Feichtinger et al., 2020).'* (Lines 683-687)

**Reply:** Regarding the distance between the AWS and the LC-X in Leuven, we don't fully understand the question.

We agree that there is quite a difference between the AWS and the LC-X in Leuven, and that due to this difference the AWS stations cannot serve as a direct reference for the LC-X in Leuven. We should however state that the RF model is based on the T difference between the official AWS and the LC-R (installed next to the AWS), the RF model does not take into account the standalone measurements of the AWS itself. We further installed reference stations (LC-R) at three different locations (Uccle, Diepenbeek, Humain) to make the model more robust against spatial differences.

For addressing the UHI, the authors will not take AWS measurements into account but exclusively focus on the LC-X station in and outside Leuven.

**Reply:** Regarding the apparent absence of most urban land use classes, we are a bit confused. Our weather station network mainly focusses on the urban LCZ classes, natural LCZ classes are not as well represented.

The selection procedure for suitable locations started as a stratified sampling based on the concept of LCZs (by looking at building height, building density and vegetation cover). The goal was to divide the stations across the LCZ classes so that they would represent the spatial coverage of each LCZ. We should state that the network was implemented with the intention of gaining knowledge on the mitigating effect of green and blue infrastructures within urban settings. The initial study area thus mainly included the city centre of Leuven. As a result, the network has a clear bias towards urban classes.

Figure 1 shows the distribution of the weather stations across the LCZ classes (left panel) and proportion of LCZ classes across study area (right panel). Due to the complex urban settings in which the network is deployed, practical limitations apply to the eligible locations for installation. We rely on volunteering citizens, private companies and government institutions giving permission to install a weather station on their property. Due to a technical limitation of the weather station, it cannot be installed in natural environments without a LAN connection within 50 to 100 meters of the weather station.

[Figure]

*Figure 1: Distribution of weather stations across LCZ classes (left panel) and proportion of LCZ classes across study area (right panel).*

**Reply:** We understand the reviewer's comment on the lacking of the UHI in the conclusion. We have added a section with additional applications of this dataset, including the investigation of the UHI effect. These applications were summarized in the conclusion.

*'6. Application potential of the quality controlled and corrected Leuven.cool dataset*

*A validation of the proposed QC method showed that it can reduce the mean temperature difference and standard deviation from 0.15 ± 0.56 °C to 0.00 ± 0.28 °C. The QC method can correct the temperature difference equally across different hours of the day and months of the year (Figure 16) as well and under different radiation and windspeed conditions (Figure 17).*

*The quality-controlled Leuven.cool dataset enables a detailed comparison with other crowdsourced datasets for which less or even no metadata is available. As such, the Leuven.cool stations can serve as gatekeepers for other crowdsourced observations. In the past this role has been limited to standard weather station network which mostly only have a limited number of observations available (Chapman et al., 2017).*

*Numerous studies have shown that the UHI effect causes night time temperature differences up 6 to 9 °C during clear nights (Chapman et al., 2017; Feichtinger et al., 2020; Napoly et al., 2018; Stewart, 2011; Venter et al., 2021). These thresholds are much higher than mean bias obtained after correction. The dense quality controlled Leuven.cool dataset thus allows for microscale modelling of urban weather patterns, including the urban heat island (Chapman et al., 2017; de Vos et al., 2020; Feichtinger et al., 2020; Napoly et al., 2018). Since such high-quality datasets contain measurements with both high spatial and temporal resolution, they can easily be used to obtain spatially continuous temperature patterns across a region (e.g. Feichtinger et al., 2020; Napoly et al., 2018). Interpolation methods based on single pair stations or mobile transect methods are much less trustworthy (Napoly et al., 2018). Dense weather station networks can be used to investigate the inter- and intra LCZ variability within a city (Fenner et al., 2017; Verdonck et al., 2018). The dataset can further help investigate the relation between temperature and human and ecosystem health (e.g. Aerts et al., 2022; Troeyer et al., 2020) and their effect on evolutionary processes (e.g. Brans et al., 2022).*

*The dataset can also help refine existing weather forecast models which are currently mostly based on official rural observations (Sgoff et al., 2022). Nipen et al. (2020) showed that the inclusion of citizen observations improves the accuracy of short-term temperature forecasts in regions where official stations are sparse. Mandement & Caumont (2020) used crowdsourced weather stations to improve the observation and prediction of near convection. After quality control and correction, also the wind (Chen*

*et al., 2021) and precipitation measurements (de Vos et al., 2019) can be useful to improve detection and forecasting. Further, the Leuven.cool dataset could be a useful input in air pollution prediction models (e.g. IFD-model (Lefebvre et al., 2011)).'* (Lines 673-701)

*'The quality-controlled Leuven.cool dataset enables a detailed comparison with other crowdsourced datasets for which less or even no metadata is available. The dense dataset further allows for microscale modelling of urban weather patterns, such as the urban heat island, and can help identify the relation between temperature and human and ecosystem and their effect on evolutionary processes. Lastly the dataset could be used to refine existing forecast models which are currently mostly based on official rural observations.'* (Lines 723-727)

**Reply:** We agree with the reviewer's comment that UHI effect is mostly investigated under high radiation, low relative humidity and low wind- speed conditions. As a consequence, we must make sure that the QC method is also valid under these meteorological conditions. If this is not the case, the dataset cannot be used to study the UHI with high accuracy.

We should however state that QC also works for high radiation and low windspeed conditions. The QC method can correct the temperature bias equally across different hours of the day and months of the year (Figure 16) as well as under different radiation and windspeed conditions (Figure 17). Most UHI indicators are based on either maximum daytime or minimum nighttime temperatures during extreme heat events (high radiation + low windspeed), while these time periods show the highest temperature bias (Figure 10 and Figure 11). Before correction these indicators could have either a positive (daytime) or negative (nighttime) bias, the ideal meteorological conditions would result in a positive bias. This QC method ensures that day -and nighttime effects as well as effects due to certain meteorological conditions have been corrected for.

**References:**

Aerts, R., Vanlessen, N., Dujardin, S., Nemery, B., Van Nieuwenhuyse, A., Bauwelinck, M., Casas, L., Demoury, C., Plusquin, M., & Nawrot, T. S. (2022). Residential green space and mental health-related prescription medication sales: An ecological study in Belgium. *Environmental Research*, *211*. https://doi.org/10.1016/j.envres.2022.113056

Ahrens, C. D. (2009). *Meteorology Today: An Introduction to Weather, Climate, and the Environment* (Ninth Edit). Brooks/Cole.

Brans, K. I., Tüzün, N., Sentis, A., De Meester, L., & Stoks, R. (2022). Cryptic eco-evolutionary feedback in the city: Urban evolution of prey dampens the effect of urban evolution of the predator. *Journal of Animal Ecology*, *91*(3), 514–526. https://doi.org/10.1111/1365-2656.13601

Chapman, L., Bell, C., & Bell, S. (2017). Can the crowdsourcing data paradigm take atmospheric science to a new level? A case study of the urban heat island of London quantified using Netatmo weather stations. *International Journal of Climatology*, *37*(9), 3597–3605. https://doi.org/10.1002/joc.4940

Chen, Y., Weng, Q., Tang, L., Liu, Q., Zhang, X., & Bilal, M. (2021). Automatic mapping of urban green spaces using a geospatial neural network. *GIScience & Remote Sensing*, *00*(00), 1–19. https://doi.org/10.1080/15481603.2021.1933367

de Vos, Droste, A. M., Zander, M. J., Overeem, A., Leijnse, H., Heusinkveld, B. G., Steeneveld, G. J., & Uijlenhoet, R. (2020). Hydrometeorological monitoring using opportunistic sensing networks in the Amsterdam metropolitan area. *Bulletin of the American Meteorological Society*, *101*(2), E167–E185. https://doi.org/10.1175/BAMS-D-19-0091.1

de Vos, L., Leijnse, H., Overeem, A., & Uijlenhoet, R. (2019). Quality Control for Crowdsourced Personal Weather Stations to Enable Operational Rainfall Monitoring. *Geophysical Research Letters*, *46*(15), 8820–8829. https://doi.org/10.1029/2019GL083731

Epstein, Y., & Moran, D. S. (2006). Thermal comfort and the heat stress indices. In *Industrial Health*. https://doi.org/10.2486/indhealth.44.388

Feichtinger, M., Wit, R. De, Goldenits, G., Kolejka, T., & Hollósi, B. (2020). Urban Climate Case-study of neighborhood-scale summertime urban air temperature for the City of Vienna using crowd-sourced data. *Urban Climate*, *32*(August 2019), 1–12. https://doi.org/10.1016/j.uclim.2020.100597

Fenner, D., Meier, F., Bechtel, B., Otto, M., & Scherer, D. (2017). Intra and inter "local climate zone" variability of air temperature as observed by crowdsourced citizen weather stations in Berlin, Germany. *Meteorologische Zeitschrift*, *26*(5), 525–547. https://doi.org/10.1127/metz/2017/0861

Georgi, N. J., & Zafiriadis, K. (2006). The impact of park trees on microclimate in urban areas. *Urban Ecosystems*, *9*(3), 195–209. https://doi.org/10.1007/s11252-006-8590-9

Lefebvre, W., Vercauteren, J., Schrooten, L., Janssen, S., Degraeuwe, B., Maenhaut, W., de Vlieger, I., Vankerkom, J., Cosemans, G., Mensink, C., Veldeman, N., Deutsch, F., Van Looy, S., Peelaerts, W., & Lefebre, F. (2011). Validation of the MIMOSA-AURORA-IFDM model chain for policy support: Modeling concentrations of elemental carbon in Flanders. *Atmospheric Environment*, *45*(37), 6705–6713. https://doi.org/10.1016/j.atmosenv.2011.08.033

Logan, T. M., Zaitchik, B., Guikema, S., & Nisbet, A. (2020). Night and day: The influence and relative importance of urban characteristics on remotely sensed land surface temperature. *Remote Sensing of Environment*, *247*(June), 111861. https://doi.org/10.1016/j.rse.2020.111861

Mandement, M., & Caumont, O. (2020). Contribution of personal weather stations to the observation of deep-convection features near the ground. *Natural Hazards and Earth System Sciences*, *20*(1), 299–322. https://doi.org/10.5194/nhess-20-299-2020

Napoly, A., Grassmann, T., Meier, F., & Fenner, D. (2018). Development and Application of a Statistically-Based Quality Control for Crowdsourced Air Temperature Data. *Frontiers in Earth Science*, *6*(August), 1–16. https://doi.org/10.3389/feart.2018.00118

Nipen, T. N., Seierstad, I. A., Lussana, C., Kristiansen, J., & Hov, Ø. (2020). Adopting citizen observations in operational weather prediction. *Bulletin of the American Meteorological Society*, *101*(1), E43–E57. https://doi.org/10.1175/BAMS-D-18-0237.1

Sgoff, C., Acevedo, W., Paschalidi, Z., Ulbrich, S., Bauernschubert, E., Kratzsch, T., & Potthast, R. (2022). Assimilation of crowd-sourced surface observations over Germany in a regional weather prediction system. *Quarterly Journal of the Royal Meteorological Society*, *148*(745), 1752–1767. https://doi.org/10.1002/qj.4276

Stewart, I. D. (2011). A systematic review and scientific critique of methodology in modern urban heat island literature. *International Journal of Climatology*, *31*, 200–217. https://doi.org/10.1002/joc.2141

Troeyer, K. De, Bauwelinck, M., Aerts, R., Profer, D., Berckmans, J., Delcloo, A., Hamdi, R., Schaeybroeck, B. Van, Hooyberghs, H., Lauwaet, D., Demoury, C., & Nieuwenhuyse, A. Van. (2020). Heat related mortality in the two largest Belgian urban areas : A time series analysis. *Environmental Research*, *188*(February), 109848. https://doi.org/10.1016/j.envres.2020.109848

Venter, Z. S., Chakraborty, T., & Lee, X. (2021). Crowdsourced air temperatures contrast satellite measures of the urban heat island and its mechanisms. *Science Advances*, *7*(22), 1–9. https://doi.org/https://dx.doi.org/10.1126/sciadv.abb9569

Verdonck, M. L., Demuzere, M., Hooyberghs, H., Beck, C., Cyrys, J., Schneider, A., Dewulf, R., & Van Coillie, F. (2018). The potential of local climate zones maps as a heat stress assessment tool, supported by simulated air temperature data. *Landscape and Urban Planning*, *178*(July 2017), 183–197. https://doi.org/10.1016/j.landurbplan.2018.06.004

Wang, J., Zhou, W., & Jiao, M. (2022). Location matters: planting urban trees in the right places improves cooling. *Frontiers in Ecology and the Environment*, *20*(3), 147–151. https://doi.org/https://doi.org/10.1002/fee.2455

Wang, Q., Wang, X., Zhou, Y., Liu, D., & Wang, H. (2022). The dominant factors and influence of urban characteristics on land surface temperature using random forest algorithm. *Sustainable Cities and Society*, *79*(January), 103722. https://doi.org/10.1016/j.scs.2022.103722

Ziter, C. D., Pedersen, E. J., Kucharik, C. J., & Turner, M. G. (2019). Scale-dependent interactions between tree canopy cover and impervious surfaces reduce daytime urban heat during summer. *Proceedings of the National Academy of Sciences*, *116*(15), 7575–7580. https://doi.org/10.1073/pnas.1817561116

---

## Author Response (AR2)

**Author's response to comments revised submission – 2022-09-29 (version 5)**

**Authors' response to editor - manuscript ESSD-2022-113**

Responses to Editor comments: The text in blue represents the authors' reply. Text in italic represents text that was changed in the manuscript, with line numbers referring to those in the revised manuscript.

1.  According to our rules (https://www.earth-system-science-data.net/policies/data_policy.html) the data citation is mandatory (e.g., Wagner et al., 2020). Please include citations to each DOI.

**Reply:** Citations of the Leuven.cool dataset (Beele et al., 2022) have been added to each DOI. Since the dataset was not yet cited in the text, the DOI and citation have been added to section 2.2.

*'... After evaluation, the QC method is applied to the data of the Leuven.cool network, making it a very suitable data set to study in detail local weather phenomena such as the urban heat island (UHI) effect. (https://doi.org/10.48804/SSRN3F (Beele et al., 2022))'* (Lines 28-29)

*'The data is currently available from July 2019 (2019Q3) until December 2021 (2021Q4) (https://doi.org/10.48804/SSRN3F (Beele et al., 2022)).'* (Lines 189-190)

2.  Regarding the figure B2: with the next revision, please check if a copyright statement/image credit is required and add it to the figure caption, if applicable. If you are the originator, you can just inform us via email.

**Reply:** Thanks for the good remark. Figure B2 had a different background map (OpenStreetMap) compared to the background used in Figure 2 (ESRI Topographic Map). To improve the uniformity of the figures, the background of figure B2 was changed accordingly. An additional citation for the ESRI Topographic Map (ESRI World Topographic Map, 2022).

[Figure]

*Figure B2: Training areas used for the creation of the LCZ map. Delineation is based on land use, building height, building density and green ratio data. Background map: ESRI.*

(Figure B2, line 770)

**Additional note:** If the editor/type setters would find the manuscript too lengthy, all appendices can easily be moved to the supplementary materials. Since they were not reviewed as supplementary materials, we currently incorporated them as appendices.

**Additional note:** The order of the subfigures in Figure D1, Figure D2 and Figure D3 has been changed around to match the order of the figures in the main text. The accompanying caption should be sufficient to explain each subfigure.

**Additional note:** Some typos have been corrected.

**References**

Beele, E., Reyniers, M., Aerts, R., and Somers, B.: Replication Data for: Quality control and correction method for air temperature data from a citizen science weather station network in Leuven, Belgium, https://doi.org/https://doi.org/10.48804/SSRN3F, 2022.

ESRI World Topographic Map: http://www.arcgis.com/home/item.html?id=30e5fe3149c34df1ba922e6f5bbf808f, last access: 1 March 2022.